# Provably Efficient Reinforcement Learning in Partially Observable Dynamical Systems

**Masatoshi Uehara**
Cornell University
mu223@cornell.edu

**Ayush Sekhari**
MIT
sekhari@mit.edu

**Nathan Kallus**
Cornell University
kallus@cornell.edu

**Jason D. Lee**
Princeton University
jasonlee@princeton.edu

**Wen Sun**
Cornell University
ws455@cornell.edu

## Abstract

We study Reinforcement Learning for partially observable dynamical systems using function approximation. We propose a new *Partially Observable Bilinear Actor-Critic framework*, that is general enough to include models such as observable tabular Partially Observable Markov Decision Processes (POMDPs), observable Linear-Quadratic-Gaussian (LQG), Predictive State Representations (PSRs), as well as a newly introduced model Hilbert Space Embeddings of POMDPs and observable POMDPs with latent low-rank transition. Under this framework, we propose an actor-critic style algorithm that is capable of performing agnostic policy learning. Given a policy class that consists of memory based policies (that look at a fixed-length window of recent observations), and a value function class that consists of functions taking both memory and future observations as inputs, our algorithm learns to compete against the best memory-based policy in the given policy class. For certain examples such as undercomplete observable tabular POMDPs, observable LQGs and observable POMDPs with latent low-rank transition, by implicitly leveraging their special properties, our algorithm is even capable of competing against the globally optimal policy without paying an exponential dependence on the horizon in its sample complexity.

## 1 Introduction

Large state space and partial observability are two key challenges of Reinforcement Learning (RL). While recent advances in RL for fully observable systems have focused on the challenge of scaling RL to large state space in both theory and in practice using rich function approximation, the understanding of large-scale RL under partial observability is still limited. In POMDPs, for example, a core issue is that the optimal policy is not necessarily Markovian since the observations are not Markovian.

A common heuristic to tackle large-scale RL with partial observability in practice is to simply maintain a time window of the history of observations, which is treated as a state to feed into the policy and the value function. Such a window of history can be often maintained explicitly via truncating away older history (e.g., DQN uses a window with length 4 for playing video games [52]; Open AI Five uses a window with length 16 for LSTMs [5]). Since even for planning under partial observations and known dynamics, finding the globally optimal policy conditional on the entire history is generally NP-hard (due to the curse of the history) [46, 55, 23], searching for a short memory-based policy can be understood as a reasonable middle ground that balances computation and optimality. The impressive empirical results of these prior works also demonstrate that in practice, there often exists a high-quality policy (not necessarily the globally optimal) that is only a function of a short window of recent observations. However, these prior works that search for the best memory-based policy unfortunately cannot ensure sample efficient PAC guarantees due to the difficulty of strategic exploration in POMDPs. The key question that we aim to answer in this work is:

36th Conference on Neural Information Processing Systems (NeurIPS 2022).

*Can we design provably efficient RL algorithms that agnostically learn the best fixed-length memory-based policy with function approximation?*

We provide affirmative answers to the above question. More formally, we study RL for partially observable dynamical systems that include not only the classic Partially Observable MDPs (POMDPs) [53, 56, 58], but also a more general model called Predictive State Representations (PSRs) [45]. We design a model-free actor-critic framework, named *PO-Bilinear Actor-Critic Class*, where we have a policy class (i.e., actors) that consists of policies that take a fixed-length window of observations as input (memory-based policy), and a newly introduced future-dependent value function class (i.e., critics) that consists of functions that take the fixed-length window of history and (possibly multi-step if the system is overcomplete) *future observations* as inputs. A future-dependent value function class is an analog of the value function class tailored to partially observable systems that only relies on observable quantities (i.e., past and future observations and actions). In our algorithm, we *agnostically* search for the best memory-based policy from the given policy class.

Our framework is based on the idea of a newly introduced notion of *future-dependent value function* equipped with future observations. While the idea of using future observations has been used in the literature on POMDPs, our work is the first to use this idea to learn a high-quality policy in a model-free manner. Existing works discuss how to use future observations only in a model-based manner [8, 27]. By leveraging these model-based viewpoints, while recent works discuss strategic exploration to learn near-optimal policies, their results are either limited to the tabular setting (and are not scalable for large state spaces) [36, 24, 3, 74, 47] or are tailored to specific non-tabular models and unclear how to incorporate general function approximation [60, 42, 9]. We break these barriers by devising a new actor-critic-based model-free view on POMDPs. We demonstrate the *scalability* and *generality* of our PO-bilinear actor-critic framework by showing PAC-guarantee on many models as follows (see Table 1 for a summary).

**Observable Tabular POMDPs.** In tabular observable POMDPs, i.e., POMDPs where *multi-step* future observations retain information about latent states, the PO-bilinear rank decomposition holds. We can ensure the sample complexity is $\mathrm{Poly}(S, A^M, O^M, A^K, O^K, H, 1/\sigma_1)$ where $\sigma_1 = \min_x \|\mathbb{O}x\|_1/\|x\|_1$ ($\mathbb{O}$ is an emission matrix),and $S, A, O$ are the cardinality of state, action, observation space, respectively, $H$ is the horizon, and $K$ is the number of future observations.[1] In the special undercomplete ($O \geq S$) case, our framework is also flexible enough to set the memory length according to the property of the problems in order to search for the globally optimal policy. More specifically, using the latest result from [23] about belief contraction, we can set $M = \tilde{O}((1/\sigma_1^4)\ln(SH/\epsilon))$ with $\epsilon$ being the optimality threshold. This allows us to compete against the globally optimal policy without paying an exponential dependence on $H$.

**Observable Linear Quadratic Gaussian (LQG).** In observable LQG – a classic partial observable linear dynamical system, our algorithm can compete against the *globally optimal policy* with a sample complexity scaling polynomially with respect to the horizon, dimensions of the state, observation, and action spaces (and other system parameters). This is achieved by simply setting the memory length $M$ to $H$. The special linear structures of the problem allow us to avoid exponential dependence on $H$ even when using the full history as a memory. While the global optimality results in tabular POMDPs and LQG exist by using different algorithms, *to the best of our knowledge, this is the first unified algorithm that can solve both tabular POMDPs and LQG simultaneously without paying an exponential dependence on horizon $H$.*

**Observable Hilbert Space Embedding POMDPs (HSE-POMDPs).** Our framework ensures the agnostic PAC guarantee on HSE-POMDPs where policy induced transitions and omission distributions have condition mean embeddings [8, 7]. This model naturally generalizes tabular POMDPs and LQG. We show that the sample complexity scales polynomially with respect to the dimensions of the embeddings. This is the *first* PAC guarantee in HSE-POMDPs.

**Predictive State Representations (PSRs).** We give the *first* PAC-guarantee on PSRs. PSRs model partially observable dynamical systems without even using the concept of latent states and strictly generalize the POMDP model. Our work significantly generalizes a prior PAC learning result for

---

[1] In Section G, we discuss how to get rid of $O^M, O^K$ using a model-based learning perspective. The intuition is that a tabular POMDP's model complexity has nothing to do with $M$ or $K$, i.e., number of parameters in transition and omission distribution is $S^2A + OA$ (even if we consider the time-inhomogeneous setting, it scales with $H(S^2A + OA)$, but no $O^M$ and $O^K$) and the PO-bilinear rank is still $S$.

| Model | Observable tabular POMDPs | Observable LQG | Low-rank $M$-step decodable POMDPs | Observable HSE-POMDPs | PSRs | Low rank observable POMDPs |
|---|---|---|---|---|---|---|
| PO-Bilinear Rank | $(OA)^M S(\dagger)$ (Can be $S$) | $O(Md_a^2 d_s^2)(\dagger)$ | Rank ($\dagger$) | Feature dimension on $(z,s)$ | $(OA)^M \times$ # of core tests | Rank ($\dagger$) |
| PAC Learning | Known | Known | Known | New | New | New |

Table 1: Summary of settings that are from PO-Bilinear AC class. The 2nd row gives the parameters that bound the PO-Bilinear rank. Here $M$ denotes the length of memory used to define memory-based policies $\pi(\cdot|\bar{z}_h)$ where $\bar{z}_h = (o_{h-M:h}, a_{h-M:h-1})$ denotes the $M$-step memory. In the 3rd row, "known" means that sample-efficient algorithms already exist. "New" means our result gives the first sample-efficient algorithm. However, even in "known" case, agnostic guarantees are new; hence, when the policy class is small, we can gain some benefit. The symbol $\dagger$ means we can compete with the globally optimal policy without paying an exponential dependence on horizon $H$. For the tabular case, the PO-bilinear rank can be improved to $S$ when we use the most general definition (Refer to Section G. For LQG, $d_a$ and $d_s$ are the dimension of action and state spaces. For PSRs, $O$ and $A$ denote the size of observation and action spaces.

reactive PSRs (i.e., reactive PSRs require a strong condition that the optimal policy only depends on the latest observation) which is a much more restricted setting [33].

**$M$-step decodable POMDPs [19].** Our framework can capture $M$-step decodable POMDPs where there is a (unknown) decoder that can perfectly decode the latent state by looking at the latest $M$-memory. Our algorithm can compete against the globally optimal policy with the sample complexity scaling polynomially with respect to horizon $H$, $S$, $A^M$, and the statistical complexities of function classes, without any explicit dependence on $O$. This PAC result is similar to the one from [19].

**Observable POMDPs with low-rank latent transition.** Our framework captures observable POMDPs where the latent transition is low-rank. This is the *first* PAC guarantee in this model. Under this model, we first show that with $M = \tilde{O}\left((1/\sigma_1^4)\ln(dH/\epsilon)\right)$ where $d$ is the rank of the latent transition matrix, there exists an $M$-memory policy that is $\epsilon$-near optimal with respect to the globally optimal policy. Then, starting with a general model class that contains the ground truth transition and omission distribution (i.e., realizability in model class), we first convert the model class to a policy class and a future-dependent value function class, and we then show that our algorithm competes against the globally optimal policy with a sample complexity scaling polynomially with respect to $H, d, |\mathcal{A}|^{(1/\sigma_1^4)\ln(dH/\epsilon)}, 1/\sigma_1$, and the statistical complexity of the model class. Particularly, the sample complexity has no explicit dependence on the size of the state and observation space, instead it just depends on the statistical complexity of the given model class.

## 1.1 Related Works

We discuss related works about online RL for POMDPs. Additional works related to system identification, generalization and function approximation of RL in MDPs, PSRs and future-dependent value functions are provided in Section A.

Prior works [38, 20] showed $A^H$-type sample complexity bounds for general POMDPs. Exponential dependence can be circumvented with more structures. First, in the tabular setting, under observability assumptions, in [3, 24, 34, 47, 22], favorable sample complexities are obtained by leveraging the spectral learning technique [29] (see section 1.1 in [34] for an excellent summary). Second, in LQG, which is a partial observable version of LQRs, in [42, 60], sub-linear regret algorithms are proposed. These works use random policies for exploration, which is sufficient for LQG. Since random exploration strategy is not enough for tabular POMDPs, it is unclear if the existing techniques from LQG can be applied to solve general POMDPs. Third, the recent work [19] provides a new model called $M$-step decodable POMDP (when $M = 1$, it is Block MDP) with an efficient algorithm.

Our framework captures *all* above mentioned POMDP models. In addition, we propose a new model called HSE-POMDPs which extends prior works on HSE-HMM[7] to POMDPs and includes LQG and tabular POMDPs. Our algorithm delivers the first PAC bound for this model. Finally, we remark it is unclear whether our framework can capture several existing POMDP models [9, 41].

## 2 Preliminary

We introduce the background for POMDPs. We consider an episodic POMDP specified by $\mathcal{M} = \langle \mathcal{S}, \mathcal{O}, \mathcal{A}, H, \mathbb{T}, \mathbb{O} \rangle$, where $\mathcal{S}$ is the *unobserved* state space, $\mathcal{O}$ is the observation space, $\mathcal{A}$ is the action space, $H$ is the horizon, $\mathbb{T} : \mathcal{S} \times \mathcal{A} \rightarrow \Delta(\mathcal{S})$ is the transition probability, $\mathbb{O} : \mathcal{S} \rightarrow \Delta(\mathcal{O})$ is the emission probability, and $r : \mathcal{O} \times \mathcal{A} \rightarrow \mathbb{R}$ is the reward. Here, $\mathbb{T}, \mathbb{O}$ are unknown distributions.

For notation simplicity, we consider the time-homogeneous case in this paper; Extension to the time-inhomogeneous setting is straightforward.

In our work, we consider $M$-memory policies. Let $\mathcal{Z}_h = (\mathcal{O} \times \mathcal{A})^{\min\{h,M\}}$ and $\bar{\mathcal{Z}}_h = \mathcal{Z}_{h-1} \times \mathcal{O}$. An element $z_h \in \mathcal{Z}_h$ is represented as $z_h = [o_{\max(h-M+1,1):h}, a_{\max(h-M+1,1):h}]$, and an element $\bar{z}_h \in \bar{\mathcal{Z}}_h$ is represented as $\bar{z}_h = [o_{\max(h-M,1):h}, a_{\max(h-M,1):h-1}]$ (thus, $\bar{z}_h = [z_{h-1}, o_h]$). Figure 2 illustrates this situation. An $M$-memory policy is defined as $\pi = \{\pi_h\}_{h=1}^H$ where each $\pi_h$ is a mapping from $\bar{\mathcal{Z}}_h$ to a distribution over actions $\Delta(\mathcal{A})$.

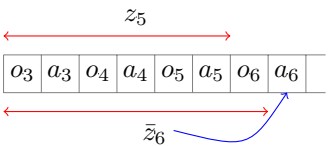

Figure 1: Case with M= 3. A 3-memory policy determines action $a_6$ based on $\bar{z}_6$.

In a POMDP, an $M$-memory policy generates the data as follows. Each episode starts with the initial state $s_1$ sampled from some unknown distribution. At each step $h \in [H]$, from $s_h \in \mathcal{S}$, the agent observes $o_h \sim \mathbb{O}(\cdot|s_h)$, executes action $a_h \sim \pi_h(\cdot|\bar{z}_h)$, receives reward $r(s_h, a_h)$, and transits to the next latent state $s_{h+1} \sim \mathbb{T}(\cdot|s_h, a_h)$. Note that the agent does not observe the underlying states but only the observations $\{o_h\}_{h \leq H}$. We denote $J(\pi)$ as the value of the policy $\pi$, i.e., $\mathbb{E}[\sum_{h=1}^H r_h; a_{1:H} \sim \pi]$ where the expectation is taken w.r.t. the stochasticity of the policy $\pi$, emissions distribution $\mathbb{O}$ and transition dynamics $\mathbb{T}$.

We define a value function for a policy $\pi$ at step $h$ to be the expected cumulative reward to go under the policy $\pi$ starting from a $z \in \mathcal{Z}_{h-1}$ and $s \in \mathcal{S}$, i.e. $V_h^\pi : \mathcal{Z}_{h-1} \times \mathcal{S} \to \mathbb{R}$ where $V_h^\pi(z,s) = \mathbb{E}[\sum_{h'=h}^H r_{h'} \mid z_{h-1} = z, s_h = s; a_{h:H} \sim \pi]$. The notation $\mathbb{E}[\cdot; a_{h:H} \sim \pi]$ means the expectation is taken under a policy $\pi$ from $h$ to $H$. Compared to the standard MDP setting, the expectation is conditional on not only $s_h$ but also $z_{h-1}$ since we consider $M$-memory policies. The corresponding Bellman equation for $V_h^\pi$ is $V_h^\pi(z_{h-1}, s_h) = \mathbb{E}\left[r_h + V_{h+1}^\pi(z_h, s_{h+1}) \mid z_{h-1}, s_h; a_h \sim \pi\right]$

**The Actor-critic function approximation setup.** Our goal is to find a near optimal policy that maximizes the policy value $J(\pi)$ in an online manner. Since any POMDPs can be converted into MDPs by setting the state at level $h$ to the observable history up to $h$, any off-the-shelf online provably efficient algorithms for MDPs can be applied to POMDPs. By defining $\mathcal{H}_h$ as the whole history up to step $h \in [H]$ (i.e., a history $\tau_h \in \mathcal{H}_h$ is in the form of $o_{1:h}, a_{1:h-1}$), these naïve algorithms ensure that output policies can compete against the globally optimal policy $\pi_{\mathrm{gl}}^\star = \mathrm{argmax}_{\pi \in \bar{\Pi}} J(\pi)$ where $\tilde{\Pi} = \{\bar{\Pi}_h\}, \tilde{\Pi}_h = [\mathcal{H}_h \to \Delta(\mathcal{A})]$. However, this conversion results in the error with exponential dependence on the horizon $H$, which is prohibitively large in the long horizon setting.

Instead of directly competing against the globally optimal policy, we aim for *agnostic policy learning*, i.e., compete against the best policy in a given $M$-memory policy class. Our function approximation setup consists of two function classes, $(a)$ A policy class $\Pi$ consisting of $M$-memory policies $\Pi := \{\Pi_h\}_{h=1}^H$ where $\Pi_h \subset [\bar{\mathcal{Z}}_h \to \Delta(\mathcal{A})]$ (i.e., actors), $(b)$ A set of value future-dependent value functions $\mathcal{G} = \{\mathcal{G}_h\}_{h=1}^H$ where $\mathcal{G}_h \subset [\bar{\mathcal{Z}}_h \to \mathbb{R}]$, whose role is to approximate $V_h^\pi$ (i.e., critics). Our goal is to provide an algorithm that outputs a policy $\hat{\pi} = \{\hat{\pi}_h\}$ that has a low excess risk, where excess risk is defined by $R(\pi) := J(\hat{\pi}) - J(\pi^\star)$ where $\pi^\star = \mathrm{argmax}_{\pi \in \Pi} J(\pi)$ is the best policy in class $\Pi$. To motivate this agnostic setting, $M$-memory policies are also widely used in practice, e.g., DQN [52] sets $M = 4$. Besides, there are natural examples where $M$-memory policies are close to the globally optimal policy with $M$ being only polynomial with respect to other problem dependent parameters, e.g., observable POMDPs [23] and LQG [42, 60, 48]. We will show the global optimality in these two examples later, without any exponential dependence on $H$ in the sample complexity.

**Remark 1** (Limits of existing MDP actor-critic framework). *While general actor-critic framework proposed in MDPs [33] is applicable to POMDPs via the naïve POMDP to MDP reduction, it is unable to leverage any benefits from the restricted policy class. This naïve reduction (from POMDP to MDP) uses full history and will incur sample complexity that scales exponentially in the horizon.*

**Additional notation.** Let $[H] = \{1, \cdots, H\}$ and $[t] = \{1, \cdots, t\}$. Give a matrix $A$, we denote its pseudo inverse by $A^\dagger$ and the operator norm by $\|A\|$. We define the $\ell_1$ norm $\|A\|_1 = \max_{x:x\neq 0} \|Ax\|_1/\|x\|_1$. The outer product is denoted by $\otimes$. Let $d_h^\pi(\cdot) \in \bar{Z}_h \times \mathcal{S}$ be the marginal distribution at $h$ and $\delta(\cdot)$ be the Dirac delta function. We denote the policy $\delta(a = a')$ by $\mathrm{do}(a')$. We denote a uniform action by $\mathcal{U}(\mathcal{A})$. Given a function class $\mathcal{G}$, we define $\|\mathcal{G}\|_\infty = \sup_{g \in \mathcal{G}} \|g\|_\infty$.

# 3 Future-Dependent Value Functions and the PO-bilinear Framework

Unlike MDPs, we cannot directly work with value functions $V_h^\pi(s)$ in POMDPs, since they depend on the unobserved state $s$. To handle this issue, below we first introduce new future-dependent value functions by using future observations, and then discuss the PO-bilinear framework.

## 3.1 Future-Dependent Value Functions

**Definition 1** (K-step future-dependent value functions). *Fix a set of policies $\pi^{out} = \{\pi_i^{out}\}_{i=1}^K$ where $\pi_i^{out} : \mathcal{O} \to \Delta(\mathcal{A})$. Value future-dependent value functions $g_h^\pi : \mathcal{Z}_{h-1} \times \mathcal{O}^K \times \mathcal{A}^{K-1} \to \mathbb{R}$ at step $h \in [H]$ for a policy $\pi$ are defined as the solution to the following integral equation:*

$$\forall z_{h-1} \in \mathcal{Z}_{h-1}, s_h \in \mathcal{S}, \qquad \mathbb{E}[g_h^\pi(z_{h-1}, o_{h:h+K-1}, a_{h:h+K-2}) \mid z_{h-1}, s_h; a_{h:h+K-2} \sim \pi^{out}] = V_h^\pi(z_{h-1}, s_h),$$

*where the expectation is taken under the policy $\pi^{out}$.*

Future-dependent value functions do not necessarily exist, nor are needed to be unique. At an intuitive level, K-step future-dependent value functions are embeddings of the value functions onto the observation space, and its existence essentially means that K-step futures have sufficient information to recover the latent state dependent value function. The proper choice of $\pi^{out}$ would depend on the underlying models. For example, we use uniform policy in the tabular case, and $\delta(a = 0)$ in LQG. For notational simplicity, we mostly focus on the case of $K = 1$, though we will also discuss the general case of $K \geq 2$. The simplified definition for 1-step future-dependent value functions is provided in the following. Note that this definition is agnostic to $\pi^{out}$.

**Definition 2** (1-step future-dependent value functions). *One-step future-dependent value functions $g_h^\pi : \mathcal{Z}_{h-1} \times \mathcal{O} \to \mathbb{R}$ at step $h \in [H]$ for a policy $\pi$ are defined as the solution to the following:*

$$\forall z_{h-1} \in \mathcal{Z}_{h-1}, s_h \in \mathcal{S} : \qquad \mathbb{E}[g_h^\pi(z_{h-1}, o_h) \mid z_{h-1}, s_h] = V_h^\pi(z_{h-1}, s_h). \qquad (1)$$

In Section 4, we will demonstrate the form of the future-dependent value function for various examples. The idea of encoding latent state information using the statistics of (multi-step) futures have been widely used in learning models of HMMs [63, 29], PSRs [8, 7, 27, 67], and system identification [72]. Existing provably efficient (online) RL works for POMDPs elaborate on this viewpoint [36, 24, 3]. Compared to them, the novelty of future-dependent value functions is that it is introduced to recover *value functions* but not *models*. This model-free view differs from the existing dominant model-based view in online RL for POMDPs. In our setup, we can control systems if we can recover value functions on the underlying states even if we fail to identify the underlying model.

## 3.2 The PO-Bilinear Actor-critic Framework for POMDPs

With the definition of future-dependent value functions, we are now ready to introduce the PO-bilinear actor-critic (AC) class for POMDPs. We will focus on the case of $K = 1$ here. Let $\mathcal{G} = \{\mathcal{G}_h\}_{h=1}^H$, where $\mathcal{G}_h \subset [\bar{\mathcal{Z}}_h \to \mathbb{R}]$, be a class consisting of functions that satisfy the following realizability assumption w.r.t. the policy class $\Pi$.

**Assumption 1** (Realizability). *We assume that $\mathcal{G}$ is realizable w.r.t. the policy class $\Pi$, i.e., $\forall \pi \in \Pi, h \in [H]$, there exists at least one $g_h^\pi \in \mathcal{G}_h$ such that $g_h^\pi$ is a future-dependent value function w.r.t. the policy $\pi$. Note that realizability implicitly requires the existence of $(g_h^\pi)$.*

We next introduce the PO-Bilinear Actor-critic class. For each level $h \in [H]$, we first define the Bellman loss:

$$\mathrm{Br}_h(\pi, g; \pi^{in}) := \mathbb{E}[g_h(\bar{z}_h) - r_h - g_{h+1}(\bar{z}_{h+1}) : a_{1:h-1} \sim \pi^{in}, a_h \sim \pi]$$

given M-memory policies $\pi = \{\pi_h\}, \pi^{in} = \{\pi_h^{in}\}$ and $g = \{g_h\}$. Letting $g^\pi = \{g_h^\pi\}_{h=1}^H$ be a future-dependent value function for $\pi$, our key observation is that future-dependent value functions satisfy $0 = \mathrm{Br}_h(\pi, g^\pi; \pi^{in})$ for any M memory roll-in policy $\pi^{in} = \{\pi_h^{in}\}_{h=1}^H$, and any evaluation pair $(\pi, g^\pi)$. This is an analog of Bellman equations on MDPs. The above equation tells us that $\mathrm{Br}_h(\pi, g; \pi^{in})$ is a right loss to quantify how much the estimator $g$ is different from $g_h^\pi$. When $\mathrm{Br}_h(\pi, g; \pi^{in})$ has a low-rank structure in a proper way, we can efficiently learn a near-optimal M memory policy. The following definition precisely quantifies the low-rank structure that we need for sample efficient learning.

**Definition 3** (PO-bilinear AC Class, $K = 1$). *The model is a PO-bilinear Actor-critic class of rank $d$ if $\mathcal{G}$ is realizable, and there exist $W_h : \Pi \times \mathcal{G} \to \mathbb{R}^d$ and $X_h : \Pi \to \mathbb{R}^d$ such that for all $\pi', \pi \in \Pi, g \in \mathcal{G}$ and $h \in [H]$,*

1. $\mathbb{E}[g_h(\bar{z}_h) - r_h - g_{h+1}(\bar{z}_{h+1}); a_{1:h-1} \sim \pi', a_h \sim \pi] = \langle W_h(\pi, g), X_h(\pi') \rangle$.

2. $W_h(\pi, g^\pi) = 0$ for any $\pi \in \Pi$ and the corresponding future-dependent value function $g^\pi \in \mathcal{G}$.

*We define $d$ as the PO-bilinear rank.*

**Remark 2** (Two Important Extensions). *While the above definition is enough to capture most of the examples we discuss later in this work, including undercomplete tabular POMDPs, LQG, HSE-POMDPs, we provide two useful extensions. The first extension incorporates discriminators into the framework, which can be used to capture the M-step decodable POMDPs and POMDPs with low-rank latent transition (see Section F). The second extension incorporates multi-step futures, which can be used to capture overcomplete POMDPs and general PSRs (see Section B.*

## 4 Examples of PO-Bilinear Actor-critic Classes

We consider three examples (observable tabular POMDPs, LQG, HSE-POMDPs) that admit PO-bilinear rank decomposition. Our framework can also capture PSRs, $M$-step decodable POMDPs and low rank observable POMDPs, of which the discussions are deferred to Section E, G.1 and G.2, respectively. We mainly focus on one-step future, i.e., $K = 1$, and briefly discuss the extension to $K > 1$ in the tabular case. In this section, except for LQG, we assume $r_h \in [0, 1]$ for any $h \in [H]$. All the missing proofs are deferred to Section C.

### 4.1 Observable Undercomplete Tabular POMDPs

**Example 1** (Observable undercomplete tabular POMDPs). *Let $\mathbb{O} \in \mathbb{R}^{|\mathcal{O}| \times |\mathcal{S}|}$ where the entry indexed by a pair $(o, s)$ is defined as $\mathbb{O}_{o,s} = \mathbb{O}(o|s)$. Assume that $\mathrm{rank}(\mathbb{O}) = |\mathcal{S}|$, which we call observability. This requires undercompletenes $|\mathcal{O}| \geq |\mathcal{S}|$.*

The following lemma shows that $\mathbb{O}$ being full rank implies the existence of future-dependent value functions $g_h^\pi$.

**Lemma 1.** *For Example 1, there exists a $g_h^\pi$ satisfying Definition 2 for any $\pi \in \Pi$ and $h \in [H]$.*

*Proof.* Consider any function $f : \mathcal{Z}_{h-1} \times \mathcal{S} \to \mathbb{R}$ (thus, this captures all possible $V_h^\pi$). Denote $\mathbf{1}(z)$ as the one-hot encoding of $z$ over $\mathcal{Z}_{h-1}$ (similarly for $\mathbf{1}(s)$). We have $f(z, s) = \langle f, \mathbf{1}(z) \otimes \mathbf{1}(s) \rangle = \langle f, \mathbf{1}(z) \otimes (\mathbb{O}^\dagger \mathbb{O} \mathbf{1}(s)) \rangle$, where we use the assumption that $\mathrm{rank}(\mathbb{O}) = |\mathcal{S}|$ and thus $\mathbb{O}^\dagger \mathbb{O} = I$. Then,

$$f(z, s) = \langle f, \mathbf{1}(z) \otimes (\mathbb{O}^\dagger \mathbb{E}_{o \sim O(s)} \mathbf{1}(o)) \rangle = \mathbb{E}_{o \sim O(s)} \langle f, \mathbf{1}(z) \otimes \mathbb{O}^\dagger \mathbf{1}(o) \rangle, \tag{2}$$

which means $g_h^\pi(z, o) := \langle V_h^\pi, \mathbf{1}(z) \otimes \mathbb{O}^\dagger \mathbf{1}(o) \rangle$. $\qquad\qquad\square$

We next show that the PO-Bilinear rank (Definition 3) is bounded by $|\mathcal{S}|(|\mathcal{O}||\mathcal{A}|)^M$.

**Lemma 2.** *Assume $\mathbb{O}$ is full column rank. Set the future-dependent value function class $\mathcal{G}_h = [\mathcal{Z}_{h-1} \times \mathcal{O} \to [0, C_\mathcal{G}]]$ for certain $C_\mathcal{G} \in \mathbb{R}$, and policy class $\Pi_h = [\bar{\mathcal{Z}}_h \to \Delta(\mathcal{A})]$. Then, the model is a PO-biliner AC class (Definition 3) with PO-bilinear rank at most $|\mathcal{S}|(|\mathcal{O}||\mathcal{A}|)^M$.*

Later, we will see that the PO-bilinear rank in the more general definition is just $|\mathcal{S}|$ in Section F. This fact will result in a significant improvement in terms of the sample complexity, and will result in a sample complexity that does not incur $|\mathcal{O}|^M$.

Lastly, we touch on overcomplete POMDPs ($|\mathcal{O}| \leq |\mathcal{S}|$) when we use multi-step futures. For details, refer to Section C.2. In this case, the existence of future-dependent value function is ensured when $|\mathcal{O}|^K |\mathcal{A}|^{K-1} \times |\mathcal{S}|$ matrix with entries equal to $\mathbb{P}(o_{h:h+K-1}, a_{h:h+K-2} \mid s_h; a_{h:h+K-2} \sim U(\mathcal{A}))$.

### 4.2 Observable Linear Quadratic Gaussian

The next example is Linear Quadratic Gaussian (LQG) with continuous state and action spaces. The details are deferred to Section M. Here, we set $M = H - 1$ so that the policy class $\Pi$ contains the globally optimal policy.

**Example 2** (Linear Quadratic Gaussian (LQG)). *Consider LQG:*

$$s' = As + Ba + \epsilon, \ o = Cs + \tau, \ r = -(s^\top Qs + a^\top Ra)$$

*where $\epsilon, \tau$ are Gaussian distribution with mean $0$ and variances $\Sigma_\epsilon$ and $\Sigma_\tau$, respectively, and $s \in \mathbb{R}^{d_s}, o \in \mathbb{R}^{d_o}$, and $a \in \mathbb{R}^{d_a}$, and $Q, R$ are positive definite matrices.*

We define the policy class as the linear policy class $\Pi_h = \{\delta(a_h = K_h \bar{z}_h) \mid K_h \in \mathbb{R}^{|\mathcal{A}| \times d_{\bar{z}_h}}\}$, where $d_{\bar{z}_h}$ is a dimension of $\bar{z}_h \in \bar{\mathcal{Z}}_h$. This choice is natural since the globally optimal policy is known to be linear with respect to the entire history [6, Chapter 4]. We define two quadratic features, $\phi_h(z_{h-1}, s_h) = (1, [z_{h-1}^\top, s_h^\top] \otimes [z_{h-1}^\top, s_h^\top])^\top$ with $z_{h-1} \in \mathcal{Z}_{h-1}, s_h \in \mathcal{S}$, and $\psi_h(z_{h-1}, o_h) = (1, [z_{h-1}^\top, o_h^\top] \otimes [z_{h-1}^\top, o_h^\top])^\top$ with $z_{h-1} \in \mathcal{Z}_{h-1}, o_h \in \mathcal{O}$. We have the following lemma.

**Lemma 3** (PO-bilinear rank of observable LQG). *Assume* $\mathrm{rank}(C) = d_s$. *Then, the following holds. (1) For any policy $\pi$ linear in $\bar{z}_h$, a one-step future-dependent value function $g_h^\pi(\cdot)$ exists, and is linear in $\psi_h(\cdot)$. (2) Letting $d_{\psi_h}$ be the dimension of $\psi_h$, we set $\mathcal{G}_h = \{\theta^\top \psi_h(\cdot)|\theta \in \mathbb{R}^{d_{\psi_h}}\}$ and $\Pi$ being linear in $\bar{z}_h$. Then LQG satisfies Definition 3 with PO-bilinear rank at most $O(\{1 + (H-1)(d_o + d_a) + d_s\}^2)$*

We have two remarks. First, when $\pi_t^{out} = \delta(a = 0)$, K-step future-dependent value functions exist when $[C^\top, (CA)^\top, \ldots, (CA^{K-1})^\top]$ is full raw rank. This assumption is referred to as observability in control theory [28]. Secondly, the PO-bilinear rank scales polynomially with respect to $H, d_o, d_a, d_s$ even with $M = H - 1$. As we show in Section M, due to this fact, we can compete against the *globally* optimal policy with polynomial sample complexity.

### 4.3 Observable Hilbert Space Embedding POMDPs

We consider HSE-POMDPs that generalize tabular POMDPs and LQG. Proofs here are deferred to Section C.4. Consider any $h \in [H]$. Given a policy $\pi_h : \bar{\mathcal{Z}}_h \to \mathcal{A}$, we define the induced transition operator $\mathbb{T}_\pi = \{\mathbb{T}_{\pi;h}\}_{h=1}^H$ as $(z_h, s_{h+1}) \sim \mathbb{T}_{\pi;h}(z_{h-1}, s_h)$, where we have $o_h \sim \mathbb{O}(s_h), a_h \sim \pi_h(\bar{z}_h), s_{h+1} \sim \mathbb{T}(s_h, a_h)$. Namely, $\mathbb{T}_\pi$ is the transition kernel of some Markov chain induced by the policy $\pi$. The HSE-POMDP assumes two conditional distributions $\mathbb{O}(\cdot|s)$ and $\mathbb{T}_\pi(\cdot, \cdot|z, s)$ have conditional mean embeddings.

**Example 3** (HSE-POMDPs). *We introduce features $\phi_h : \mathcal{Z}_{h-1} \times \mathcal{S} \to \mathbb{R}^{d_{\phi_h}}, \psi_h : \mathcal{Z}_{h-1} \times \mathcal{O} \to \mathbb{R}^{d_{\psi_h}}$. We assume the existence of the conditional mean embedding operators: (1) there exists a matrix $K_h$ such that for all $z \in \mathcal{Z}_{h-1}, s \in \mathcal{S}$, $\mathbb{E}_{o \sim \mathbb{O}(\cdot|s)} \psi_h(z, o) = K_h \phi_h(z, s)$ and (2) for all $\pi \in \Pi$, there exists a matrix $T_{\pi;h}$, such that $\mathbb{E}_{z_h, s_{h+1} \sim \mathbb{T}_{\pi;h}(z_{h-1}, s_h)} \phi_{h+1}(z_h, s_{h+1}) = T_{\pi;h} \phi_h(z_{h-1}, s_h)$.*

The existence of conditional mean embedding is a common assumption in prior RL works on learning dynamics of HMMs, PSRs, [64, 7] and Bellman complete linear MDPs [77, 18, 11, 26]. HSE-POMDPs naturally capture tabular POMDPs and LQG. For tabular POMDPs, $\psi_h$ and $\phi_h$ are one-hot encoding features. In LQG, $\phi_h$ and $\psi_h$ are quadratic features we define in Section 4.2. Here for simplicity, we focus on finite-dimensional features $\phi_h$ and $\psi_h$. Extension to infinite-dimensional Reproducing kernel Hilbert Space is deferred to Section C.4.

The following shows the existence of future-dependent value functions and the PO-bilinear rank decomposition.

**Lemma 4** (PO-bilinear rank of observable HSE-POMDPs). *Assume $K_h$ is full column rank (observability), and $V_h^\pi(\cdot)$ is linear in $\phi_h$ for any $\pi \in \Pi, h \in [H]$. Then the following holds. (1) A one-step future-dependent value function $g_h^\pi(\cdot)$ exists for any $\pi \in \Pi, h \in [H]$, and is linear in $\psi_h$. (2) We set a value function class $\mathcal{G}_h = \{w^\top \psi_h(\cdot)|w \in \mathbb{R}^{d_{\psi_h}}\}$, policy class $\Pi_h \subset [\bar{\mathcal{Z}}_h \to \Delta(\mathcal{A})]$. Then HSE-POMDP satisfies Definition 3 with PO-bilinear rank at most $\max_{h \in [H]} d_{\phi_h}$.*

The first statement can be verified by noting that when $V_h^\pi(\cdot) = \langle \theta_h, \phi_h(\cdot) \rangle$, future-dependent value functions take the following form $g_h^\pi(\cdot) = \langle (K_h^\dagger)^\top \theta_h), \psi_h(\cdot) \rangle$ where we leverage the existence of the conditional mean embedding operator $K_h$, and that $K_h$ is full column rank (thus $K_h^\dagger K_h = \mathbb{I}_{d_{\phi_h}}$). Note that the PO-bilinear rank depends only on the dimension of the features $\phi_h$ without any explicit dependence on the length of memory.

## 5 Algorithm and Complexity

In this section, we first give our algorithm followed by a general sample complexity analysis. We then instantiate our analysis to specific models considered in Section 4.

### 5.1 Algorithm

We first focus on the cases where models satisfy the PO-bilinear AC model (i.e., Definition 3) with finite action and with one-step future-dependent value function.

---

**Algorithm 1** PaRtially ObserVAble BiLinEar (PROVABLE) # multi-step version is in Algorithm 2

---

1: **Input:** Value class $\mathcal{G} = \{\mathcal{G}_h\}, \mathcal{G}_h \subset [\mathcal{Z}_{h-1} \to \mathbb{R}]$, Policy class $\Pi = \{\Pi_h\}, \Pi_h \subset [\bar{\mathcal{Z}}_{h-1} \to \mathbb{R}]$, parameters $m \in \mathbf{N}, R \in \mathbb{R}$, Initialize $\pi^0 \in \Pi$

2: Form the first step dataset $\mathcal{D}^0 = \{o^i\}_{i=1}^m$, with $o^i \sim \mathbb{O}(\cdot|s_1)$

3: **for** $t = 0 \to T - 1$ **do**

4:      For any $h \in [H]$, collect $m$ i.i.d tuple as follows: $(\bar{z}_h, s_h) \sim d_h^{\pi^t}, a_h \sim \mathcal{U}(\mathcal{A}), r_h = r_h(o_h, a_h), s_{h+1} \sim \mathbb{T}(s_h, a_h), o_{h+1} \sim \mathbb{O}(\cdot|s_{h+1})$.

5:      Define $\mathcal{D}_h^t = \{(\bar{z}_h^i, a_h^i, r_h^i, o_{h+1}^i)\}_{i=1}^m$      # note latent state $s$ is not in the dataset

6:      Define the Bellman error $\forall(\pi, g) \in \Pi \times \mathcal{G}$,

$$\sigma_h^t(\pi, g) := \mathbb{E}_{\mathcal{D}_h^t} \left[ \pi_h(a_h \mid \bar{z}_h) |\mathcal{A}| \{g_{h+1}(\bar{z}_{h+1}) + r_h\} - g_h(\bar{z}_h) \right].$$

7:      Select policy optimistically as follows

$$(\pi^{t+1}, g^{t+1}) := \text{argmax}_{\pi \in \Pi, g \in \mathcal{G}} \, \mathbb{E}_{\mathcal{D}^0}[g_1(o)] \quad \text{s.t.} \quad \forall h \in [H], \forall i \in [t], (\sigma_h^i(\pi, g))^2 \leq R.$$

8: **end for**

9: **Output:** Randomly choose $\hat{\pi}$ from $(\pi_1, \cdots, \pi_T)$.

---

We present our algorithm PROVABLE in Algorithm 1. Note PROVABLE is agnostic to the form of $X_h$ and $W_h$. Inside iteration $t$, given the latest learned policy $\pi^t$, we define Bellman error for all pairs $(\pi, g)$ where the Bellman error is averaged over the samples from $\pi^t$. Here, to evaluate the Bellman loss for any policy $\pi \in \Pi$, we use importance sampling by running $\mathcal{U}(\mathcal{A})$ rather than executing a policy $\pi$ so that we can reuse samples.[2] A pair $(\pi, g)$ that has a small total Bellman error intuitively means that given the data so far, $g$ could still be a value future-dependent value function for the policy $\pi$. Then in the constrained optimization formulation, we only focus on $(\pi, g)$ pairs whose Bellman errors are small so far. Among these $(\pi, g)$ pairs, we select the pair using the principle of optimism in the face of uncertainty. We remark the algorithm leverages some design choices from the Bilinear-UCB algorithm for MDPs [16]. The key difference between our algorithm and the Bilinear-UCB is that we leverage the actor-critic framework equipped with value future-dependent value functions to handle partially observability and agnostic learning.

**Remark 3** (Three Important Extensions). *By extending Algorithm 1, we can consider more general algorithms to include three important cases. The first extension is the minimax version with discriminators to capture low-rank observable PMODPs and M-step decodable POMDPs. The detail is in Section P. Secondly, although algorithms so far implicitly assume the action is finite, we can consider LQG with continuous action by employing a G-optimal design over actions. The detail is in Section D.2. The third extension is the multi-step future case, which can capture overcomplete POMDPs and general PSRs. The discussion is deferred to Section D.*

### 5.2 Sample Complexity

We show a sample complexity result by using reduction to supervised learning analysis. We begin by stating the following assumption which is ensured by standard uniform convergence results.

**Assumption 2** (Uniform Convergence). *Fix $h \in [H]$. Let $\mathcal{D}_h'$ be a set of $m$ i.i.d tuples following $(z_{h-1}, s_h, o_h) \sim d_h^{\pi^t}, a_h \sim \mathcal{U}(\mathcal{A}), s_{h+1} \sim \mathbb{T}(s_h, a_h), o_{h+1} \sim \mathbb{O}(s_{h+1})$. With probability $1 - \delta$,*

$$\sup_{\pi \in \Pi, g \in \mathcal{G}} |(\mathbb{E}_{\mathcal{D}_h'} - \mathbb{E})[\pi_h(a_h \mid \bar{z}_h)|\mathcal{A}|\{g_{h+1}(\bar{z}_{h+1}) + r_h\} - g_h(\bar{z}_h)]| \leq \epsilon_{gen,h}(m, \Pi, \mathcal{G}, \delta)$$

*For $h = 1$, we also require $\sup_{g_1 \in \mathcal{G}_1} |\mathbb{E}_{\mathcal{D}_1'}[g_1(o_1)] - \mathbb{E}[\mathbb{E}_{\mathcal{D}_1'}[g_1(o_1)]]| \leq \epsilon_{ini,1}(m, \mathcal{G}, \delta)$.*

**Remark 4** (Finite function classes). *The term $\epsilon_{gen}$ depends on the statistical complexities of the function classes $\Pi, \mathcal{G}$. As a simple example, we consider the case where $\Pi$ and $\mathcal{G}$ are discrete. In this case, we have $\epsilon_{gen,h}(m, \Pi, \mathcal{G}, \delta) = O(\sqrt{\ln(|\Pi||\mathcal{G}|/\delta)/m})$, and $\epsilon_{ini,1}(m, \mathcal{G}, \delta) = O(\sqrt{\ln(|\mathcal{G}|/\delta)/m})$, which are standard statistical complexities for discrete function classes $\Pi$ and $\mathcal{G}$. Achieving this result simply requires standard concentration and a union bound over all functions in $\Pi, \mathcal{G}$.*

Under Assumption 2, when the model is PO-bilinear with rank $d$, we get the following.

---

[2]This choice might limit the algorithm to the case where $\mathcal{A}$ is discrete. However, for examples such as LQG, we show that we can replace $\mathcal{U}(\mathcal{A})$ by a G-optimal design over the quadratic polynomial feature of the actions.

**Theorem 1** (PAC guarantee of PROVABLE). *Suppose we have a PO-bilinear AC class with rank $d$. Suppose Assumption 2, $\sup_{\pi \in \Pi} \|X_h(\pi)\| \le B_X$ and $\sup_{\pi \in \Pi, g \in \mathcal{G}} \|W_h(\pi, g)\| \le B_W$ for any $h \in [H]$. By setting $T = 2Hd \ln\left(4Hd\left(\frac{B_X^2 B_W^2}{\hat{\epsilon}_{gen}^2} + 1\right)\right), R = \epsilon_{gen}^2$ where*

$$\epsilon_{gen} := \max_h \epsilon_{gen,h}(m, \Pi, \mathcal{G}, \delta/(TH+1)), \tilde{\epsilon}_{gen} := \max_h \epsilon_{gen,h}(m, \Pi, \mathcal{G}, \delta/H).$$

*With probability at least $1 - \delta$, letting $\pi^\star = \operatorname{argmax}_{\pi \in \Pi} J(\pi^\star)$, we have*

$$J(\pi^\star) - J(\hat{\pi}) \le 5\epsilon_{gen}\sqrt{dH^2 \cdot \ln\left(4Hd\left(B_X^2 B_W^2/\tilde{\epsilon}_{gen}^2 + 1\right)\right)} + 2\epsilon_{ini,1}(m, \mathcal{G}, \delta/(TH+1)).$$

*The total number of samples used in the algorithm is $mTH$.*

Informally, when $\epsilon_{gen} \approx \tilde{O}(1/\sqrt{m})$, to achieve $\epsilon$-near optimality, the above theorem indicates that we just need to set $m \approx \tilde{O}(1/\epsilon^2)$, which results a sample complexity scaling $\tilde{O}(1/\epsilon^2)$ (since $T$ only scales $\tilde{O}(dH)$). We give detailed derivation and examples in the next section.

### 5.3 Examples

Hereafter, we show the sample complexity result by using Theorem 1. For complete results, refer to Section I–N.

#### 5.3.1 Finite Sample Classes

We consider the case where the hypothesis class is finite and admits PO-bilinear rank decomposition.

**Example 4** (Finite Sample Classes). *Consider the case when $\Pi$ and $\mathcal{G}$ are finite and the PO-bilinear rank assumption is satisfied. When $\Pi$ and $\mathcal{G}$ are infinite hypothesis classes, $|\mathcal{F}|$ and $|\mathcal{G}|$ are replaced with their $L^\infty$-covering numbers, respectively.*

**Theorem 2** (Sample complexity for discrete $\Pi$ and $\mathcal{G}$ (informal)). *Let $\|\mathcal{G}_h\|_\infty \le C_{\mathcal{G}}, r_h \in [0,1]$ for any $h \in [H]$ and the PO-bilinear rank assumption holds with PO-biliear rank $d$. By letting $|\Pi_{\max}| = \max_h |\Pi_h|, |\mathcal{G}_{\max}| = \max_h |\mathcal{G}_h|$, with probability $1-\delta$, we can achieve $J(\pi^\star) - J(\hat{\pi}) \le \epsilon$ when we use samples*

$$\tilde{O}\left(d^2 H^4 \max(C_{\mathcal{G}}, 1)^2 |\mathcal{A}|^2 \ln(|\mathcal{G}_{\max}||\Pi_{\max}|/\delta) \ln^2(B_X B_W/\delta)(1/\epsilon)^2\right).$$

*Here, $\operatorname{Polylog}(d, H, |\mathcal{A}|, \ln(|\mathcal{G}|), \ln(|\Pi|), \ln(1/\delta), \ln(B_X), \ln(B_W), \ln(1/\delta), 1/\epsilon)$ are omitted.*

#### 5.3.2 Observable Undercomplete Tabular POMDPs

We start with tabular POMDPs. The details here is deferred to Section K.

**Example 1** (continuing from p. 6). *In tabular models, recall the PO-bilinear rank is at most $d = |\mathcal{O}|^M |\mathcal{A}|^M |\mathcal{S}|$. We suppose $r_h \in [0,1]$ for any $h \in [H]$. Assuming $\mathbb{O}$ is full-column rank, to satisfy the realizability, we set $\mathcal{G}_h = \{\langle \theta, \mathbf{1}(z) \otimes \mathbb{O}^\dagger \mathbf{1}(o)\rangle \mid \|\theta\|_\infty \le H\}$ where $\|\mathbb{O}^\dagger\|_1 \le 1/\sigma_1$ and $\mathbf{1}(z), \mathbf{1}(o)$ are one-hot encoding vectors over $\mathcal{Z}_{h-1}$ and $\mathcal{O}$, respectively. We set $\Pi_h = [\bar{Z}_h \to \Delta(\mathcal{A})]$. Then, the following holds.*

**Theorem 3** (Sample complexity for undercomplete tabular models (Informal)). *With probability $1 - \delta$, we can achieve $J(\pi^\star) - J(\hat{\pi}) \le \epsilon$ when we use samples $\tilde{O}\left(|\mathcal{S}|^2 |\mathcal{A}|^{3M+3} |\mathcal{O}|^{3M+1} H^6 (1/\epsilon)^2 (1/\sigma_1)^2 \ln(1/\delta)\right).$ Here, $\operatorname{polylog}(|\mathcal{S}|, |\mathcal{O}|, |\mathcal{A}|, H, 1/\sigma_1, \ln(1/\delta))$ are omitted.*

*Firstly, while the above error incurs $|\mathcal{O}|^M |\mathcal{A}|^M$, we will later see in Section G.2.2 when we use the more general definition of PO-bilinear AC class and combine a model-based perspective, we might be able to remove $|\mathcal{O}|^M$ from the error bound. The intuition here is that the statistical complexity still scales with $|\mathcal{S}|^2 |\mathcal{A}| + |\mathcal{O}||\mathcal{A}|$ and does not incur $|\mathcal{O}|^M$. At the same time, although PO-bilinear rank currently scales with $|\mathcal{O}|^M ||\mathcal{A}|^M |\mathcal{S}|$, we can show that it can be just $|\mathcal{S}|$ with a more refined definition. Secondly, $\|\mathbb{O}^\dagger\|_1 \le 1/\sigma_1$ can be replaced with other analogous conditions $\|\mathbb{O}^\dagger\|_2 \le 1/\sigma_2$. Here, note $\|\mathbb{O}^\dagger\|_1 = 1/\{\min_x \|\mathbb{O}x\|_1/\|x\|_1\}, \|\mathbb{O}^\dagger\|_2 = 1/\{\min_x \|\mathbb{O}x\|_2/\|x\|_2\}$. The reason why we use 1-norm is to invoke the result [23] to achieve the near global optimality as in the next paragraph.*

**Near global optimality.** *Finally, we consider the PAC guarantee against the globally optimal policy. As shown in [23], it is enough to set $M = O((1/\sigma_1^4) \ln(SH/\epsilon))$ to compete with the globally optimal policy $\pi_{\mathrm{gl}}^\star$. Thus we achieve a quasipolynomial sample complexity when competing against $\pi_{\mathrm{gl}}^\star$.*

**Theorem 4** (Sample complexity for undercomplete tabular models (Informal) — competing against $\pi^\star_{\mathrm{gl}}$)**.** *With probability $1 - \delta$, we can achieve $J(\pi^\star_{\mathrm{gl}}) - J(\hat\pi) \leq \epsilon$ when we use samples at most*

$$\mathrm{poly}(|\mathcal{S}|, |\mathcal{A}|^{\ln(|\mathcal{S}|H/\epsilon)/\sigma_1^4}, |\mathcal{O}|^{\ln(|\mathcal{S}|H/\epsilon)/\sigma_1^4}, H, 1/\sigma_1, 1/\epsilon, \ln(1/\delta)).$$

**Remark 5** (Overcomplete Tabular POMDPs)**.** *We can similarly consider the sample complexity of overcomplete POMDPs. We would incur the additional $|\mathcal{A}|^K$. The detail is in Section L.*

### 5.3.3   Observable LQG

Now let us revisit LQG. The detail here is deferred to Section M. We show that PROVABLE can compete against the globally optimal policy with polynomial sample complexity.

**Example 2** (continuing from p. 6)**.** *In LQG, by setting $H = M - 1$, we achieve a polynomial sample complexity when competing against the globally optimal policy $\pi^\star_{\mathrm{gl}}$.*

**Theorem 5** (Sample complexity for LQG (informal) – competing against $\pi^\star_{\mathrm{gl}}$)**.** *Consider a linear policy class $\Pi_h = \{\delta(a_h = \bar{K}_h \bar{z}_h) \mid \|\bar{K}_h\| \leq \Theta\}$. and assume $\max(\|A\|, \|B\|, \|C\|, \|Q\|, \|R\|) \leq \Theta$ and all policies induce a stable system (we formalize in Section M). With probability $1 - \delta$, we can achieve $J(\pi^\star_{\mathrm{gl}}) - J(\hat\pi) \leq \epsilon$ when we use samples at most*

$$\mathrm{poly}(H, d_s, d_o, d_a, \Theta, \|C^\dagger\|, \ln(1/\delta)) \times (1/\epsilon)^2.$$

### 5.3.4   Observable HSE-POMDPs

Next, we study HSE-POMDPs. The details here is deferred to Section J.

**Example 3** (continuing from p. 7)**.** *In HSE-POMDPs, PO-bilinear rank is at most $\max_h d_{\phi_h}$. Suppose $\|\psi_h\| \leq 1$ and $V_h^\pi(\cdot) = \langle \theta_h^\pi, \phi_h(\cdot) \rangle$ such that $\|\theta_h^\pi\| \leq \Theta_V$ for any $h \in [H]$. Then, to satisfies the realizability, we set $\mathcal{G}_h = \{\langle \theta, \psi_h(\cdot) \rangle \mid \|\theta\| \leq \Theta_V/\sigma_{\min}(K)\}$ where $\sigma_{\min}(K) = \min_{h \in [H]} 1/\|K_h^\dagger\|$.*

**Theorem 6** (Sample complexity for HSE-POMDPs (Informal))**.** *Let $d_\psi = \max_h\{d_{\psi_h}\}, d_\phi = \max_h\{d_{\psi_h}\}, |\Pi_{\max}| = \max_h(|\Pi_h|)$. Suppose $r_h$ lies in $[0, 1]$ for any $h \in [H]$. Then, with probability $1 - \delta$, we can achieve $J(\pi^\star) - J(\hat\pi) \leq \epsilon$ when we use samples*

$$\tilde{O}\left(d_\phi^2 H^4 |\mathcal{A}|^2 \max(\Theta_V, 1)^2 \{d_\psi + \ln(|\Pi_{\max}|/\delta)\}(1/\sigma_{\min}(K))^2 \cdot (1/\epsilon)^2\right).$$

*Here,   $\mathrm{polylog}(d_\phi, d_\psi, |\mathcal{A}|, \Theta_V, \ln(|\Pi_{\max}|), 1/\sigma_{\min}(K), 1/\epsilon, \ln(1/\delta), \sigma_{\max}(T), \sigma_{\max}(K))$   are omitted and $\sigma_{\max}(K) = \max_{h \in [H]} \|K_h\|, \sigma_{\max}(T) = \max_{\pi \in \Pi, h \in [H]} \|T_{\pi:h}\|$.*

Note that the sample complexity above does not explicitly depend on the memory length $M$, instead it only explicitly depends on the dimension of the features $\phi, \psi$. In other words, if we have a feature mapping $\psi_h$ that can map the entire history (i.e., $M = H$) to a low-dimensional vector (e.g., LQG), our algorithm can immediately compete against the global optimality $\pi^\star_{\mathrm{gl}}$.

### 5.4   PSRs, $M$-step decodable POMDPs and Low-rank Observable POMDPs

The result of PSRs is deferred to Section Section E. Besides, our generalized framework can capture two models: (1) $M$-step decodable POMDPs, and (2) observable POMDPs with the latent low-rank transition. The discussion is deferred to Section P. The summary of the results is stated in Section 1.

## 6   Summary

We propose a PO-bilinear actor-critic framework that is the first unified framework for provably efficient RL on large-scale partially observable dynamical systems. Our framework can capture not only many models where provably efficient learning has been known such as tabular POMDPs, LQG and M-step decodable POMDPs, but also models where provably efficient RL is not known such as HSE-POMDPs, PSRs, and low-rank observable POMDPs. Our unified actor-critic based algorithm—PROVABLE provably performs agnostic learning by searching for the best memory-based policy. For special models such as observable tabular POMDPs, LQG, and low-rank POMDPs, by leveraging their special properties, i.e., the exponential stability of Bayesian filters in tabular and low-rank POMDPs, and existence of a compact featurization of histories in LQG, we are able to directly compete against the global optimality without paying an exponential dependence on horizon.

### Acknowledgement

We thank Nan Jiang for the valuable discussions on PSRs. MU and NK acknowledge funding support from NSF IIS-1846210 and Masason Foundation. WS acknowledges funding support from NSF IIS-2154711.

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
