# Contents

# A    Additional Related Works

**Generalization and function approximation of RL in MDPs.**    In Markovian environments, there is a growing literature that gives PAC bounds with function approximation under certain models. Some of the representative models are linear MDPs [36, 76], block MDPs [17, 51, 78], and low-rank MDPs [2, 71]. Several general frameworks in [33, 66, 35, 21, 16] characterize sufficient conditions for provably efficient RL. Each above model is captured in these frameworks as a special case. While our work builds on the bilinear/Bellman rank framework [16, 33], when we naïvely reduce POMDPs to MDPs, the bilinear/Bellman rank is $\Theta(A^H)$. These two frameworks are only shown applicable to reactive POMDPs where the optimal policy only depends on the latest observation. However, this assumption makes the POMDP model very restricted.

**System identification for uncontrolled partially observable systems.**    There is a long line of work on system identification for uncontrolled partially observable systems, among which the spectral learning based methods are related to our work [72, 29, 63, 8, 25, 54, 7, 40, 27, 67]. Informally, these methods leverage the high-level idea that under some observability conditions, one can use the sufficient statistics of (possibly multi-step) future observations as a surrogate for the belief states, thus allowing the learning algorithms to ignore the latent state inference and completely rely on observable quantities. Our approach shares a similar spirit in the sense that we use sufficient statistics of future observations to replace latent states, and our algorithm only relies on observable quantities. The major difference is that these prior works only focus on passive system identification for uncontrolled systems, while we need to find a high-performance policy by actively interacting with the systems for information acquisition.

**Reinforcement learning in PSRs.**    PSRs [32, 45, 62, 8, 69] are models that generalize POMDPs. PSRs also rely on the idea of using the sufficient statistics of multi-step future observations (i.e., predictive states) to serve as a summary of the history. Prior works on RL for PSRs [8, 40, 15, 44, 30] do not address the problem of strategic exploration and operate under the assumption that a pre-collected diverse training dataset is given and the data collection policy is a blind policy (i.e., it does not depend on history of observations). To our knowledge, the only existing PAC learning algorithm for PSRs is limited to a reactive PSR model where the optimal policy depends just on the latest observation [33]. Our framework captures standard PSRs models that are strictly more general than reactive PSRs.

**Future-dependent value functions.**    Analogue of future-dependent value functions (referred to as bridge functions) are used in the literature of causal inference (offline contextual bandits) [50, 13, 12, 37, 49, 61, 75] and offline RL with unmeasured confounders [4, 59]. However, their settings are not standard POMDPs in the sense that their setting is a POMDP with unmeasured confounders following [68]. Our setting is a standard POMDP without unmeasured confounders. Here, we emphasize that their setting does *not* capture our setting. More specifically, by taking [59] as an example, they require that logged data is generated by policies that can depend on latent states but cannot depend on observable states. Thus, their definition of future-dependent value functions (called as bridge functions) is not applicable to our setting since the data we use is clearly generated by policies that depend on observations. Due to this difference, their setting prohibits us from using future observations, unlike our setting. Finally, we stress that our work is online, while their setting is offline. Hence, they do not discuss any methods for exploration.

# B Supplement for Section 3

We generalize Definition 3 to capture more models. The first extension is to use multi-step future-dependent value functions. This extension is essential to capture overcomplete POMDPs and multi-step PSRs.

## B.1 PO-bilinear Actor-critic Class with Multi Step Future

In this section, we provide an extension to Definition 3 to incorporate multiple-step futures (i.e., $K > 1$). For simplicity, we assume that $\pi^{out} = \mathcal{U}(\mathcal{A})$.

The definition is then as follows. The main difference is that we roll out a policy $\mathcal{U}(\mathcal{A})$, $K - 1$ times to incorporate multi-step future-dependent value functions. We introduce the notation

$$(z_{h-1}, o_{h:h+K-1}, a_{h:h+K-2}) = \bar{z}_h^K \in \bar{\mathcal{Z}}_h^K = \mathcal{Z}_{h-1} \times \mathcal{O}^K \times \mathcal{A}^{K-1}.$$

Then, combining the Bellman equation for state-value functions and the definition of K-step future-dependent value functions, we have

$$
\begin{aligned}
0 &= \mathbb{E}[V_{h+1}^\pi(z_h, s_{h+1}) + r_h - V_h^\pi(z_{h-1}, s_h) \mid z_{h-1}, s_h; a_h \sim \pi] \\
&= \mathbb{E}[g_{h+1}^\pi(\bar{z}_{h+1}^K) \mid z_{h-1}, s_h; a_h \sim \pi, a_{h+1:h+K-1} \sim \mathcal{U}(\mathcal{A})] + \mathbb{E}[r_h \mid z_{h-1}, s_h; a_h \sim \pi] \\
&\quad - \mathbb{E}[g_h^\pi(\bar{z}_h^K) \mid z_{h-1}, s_h; a_{h:h+K-2} \sim \mathcal{U}(\mathcal{A})]
\end{aligned}
$$

Thus, by taking expectations further with respect to $(z_{h-1}, s_h)$ (i.e., $z_{h-1}, s_h$ can be sampled from some roll-in policy), we have

$$
\begin{aligned}
0 &= \mathbb{E}[g_{h+1}^\pi(\bar{z}_{h+1}^K); a_{1:h-1} \sim \pi', a_h \sim \pi, a_{h+1:h+K-1} \sim \mathcal{U}(\mathcal{A})] + \mathbb{E}[r_h; a_{1:h-1} \sim \pi', a_h \sim \pi] \\
&\quad - \mathbb{E}[g_h^\pi(\bar{z}_h^K); a_{1:h-1} \sim \pi', a_{h:h+K-2} \sim \mathcal{U}(\mathcal{A})].
\end{aligned}
$$

Hence, the Bellman loss of a pair $(\pi, g)$ under a roll-in $\pi'$ denoted by $\mathrm{Br}_h(\pi, g; \pi')$ at $h \in [H]$ is defined as

$$
\begin{aligned}
\mathrm{Br}_h(\pi, g; \pi') &= \mathbb{E}[g_{h+1}(\bar{z}_{h+1}^K); a_{1:h-1} \sim \pi', a_h \sim \pi, a_{h+1:h+K-1} \sim \mathcal{U}(\mathcal{A})] + \mathbb{E}[r_h; a_{1:h-1} \sim \pi', a_h \sim \pi] \\
&\quad - \mathbb{E}[g_h(\bar{z}_h^K); a_{1:h-1} \sim \pi', a_{h:h+K-2} \sim \mathcal{U}(\mathcal{A})].
\end{aligned}
$$

The above is a proper loss function when we use multi-step futures. Here is the structure we need for $\mathrm{Br}_h(\pi, g; \pi')$.

**Definition 4** (PO-bilinear AC Class for POMDPs with multi-step future). *The model is a PO-bilinear class of rank $d$ if $\mathcal{G}$ is realizable (regarding general K-step future-dependent value functions), and there exists $W_h : \Pi \times \mathcal{G} \to \mathbb{R}^d$ and $X_h : \Pi \to \mathbb{R}^d$ such that for all $\pi', \pi \in \Pi, g \in \mathcal{G}$ and $h \in [H]$,*

    *1. We have:*

$$
\begin{aligned}
&\mathbb{E}[g_{h+1}(\bar{z}_{h+1}^K); a_{1:h-1} \sim \pi', a_h \sim \pi, a_{h+1:h+K-1} \sim \mathcal{U}(\mathcal{A})] + \mathbb{E}[r_h; a_{1:h-1} \sim \pi', a_h \sim \pi] \\
&\quad - \mathbb{E}[g_h(\bar{z}_h^K); a_{1:h-1} \sim \pi', a_{h:h+K-2} \sim \mathcal{U}(\mathcal{A})] = \langle W_h(\pi, g), X_h(\pi') \rangle,
\end{aligned}
$$

    *2. $W_h(\pi, g^\pi) = 0$ for any $\pi \in \Pi$ and the corresponding future-dependent value function $g^\pi$ in $\mathcal{G}$.*

*We define $d$ as the PO-bilinear rank.*

# C Supplement for Section 4

## C.1 Observable Undercomplete Tabular POMDPs

We need to prove Lemma 2. In the tabular case, by setting

$$\psi_h(z, o) = \mathbf{1}(z) \otimes \mathbf{1}(o), \phi_h(z, s) = \mathbf{1}(z) \otimes \mathbf{1}(s), K_h = I_{|\mathcal{Z}_{h-1}|} \otimes \mathbb{O}$$

where $\mathbf{1}(z), \mathbf{1}(o), \mathbf{1}(s)$ are one-hot encoding vectors over $\mathcal{Z}_{h-1}, \mathcal{O}, \mathcal{S}$, respectively. Then, we can regard the tabular model as an HSE-POMDP. We can just invoke Lemma 4.

## C.2 Observable Overcomplete Tabular POMDPs

We consider overcomplete POMDPs with multi-step futures. The proofs are deferred to Section C.2. We have the following theorem. This is a generalization of Lemma 1, i.e., when $K = 1$, it is Lemma 1.

**Lemma 5.** *Define a $|\mathcal{T}^K| \times |\mathcal{S}|$-dimensional matrix $\mathbb{O}^K$ whose entry indexed by $(o_{h:h+K-1}, a_{h:h+K-2}) \in \mathcal{T}^K$ and $s_h \in \mathcal{S}$ is equal to $\mathbb{P}(o_{h:h+K-1}, a_{h:h+K-2} \mid s_h; a_{h:h+K-2} \sim \mathcal{U}(\mathcal{A}))$. When this matrix is full-column rank, K-step future-dependent value functions with respect to $\mathcal{U}(\mathcal{A})$ exist.*

Note a sufficient condition to satisfy the above is that a matrix $\mathbb{O}^K(a'_{h:h+K-2}) \in \mathbb{R}^{|\mathcal{O}|^K \times |\mathcal{S}|}$ whose entry indexed by $o_{h:h+K-1} \in \mathcal{O}^K$ and $s_h \in \mathcal{S}$ is equal to $\mathbb{P}(o_{h:h+K-1} \mid s_h; a_{h:h+K-2} = a'_{h:h+K-2})$ is full-column rank for certain $a'_{h:h+K-2} \in \mathcal{A}^{K-1}$. It says there is (unknown) action sequence with length $K$ that retains information about latent states.

We next calculate the PO-bilinear rank. Importantly, this does *not* depend on $|\mathcal{A}|^K$ and $|\mathcal{O}|^K$.

**Lemma 6.** *Set a future-dependent value function class $\mathcal{G}_h = [\bar{\mathcal{Z}}^K \to [0, C_{\mathcal{G}}]]$ for certain $C_{\mathcal{G}} \in \mathbb{R}^+$ and a policy class $\Pi_h = [\bar{\mathcal{Z}}_h \to \Delta(\mathcal{A})]$. Then, the model satisfies PO-bilinear rank condition with PO-bilinear rank (Definition 4) at most $|\mathcal{S}|(|\mathcal{O}||\mathcal{A}|)^M$.*

Note that the bilinear rank is still $|\mathcal{S}|(|\mathcal{O}||\mathcal{A}|)^M$ (just $|\mathcal{S}|$ in the more general definition in Section F). Crucially, it does not depend on the length of futures $K$.

**Proof of Lemma 5** Consider any function $f : \mathcal{Z}_{h-1} \times \mathcal{S} \to \mathbb{R}$ (thus, this captures all possible $V_h^\pi$). Denote $\mathbf{1}(z)$ as the one-hot encoding of $Z_{h-1}$ (similarly for $\mathbf{1}(s)$ over $\mathcal{S}$ and $\mathbf{1}(t)$ over $\mathcal{T}^K$). We have $f(z, s) = \langle f, \mathbf{1}(z) \otimes \mathbf{1}(s) \rangle = \langle f, \mathbf{1}(z) \otimes ((\mathbb{O}^K)^\dagger \mathbb{O}^K \mathbf{1}(s)) \rangle$, where we use the assumption that $\mathrm{rank}(\mathbb{O}^K) = |\mathcal{S}|$ and thus $(\mathbb{O}^K)^\dagger \mathbb{O}^K = I$. Then,

$$f(z_{h-1}, s_h) = \langle f, \mathbf{1}(z_{h-1}) \otimes (\mathbb{O}^K)^\dagger \mathbb{E}[\mathbf{1}(o_{h:h+K-1}, a_{h:h+K-2}) \mid s_h; a_{h:h+K-2} \sim \pi^{out}] \rangle$$
$$= \mathbb{E}[\langle f, \mathbf{1}(z_{h-1}) \otimes (\mathbb{O}^K)^\dagger \mathbf{1}(o_{h:h+K-1}, a_{h:h+K-2}) \rangle \mid z_{h-1}, s_h; a_{h:h+K-2} \sim \pi^{out}].$$

which means that the value bridge function corresponding to $f(\cdot)$ is

$$g(z, t) := \langle f, \mathbf{1}(z) \otimes (\mathbb{O}^K)^\dagger \mathbf{1}(t) \rangle.$$

∎

**Proof of Lemma 6** Recall we want to show the low-rank property of the following loss function:

$$\mathbb{E}[g_{h+1}(\bar{z}_{h+1}^K); a_{1:h-1} \sim \pi', a_h \sim \pi, a_{h+1:h+K-1} \sim \pi^{out}] + \mathbb{E}[r_h; a_{1:h-1} \sim \pi', a_h \sim \pi]$$
$$- \mathbb{E}[g_h(\bar{z}_h^K); a_{1:h-1} \sim \pi', a_{h:h+K-1} \sim \pi^{out}].$$

We consider an expectation conditioning on $z_{h-1}$ and $s_h$. For some vector $\theta_{\pi,g} \in \mathbb{R}^{|\mathcal{Z}_{h-1}| \times |\mathcal{S}|}$, which depends on $\pi$, we write it in the form of $\langle \theta_{\pi,g}, \mathbf{1}(z_{h-1}, s_h) \rangle$ where $\mathbf{1}(z_{h-1}, s_h)$ is the one-hot encoding vector over $\mathcal{Z}_{h-1} \times \mathcal{S}$. Then, the loss for $(\pi, g)$ is equal to

$$\langle \theta_{\pi,g}, \mathbb{E}[\mathbf{1}(z_{h-1}, s_h); a_{1:h-1} \sim \pi'] \rangle.$$

Hence, we can take $X(\pi') = \mathbb{E}[\mathbf{1}(z_{h-1}, s_h); a_{1:h-1} \sim \pi']$ and $W(\pi) = \theta_{\pi,g}$. ∎

## C.3 Observable Linear Quadratic Gaussian

We need to prove Lemma 3. The proof is further deferred to Section M.

## C.4 Observable HSE-POMDPs

We first provide the proof of Lemma 4. Then, we briefly mention how we extend to the infinite-dimensional setting.

**Proof of the first statement in Lemma 4**   First, we need to show value bridge functions exist. This is proved noting

$$\mathbb{E}_{o\sim\mathbb{O}(s)}[\langle (K_h^\dagger)^\top \theta_h^\pi, \psi_h(\bar{z}_h)\rangle] = \langle (K_h^\dagger)^\top \theta_h^\pi, K_h\phi_h(z_{h-1}, s_h)\rangle = \langle \theta_h^\pi, \phi_h(z_{h-1}, s_h)\rangle = V_h^\pi(z_{h-1}, s_h).$$

Thus, $\langle (K_h^\dagger)^\top \theta_h^\pi, \psi_h(\bar{z}_h)\rangle$ is a value bridge function.   ∎

**Proof of the second statement in Lemma 4**   Consider a triple $(\pi', \pi, g) \in \Pi \times \Pi \times \mathcal{G}$, with $g_h(\cdot) = \theta_h^\top \psi_h(\cdot)$ and $g_h^\pi = \langle \theta_h^\star, \psi_h(\cdot)\rangle$, we have:

$$\begin{aligned}
&\mathrm{Br}_h(\pi, g; \pi') \\
&= \mathbb{E}\left[\theta_h^\top \psi(\bar{z}_h) - r_h - \theta_{h+1}^\top \psi(\bar{z}_{h+1}); a_{1:h-1} \sim \pi', a_h \sim \pi\right] \\
&= \mathbb{E}\left[\theta_h^\top K_h \phi_h(z_{h-1}, s_h) - r_h - \theta_{h+1}^\top K_{h+1}(\phi_{h+1}(z_h, s_{h+1})); a_{1:h-1} \sim \pi', a_h \sim \pi\right] \\
&= \mathbb{E}\left[(\theta_h - \theta_h^\star)^\top K_h \phi_h(z_{h-1}, s_h) - (\theta_{h+1} - \theta_{h+1}^\star)^\top K_{h+1}(T_{\pi;h}\phi_h(z_{h-1}, s_h)); a_{1:h-1} \sim \pi'\right] \\
&= \left\langle \mathbb{E}[\phi_h(z_{h-1}, s_h); a_{1:h-1} \sim \pi'], \quad K_h^\top(\theta_h - \theta_h^\star) - T_{\pi;h}^\top K_{h+1}^\top(\theta_{h+1} - \theta_{h+1}^\star)\right\rangle,
\end{aligned}$$

which verifies the bilinear structure, i.e., $X_h(\pi') = \mathbb{E}[\phi_h(z_{h-1}, s_h); a_{1:h-1} \sim \pi']$, and $W_h(\pi, g) = K_h^\top(\theta_h - \theta_h^\star) - T_{\pi;h}^\top K_{h+1}^\top(\theta_{h+1} - \theta_{h+1}^\star)$, and shows that the bilinear rank is at most $\max_h d_{\psi_h}$.   ∎

**Infinite dimensional HSE-POMDPs**   Consider the case $\phi_h$ and $\psi_h$ are features in infinite dimensional RKHS. By assuming that the spectrum of the operator $K_h$ is decaying with a certain order, we can still ensure the existence of value bridge functions even if $d_{\phi_h}$ and $d'_{\psi_h}$ are infinite dimensional.

Next, we consider the PO-bilinear rank. We can still use the decomposition in the proof above. While the PO-bilinear rank itself in the current definition is infinite-dimensional, when we get the PAC result later, the dependence on the PO-bilinear rank comes from the information gain based on $X_h(\pi)$, which is the intrinsic dimension of $X_h(\pi)$. Thus, we can easily get the sample complexity result by replacing $d_{\psi_h}$ with the information gain over $\psi_h(\cdot)$ [65]. Generally, to take infinite dimensional models into account, the PO-bilinear rank in Definition 3 can be generalized using the critical information gain [16].

## D   Supplement for Section 5

In this section, we first consider the case with multi-step futures. Next, we present a modification to handle LQG with continuous action in Definition 3.

### D.1   Algorithm with Multi-Step Future-Dependent Value Functions

Finally, we consider the case with multi-step futures in Algorithm 2 when $\pi^{out} = \mathcal{U}(\mathcal{A})$. Recall the notation $\bar{z}_h^K = (z_{h-1}, o_{h:h+K-1}, a_{h:h+K-2})$. The only difference is in the process of data collection. Particularly, at every iteration $t$, we roll-in using $\pi^t$ to (and include) time step $h - 1$, we then roll-out by switching to $\mathcal{U}(\mathcal{A})$ for $K$ steps.

### D.2   Algorithm for LQG with Continuous Action

Our algorithm so far samples $a_h$ from $\mathcal{U}(\mathcal{A})$ and performs importance weighting in designing the loss $\sigma_h^t$, which will incur a polynomial dependence on $|\mathcal{A}|$ as we will see in the next section. However, among the examples that we consider in Section 4, LQG has continuous action. If we naïvely sample $a_h$ from a ball in $\mathbb{R}^{d_a}$ and perform (nonparametric) importance weighting, we will pay $\exp(d_a)$ in our sample complexity bound, which is not ideal for high-dimension control problems. To avoid exponential dependence on $d_a$, here we replace $\mathcal{U}(\mathcal{A})$ with a $d$-optimal design over the action's quadratic feature space.

Here, we want to evaluate the Bellman error of $(\pi, g)$ pair under a roll-in policy $\pi'$:

$$\mathrm{Br}_h(\pi, g; \pi') := \mathbb{E}[u_h(\bar{z}_h, a_h, r_h, o_{h+1}; \theta); a_{1:h-1} \sim \pi', a_h \sim \pi(\bar{z}_h)]$$

where $u_h(\bar{z}_h, a_h, r_h, o_{h+1}; \theta) = \theta_h^\top \psi(\bar{z}_h) - r_h(s_h, a_h) - \theta_{h+1}^\top \psi_{h+1}(\bar{z}_{h+1})$ for any linear deterministic policy $\pi \in \Pi$ (here $g_h(\cdot) := \theta_h^\top \psi(\cdot)$) using a *single* policy. In other words, we would like to get

---

**Algorithm 2** PaRtially ObserVAble BiLinEar (PROVABLE) # multi-step version

---

1: **Input:** Future-dependent value function class $\mathcal{G} = \{\mathcal{G}_h\}, \mathcal{G}_h \subset [\bar{Z}_h^K \to \mathbb{R}]$, Policy class $\Pi = \{\Pi_h\}, \Pi_h \subset [\bar{\mathcal{Z}}_h \to \Delta(\mathcal{A})]$, parameters $m \in \mathbb{N}, R \in \mathbb{R}$

2: Define

$$l_h(\bar{z}_h^K, a_{h+K-1}, r_h, o_{h+K}; \pi, g) := |\mathcal{A}|\pi_h(a_h \mid \bar{z}_h) \left(g_{h+1}(\bar{z}_{h+1}^K) + r_h\right) - g_h(\bar{z}_h^K).$$

3: Initialize $\pi^0 \in \Pi$

4: Form the first step dataset $\mathcal{D}^0 = \{\bar{z}_1^{K;i}\}_{i=1}^m$ where each $\bar{z}^K$ is generated by following $a_{1:K-1} \sim U(\mathcal{A})$ in an i.i.d manner.

5: **for** $t = 0 \to T - 1$ **do**

6:     for any $h \in [H]$, define the Bellman error

$$\sigma_h^t(\pi, g) = \mathbb{E}_{\mathcal{D}_h^t}[l_h(\bar{z}_h^K, a_{h+K-1}, r_h, o_{h+K}; \pi, g)]$$

   where $\mathcal{D}_h^t$ means empirical approximation by executing $a_{1:h-1} \sim \pi^t, a_{h:h+K-1} \sim \mathcal{U}(\mathcal{A})$ and collecting $m$ i.i.d tuples.

7:     Select policy optimistically as follows (here note $g = \{g_h\}_{h=1}^H$)

$$(\pi^{t+1}, g^{t+1}) := \underset{\pi \in \Pi, g \in \mathcal{G}}{\arg\max} \mathbb{E}_{\mathcal{D}^0}[g_1(\bar{z}_1^K)] \quad \text{s.t.} \quad \forall h \in [H], \forall i \in [t], \sigma_h^i(\pi, g)^2 \le R.$$

8: **end for**

9: **Output:** Randomly choose $\hat{\pi}$ from $(\pi_1, \cdots, \pi_T)$.

---

a good loss $l_h$ such that

$$\text{Br}_h(\pi, g; \pi') = \mathbb{E}[l_h(\bar{z}_h, a_h, r_h, o_{h+1}; \theta, \pi); a_{1:h-1} \sim \pi', a_h \sim \pi^e]$$

for some policy $\pi^e$ without incuring exponential dependence on $d_a$. We explain how to design such a loss function $l_h(\cdot; \pi, g)$ step by step.

**First Step** The first step is to consider the conditional expectation on $(\bar{z}_h, s_h, a_h)$. Here, using the quadratic form of $\psi$, we can show that there are some $c_0 : \bar{Z}_h \times \mathcal{S} \to \mathbb{R}, c_1 : \bar{Z}_h \times \mathcal{S} \to \mathbb{R}^{(d_a+d_s+d_{\bar{z}_h})^2}, c_2 \in \mathbb{R}$:

$$\begin{aligned}
\text{Br}_h(\pi, g; \pi') &= \mathbb{E}[u_h(\bar{z}_h, a_h, r_h, o_{h+1}; \theta) \mid \bar{z}_h, s_h, a_h; a_{1:h-1} \sim \pi, a_h \sim \pi(\bar{z}_h)] \\
&= \langle c_2(\theta), [1, [\bar{z}_h^\top, s_h^\top, a_h^\top] \otimes [\bar{z}_h^\top, s_h^\top, a_h^\top]]^\top \rangle \\
&= c_0(\bar{z}_h, s_h; \theta) + c_1^\top(\bar{z}_h, s_h; \theta)\kappa(a_h)
\end{aligned}$$

where $\kappa(a) = [a^\top, (a \otimes a)^\top]^\top$. Then, the Bellman loss we want to evaluate can be written in the form of

$$\begin{aligned}
&\mathbb{E}[u_h(\bar{z}_h, a_h, r_h, o_{h+1}; \theta); a_{1:h-1} \sim \pi', a_h \sim \pi(\bar{z}_h)] \\
&= \mathbb{E}[c_0(\bar{z}_h, s_h; \theta) + c_1^\top(\bar{z}_h, s_h; \theta)\kappa(\pi(\bar{z}_h)); a_{1:h-1} \sim \pi'].
\end{aligned}$$

**Second step** The second step is to compute a d-optimal design for the set $\{\kappa(a) : a \in \mathbb{R}^{d_a}, \|a\|_2 \le Z\}$ for certain enough large $Z \in \mathbb{R}$, and denote $a^1, \ldots, a^{d^\diamond}$ as the supports on the d-optimal design. Note in LQG, though we cannot ensure the action lives in the compact set, we can still ensure that in high probability and it suffices in our setting as we will see. Since the dimension of $k(a)$ is $d_a + d_a^2$, we can ensure $d^\diamond \le (d_a + d_a^2)(d_a + d_a^2 + 1)/2$ [43, 39]. Here is a concrete theorem we invoke.

**Theorem 7** (Property of G-optimal design). *Suppose $\mathcal{X} \in \mathbb{R}^d$ is a compact set. There exists a distribution $\rho$ over $\mathcal{X}$ such that:*

- *$\rho$ is supported on at most $d(d+1)/2$ points.*

- *For any $x' \in \mathcal{X}$, we have $x'^\top \mathbb{E}_{x \sim \rho}[xx^\top]^{-1} x' \le d$.*

We have the following handy lemma stating any $\kappa(a)$ is spanned by $\{\kappa(a^i)\}_{i=1}^{d^\diamond}$.

**Lemma 7.** *Let* $K = [\rho^{1/2}(a^1)\kappa(a^1), \rho^{1/2}(a^2)\kappa(a^2), \cdots, \rho^{1/2}(a^{d^\diamond})\kappa(a^{d^\diamond})]$ *and* $\alpha(a) = K^\top(KK^\top)^{-1}k(a)$. *Then, it satisfies*

$$\kappa(a) = K\alpha(a), \quad \|\alpha(a)\| \le (d_a + d_a^2)^{1/2}, \quad \alpha_i(a)/\rho^{1/2}(a^i) \le (d_a + d_a^2)$$

*Proof.* Since $K$ is full-raw rank from the construction of G-optimal design, $KK^\top$ is invertible. Then, we have

$$\sum_{i=1}^{d^\diamond} \alpha_i(a)\rho^{1/2}(a^i)\kappa(a^i) = KK^\top(KK^\top)^{-1}\kappa(a) = \kappa(a)$$

For the latter statement, we have

$$\langle K^\top(KK^\top)^{-1}k(a), K^\top(KK^\top)^{-1}k(a)\rangle = k(a)^\top(KK^\top)^{-1}k(a) \le (d_a + d_a^2).$$

We use a property of G-optimal design in Theorem 7.

For the last statement, we have

$$\kappa^\top(a^i)(KK^\top)^{-1}\kappa(a) \le \|\kappa^\top(a^i)\|_{(KK^\top)^{-1}}\|\kappa^\top(a)\|_{(KK^\top)^{-1}} \le (d_a + d_a^2).$$

from CS inequality. $\qquad\square$

**Third Step** The third step is combining current facts. Recall we want to evaluate

$$\mathbb{E}[u_h(\bar{z}_h, a_h, r_h, o_{h+1}; \theta); a_{1:h-1} \sim \pi', a_h \sim \pi(\bar{z}_j)] = \mathbb{E}[c_0(\bar{z}_h, s_h; \theta) + c_1^\top(\bar{z}_h, s_h; \theta)\kappa(\pi(\bar{z}_h)); a_{1:h-1} \sim \pi'].$$

In addition, the following also holds:

$$\mathbb{E}[u_h(\bar{z}_h, a_h, r_h, o_{h+1}; \theta); a_{1:h-1} \sim \pi', a_h \sim do(a^i)] = \mathbb{E}[c_0(\bar{z}_h, s_h; \theta) + c_1^\top(\bar{z}_h, s_h; \theta)\kappa(a^i); a_{1:h-1} \sim \pi']$$

$$\mathbb{E}[u_h(\bar{z}_h, a_h, r_h, o_{h+1}; \theta); a_{1:h-1} \sim \pi', a_h \sim do(0)] = \mathbb{E}[c_0(\bar{z}_h, s_h; \theta); a_{1:h-1} \sim \pi']$$

Here, we use $\kappa(0) = 0$. This concludes that

$$\mathbb{E}[u_h(\bar{z}_h, a_h, r_h, o_{h+1}; \theta); a_{1:h-1} \sim \pi', a_h \sim \pi(\bar{z}_j)]$$

$$= \mathbb{E}[c_0(\bar{z}_h, s_h; \theta) + c_1^\top(\bar{z}_h, s_h; \theta)\kappa(\pi(\bar{z}_h)); a_{1:h-1} \sim \pi']$$

$$= \mathbb{E}[c_0(\bar{z}_h, s_h; \theta) + c_1^\top(\bar{z}_h, s_h; \theta)\{\sum_{i=1}^{d^\diamond} \alpha_i(\pi(\bar{z}_h))\kappa(a^i)\}; a_{1:h-1} \sim \pi']$$

$$= \mathbb{E}\left[c_0(\bar{z}_h, s_h; \theta)\left(1 - \sum_{i=1}^{d^\diamond} \alpha_i(\pi(\bar{z}_h))\right) + \sum_{i=1}^{d^\diamond} \alpha_i(\pi(\bar{z}_h))\left(c_1^\top(\bar{z}_h, s_h; \theta)\kappa(a^i) + c_0(\bar{z}_h, s_h; \theta)\right); a_{1:h-1} \sim \pi'\right]$$

$$= \mathbb{E}\left[\left(1 - \sum_{i=1}^{d^\diamond} \alpha_i(\pi(\bar{z}_h))\right)u_h(\bar{z}_h, a_h, r_h, o_{h+1}; \theta); a_{1:h-1} \sim \pi', a_h \sim do(0)\right]$$

$$+ \sum_{i=1}^{d^\diamond} \mathbb{E}\left[\alpha_i(\pi(\bar{z}_h))u_h(\bar{z}_h, a_h, r_h, o_{h+1}; \theta); a_{1:h-1} \sim \pi', a_h \sim do(a^i)\right].$$

Thus, we can perform policy evaluation for a policy $\pi$ if we can do intervention from $do(0), do(a^1), \cdots, do(a^{d^\diamond})$.

**Fourth Step** The fourth step is replacing $do(0), do(a^1), \cdots, do(a^{d^\diamond})$ with a single policy that uniformly randomly select actions from the set $\{0, a^1, \ldots, a^{d^\diamond}\}$, which we denote as $a \sim U(1 + d^\diamond)$. Using importance weighting, we define the loss function for $\pi, \theta$ as follows:

$$\mathbb{E}[f_h(\bar{z}_h, a_h, r_h, o_{h+1}; \theta, \pi); a_{1:h-1} \sim \pi', a_h \sim U(1 + d^\diamond)] \qquad (3)$$

where $U(1 + d^\diamond)$ is a uniform action over $0, a^1, \cdots, a^{d^\diamond}$ and

$$f_h(\bar{z}_h, a_h, r_h, o_{h+1}; \theta, \pi)$$

$$= |1 + d^\diamond|\left(\mathbb{I}(a_h = 0)\left(1 - \sum_{i=1}^{d^\diamond} \alpha_i(\pi(\bar{z}_h))\right) + \sum_{i=1}^{d^\diamond} \mathbb{I}(a_h = a^i)\alpha_i(\pi(\bar{z}_h))\right)u_h(\bar{z}_h, a_h, r_h, o_{h+1}; \theta).$$

The term 3is equal to $\text{Br}_h(\pi, g; \pi')$ we want to evaluate.

**Summary** To summarize, we just need to use the following loss function in line 6 in Algorithm 1:

$$\mathbb{E}_{\mathcal{D}_h^t}[l_h(\bar{z}_h, a_h, r_h, o_{h+1}; \theta, \pi)]$$

where $l_h(\bar{z}_h, a_h, r_h, o_{h+1}; \theta, \pi)$ is

$$\mathbb{I}(\|\bar{z}_h\| \leq Z_1)\mathbb{I}(\|r_h\| \leq Z_2)\mathbb{I}(\|o_{h+1}\| \leq Z_3)f_h(\bar{z}_h, a_h, r_h, o_{h+1}; \theta, \pi)$$

and $\mathcal{D}_h^t$ is a set of $m$ i.i.d samples following the distribution induced by executing $a_{1:h-1} \sim \pi'$, $a_h \sim U(1 + d^\diamond)$. Values $Z_1, Z_2, Z_3$ in indicators functions are some large values selected properly later. Due to unbounded Gaussian noises in LQG, indicators functions for truncation is introduced here for technical reason to get valid concentration in Assumption 2.

# E Predictive State Representations

We first give a summary of our results in PSRs. Then, we first add several discussions to explain core tests in detail. Next, we show the existence and form of future-dependent value functions. Finally, we calculate the PO-bilinear rank. In this section, we will focus on the general case where tests could be multiple steps.

## E.1 Summary

In this section, we demonstrate that our definition and algorithm applies to PSRs — models that strictly generalize POMDPs [45, 62]. Below, we first briefly introduce PSRs, followed by showing that it is a PO-bilinear AC model. Throughout this section, we will focus on discrete linear PSRs. We also suppose reward at $h$ is deterministic function of $(o_h, a_h)$ conditional on $\tau_{h+1}^a$ where $\tau_h^a = (o_1, a_1, \cdots, o_{h-1}, a_{h-1})$. Given $\tau_h^a$, the dynamical system generates $o_h \sim \mathbb{P}(\cdot|\tau_h^a)$. Here we use the superscript $a$ on $\tau_h^a$ to emphasize that the $\tau_h^a$ ends with the action $a_{h-1}$.

PSRs use the concept of *test*, which is a sequence of future observations and actions, i.e., for some test $t = (o_{h:h+W-1}, a_{h:h+W-2})$ with length $W \in \mathbb{N}^+$, we define the probability of test $t$ being successful $\mathbb{P}(t|\tau_h^a)$ as $\mathbb{P}(t|\tau_h^a) := \mathbb{P}(o_{h:h+W-1}|\tau_h^a; \mathrm{do}(a_{h:h+W-2}))$ which is the probability of observing $o_{h:h+W-1}$ by actively executing actions $a_{h:h+W-2}$ conditioned on history $\tau_h^a$.

We now explain one-step observable PSRs while deferring the general multi-step observable setting to Section E. A one-step observable PSR uses the observations in $\mathcal{O}$ as tests, i.e., tests with length 1.

**Definition 5** (Core test set and linear PSRs). *A core test set $\mathcal{T} \subset \mathcal{O}$ contains a finite number of tests (i.e., observations from $\mathcal{O}$). For any $h$, any history $\tau_h^a$, any future test $t_h = (o_{h:h+W-1}, a_{h:h+W-2})$ for any $W \in \mathbb{N}^+$, there exists a vector $m_{t_h} \in \mathbb{R}^{|\mathcal{T}|}$, such that the probability of $t_h$ succeeds conditioned on $\tau_h^a$ can be expressed as: $\mathbb{P}(t_h|\tau_h^a) = m_{t_h}^\top[\mathbb{P}(o|\tau_h^a)]_{o \in \mathcal{T}}$, where we denote $\mathbf{q}_{\tau_h^a} := [\mathbb{P}(o|\tau_h^a)]_{o \in \mathcal{T}}$ as a vector in $\mathbb{R}^{|\mathcal{T}|}$ with entries equal to $\mathbb{P}(o|\tau_h^a)$ for $o \in \mathcal{T}$. The vector $\mathbf{q}_{\tau_h^a}$ is called predictive state.*

A core test set $\mathcal{T}$ that has the smallest number of tests is called a *minimum core test set* denoted as $\mathcal{Q}$. PSRs are strictly more expressive than POMDPs in that all POMDPs can be embedded into PSRs whose size of the minimum core tests is at most $|\mathcal{S}|$; however, vice versa does not hold [45]. For example, in observable undercomplete POMDPs (i.e., $\mathbb{O}$ full column rank), the observation set $\mathcal{O}$ can serve as a core test set, but the minimum core test set $\mathcal{Q}$ will have size $|\mathcal{S}|$. Here, we assume we know a core test set $\mathcal{T}$ that contains $\mathcal{Q}$; however, we are agnostic to which set is the actual $\mathcal{Q}$. In the literature on PSRs, this setting is often referred to as transform PSRs [8, 57].

Now we define a future-dependent value function in PSRs. First, given an $M$-memory policy, define $\mathcal{V}_h^\pi(\tau_h^a) = \mathbb{E}[\sum_{t=h}^H r_t | \tau_h^a; a_{h:H} \sim \pi]$, i.e., the expected total reward under $\pi$, conditioned on the history $\tau_h^a$. Note that our value function here depends on the entire history.

**Definition 6** (General future-dependent value functions). *Consider an $M$-memory policy $\pi$. One-step general future-dependent value functions $g_h^\pi : \mathcal{Z}_{h-1} \times \mathcal{T} \to \mathbb{R}$ at step $h \in [H]$ are defined as solutions to*

$$\mathcal{V}_h^\pi(\tau_h^a) = \mathbb{E}[g_h^\pi(z_{h-1}, o_h) \mid \tau_h^a]. \tag{4}$$

This definition is more general than Definition 3 since (4) implies (1) in POMDPs by setting $\mathcal{O} = \mathcal{T}$. In PSRs, we can show the existence of this general future-dependent value function.

**Lemma 8** (The existence of future-dependent value functions for PSRs). *Suppose $\mathcal{T}$ is a core test set. Then, a one-step future-dependent value function $g_h^\pi$ always exists.*

The high-level derivation is as follows. Using the linear PSR property, one can first show that $\mathcal{V}_h^\pi(\tau_h^a)$ has a bilinear form $\mathcal{V}_h^\pi(\tau_h^a) = \mathbf{1}(z_{h-1})^\top \mathbb{J}_h^\pi \mathbf{q}_{\tau_h^a}$, where $\mathbf{1}(z) \in \mathbb{R}^{|\mathcal{Z}_{h-1}|}$ denotes the one-hot encoding vector over $\mathcal{Z}_{h-1}$, and $\mathbb{J}_h^\pi$ is a $|\mathcal{Z}_{h-1}| \times |\mathcal{T}|$ matrix. Then, given any $\tau_h^a$ and $o \sim \mathbb{P}(\cdot|\tau_h^a)$, for some $|\mathcal{Z}_{h-1}| \times |\mathcal{T}|$ matrix $\mathbb{J}$, we can show $g_h(z_{h-1}, o) := \mathbf{1}(z_{h-1})^\top \mathbb{J}[\mathbf{1}(t = o)]_{t \in \mathcal{T}}$ satisfies the above, where $[\mathbf{1}(t = o)]_{t \in \mathcal{T}} \in \mathbb{R}^{|\mathcal{T}|}$ is a one-hot encoding vector over $\mathcal{T}$ and serves as an unbiased estimate of $\mathbf{q}_{\tau_h^a}$.

Finally, we show that PSR admits PO-bilinear rank decomposition (Definition 3).

**Lemma 9.** *Suppose a core test set $\mathcal{T}$ includes a minimum core test set $\mathcal{Q}$. Set $\Pi_h = [\bar{\mathcal{Z}}_h \to \Delta(\mathcal{A})]$ and $\mathcal{G}_h = \left\{ (z_{h-1}, o) \mapsto \mathbf{1}(z_{h-1})^\top \mathbb{J}[\mathbf{1}(t = o)]_{t \in \mathcal{T}} \mid \mathbb{J} \in \mathbb{R}^{|\mathcal{Z}_{h-1}| \times |\mathcal{T}|} \right\}$, the PO-bilinear rank is at most $(|\mathcal{O}||\mathcal{A}|)^M |\mathcal{Q}|$.*

Then, Algorithm 1 is directly applicable to PSRs. Note that here the PO-bilinear rank, fortunately, rank scales with $|\mathcal{Q}|$ but not $|\mathcal{T}|$. The dependence $(|\mathcal{O}||\mathcal{A}|)^M$ comes from the dimension of the "feature" of memory $\mathbf{1}(z_{h-1})$. If one has a compact feature representation $\phi : \mathcal{Z}_{h-1} \to \mathbb{R}^d$, such that $\mathcal{V}_h^\pi(\tau_h^a) = \phi(z_{h-1})^\top \mathbb{J}_h^\pi \mathbf{q}_{\tau_h^a}$ is linear with respect to feature $\phi(z_{h-1})$, then the PO-bilinear rank is $d|\mathcal{Q}|$. This implies that if one has a compact featurization of memory a priori, one can avoid exponential dependence on $M$.

**Sample complexity.** We finally briefly mention the sample complexity result. The detail is deferred to Section N. The sample complexity to satisfy $J(\pi^\star) - J(\hat{\pi}) \leq \epsilon$ is given as

$$\tilde{O}\left( \frac{|\mathcal{O}|^M |\mathcal{A}|^{M-1} |\mathcal{Q}|^2 \max(\Theta, 1) H^4 |\mathcal{A}|^2 \ln(|\mathcal{G}_{\max}||\Pi_{\max}|/\delta) \ln(\Theta_W)^2}{\epsilon^2} \right)$$

where $\Theta_W$ and $\Theta$ are some parameters associated with PSRs. Here, there is no explicit dependence on $|\mathcal{T}|$. Note that in the worst case, $\ln|\mathcal{G}_{\max}|$ scales as $O(|\mathcal{Z}_{h-1}||\mathcal{T}|)$, and $\ln|\Pi_{\max}|$ scales as $O(|\mathcal{Z}_{h-1}||\mathcal{O}||\mathcal{A}|)$.

## E.2 Definition of PSRs

We first define core tests and predictive states [45, 62]. This definition is a generalization of Definition 5 with multi-step futures.

We slighly abuse notation and denote $\tau_h^a := (o_1, a_1, \ldots, o_{h-1}, a_{h-1})$ throughout this whole section — note that $\tau_h^a$ here does not include $o_h$.

**Definition 7** (Core test sets and PSRs). *A set $\mathcal{T} \subset \cup_{C \in \mathbf{N}^+} \mathcal{O}^C \times \mathcal{A}^{C-1}$ is called a core test set if for any $h \in [H]$, $W \in \mathbf{N}^+$, any possible future (i.e., test) $t_h = (o_{h:h+W-1}, a_{h:h+W-2}) \in \mathcal{O}^W \times \mathcal{A}^{W-1}$ and any history $\tau_h^a$, there exists $m_{t_h} \in \mathbb{R}^{|\mathcal{T}|}$ such that*

$$\mathbb{P}(o_{h:W+h-1} \mid \tau_h^a; do(a_{h:W+h-2})) = \langle m_{t_h}, [\mathbb{P}(t \mid \tau_h^a)]_{t \in \mathcal{T}} \rangle.$$

*The vector $[\mathbb{P}(t \mid \tau_h^a)]_{t \in \mathcal{T}} \in \mathbb{R}^{|\mathcal{T}|}$ is referred to as the predictive state.*

We often denote $\mathbf{q}_{\tau_h^a} = [\mathbb{P}(t \mid \tau_h^a)]_{t \in \mathcal{T}_h}$. To understand the above definition, we revisit observable undercomplete POMDPs and overcomplete POMDPs.

**Example 1** (Observable undercomplete POMDPs). *In undercomplete POMDPs, when $\mathbb{O}$ is full-column rank, $\mathcal{O}$ is a core test. Recall $\mathbb{O}$ is a matrix in $\mathbb{R}^{|\mathcal{O}| \times |\mathcal{S}|}$ whose entry indexed by $o_i \in \mathcal{O}, s_j \in \mathcal{S}$ is equal to $\mathbb{O}(o_i \mid s_j)$.*

**Lemma 10** (Core tests in undercomplete POMDPs). *When $\mathbb{O}$ is full-column rank, $\mathcal{O}$ is a core test set.*

*Proof.* Consider any $h \in [H]$. Given a $|\mathcal{S}|$-dimensional belief state $\mathbf{s}_{\tau_h^a} = [\mathbb{P}(\cdot \mid \tau_h^a)]_{|\mathcal{S}|}$ with each entry $\mathbb{P}(s_h \mid \tau_h^a)$, for any future $t = (o_{h:h+W}, a_{h:h+W-1})$, there exists a $|\mathcal{S}|$-dimensional vector $\mathbf{m}'_t$ such that $\mathbb{P}(o_{h:h+W} \mid \tau_h^a; do(a_{h:h+W-1})) = \langle \mathbf{m}'_t, \mathbf{s}_{\tau_h^a} \rangle$. More specifically, $\mathbf{m}'_t$ can be written as:

$$(\mathbf{m}'_t)^\top = \mathbb{O}(o_{h+W} \mid \cdot)^\top \prod_{\tau=h}^{h+W-1} \mathbb{T}_{a_h} \text{diag}(\mathbb{O}(o_h \mid \cdot))$$

where $\mathbb{O}(o|\cdot) \in \mathbb{R}^{|\mathcal{S}|}$ is a vector with the entry indexed by $s$ equal to $\mathbb{O}(o|s)$, $\mathbb{T}_a \in \mathbb{R}^{|\mathcal{S}| \times |\mathcal{S}|}$ is a matrix with the entry indexed by $(s, s')$ equal to $\mathbb{T}(s' \mid s, a_h)$. Here, note given a vector $C$, $\mathrm{diag}(C)$ is define as a $|C| \times |C|$ diagonal matrix where the diagonal element corresponds to $C$. Thus, we have

$$\mathbb{P}(o_{h:h+W} \mid \tau_h^a; do(a_{h:h+W-1})) = \langle \mathbf{m}'_t, \mathbf{s}_{\tau_h^a} \rangle = \langle \mathbf{m}'_t, \mathbb{O}^\dagger \mathbf{q}_{\tau_h^a} \rangle = \langle (\mathbb{O}^\dagger)^\top \mathbf{m}'_t, \mathbf{q}_{\tau_h^a} \rangle,$$

where $\mathbf{q}_{\tau_h^a} \in \mathbb{R}^{|\mathcal{O}|}$ and $\mathbf{q}_{\tau_h^a}(o) = \mathbb{P}(o|\tau_h^a)$. This concludes the proof. $\qquad\square$

**Example 2** (Overcomplete POMDPs). *We consider overcomplete POMDPs so that we can permit* $|\mathcal{S}| \geq |\mathcal{O}|$.

**Lemma 11** (Core tests in overcomplete POMDPs). *Recall* $\mathcal{T}^K = \mathcal{O} \times (\mathcal{O} \times \mathcal{A})^{K-1}$. *Define a* $|\mathcal{T}^K| \times |\mathcal{S}|$-*dimensional matrix* $\mathbb{O}^K$ *whose entry indexed by* $(o_{h:h+K-1}, a_{h:h+K-2}) \in \mathcal{T}^K, s_h \in \mathcal{S}$ *is equal to* $\mathbb{P}(o_{h:h+K-1}, a_{h:h+K-2} \mid s_h; a_{h:h+K-2} \sim \mathcal{U}(\mathcal{A}))$. *When this matrix is full-colmun rank for all* $h$, $\mathcal{T}^K$ *is a core test set.*

*Proof.* Fix a test $t = (o'_{h:h+K-1}, a'_{h:h+K-2})$ and consider a step $h \in [H]$. Then,

$$\mathbb{P}(o'_{h:h+K-1}, a'_{h:h+K-2} \mid s_h; a_{h:h+K-2} \sim \mathcal{U}(\mathcal{A}))$$
$$= \mathbb{E}[\mathbf{1}(o_{h:h+K-1} = o'_{h:h+K-1}, a_{h:h+K-2} = a'_{h:h+K-2}) \mid s_h; a_{h:h+K-2} \sim \mathcal{U}(\mathcal{A}))]$$
$$= \mathbb{E}[(1/|\mathcal{A}|^{K-1})\mathbf{1}(o_{h:h+K-1} = o'_{h:h+K-1}, a_{h:h+K-2} = a'_{h:h+K-2}) \mid s_h; a_{h:h+K-2} \sim do(a'_{h:h+K-2})]$$
$$= \mathbb{E}[(1/|\mathcal{A}|^{K-1})\mathbf{1}(o_{h:h+K-1} = o'_{h:h+K-1}) \mid s_h; a_{h:h+K-2} \sim do(a'_{h:h+K-2})]$$
$$= (1/|\mathcal{A}|^{K-1})\mathbb{P}(o'_{h:h+K-1} \mid s_h; do(a'_{h:h+K-2})).$$

Thus, the assumption that $\mathcal{T}^K$ is full column rank implies that that the matrix $\bar{\mathbb{J}}_h \in \mathbb{R}^{|\mathcal{T}^K| \times |\mathcal{S}|}$ with the entry indexed by $(t, s_h)$ being equal to $\mathbb{P}(o'_{h:h+K-1} \mid s_h; do(a'_{h:h+K-2}))$ is full-column rank.

Define a $|\mathcal{T}^K|$-dimensional state $\mathbf{q}_{\tau_h^a} = [\mathbb{P}(t \mid \tau_h^a)]_{t \in \mathcal{T}^K}$ given history $\tau_h^a$. By definition, we have

$$\mathbf{q}_{\tau_h^a} = \bar{\mathbb{J}}_h \mathbf{s}_{\tau_h^a}$$

Using $\bar{\mathbb{J}}_h$ is full-column rank, we have $\mathbf{s}_{\tau_h^a} = \bar{M}_h^\dagger \mathbf{q}_{\tau_h^a}$. Thus, using the format of $\mathbf{m}'_t$ from the proof of Lemma 10, we can conclude that for any test $t = (o_{h:h+W}, a_{h:h+W-1})$, we have $\mathbb{P}(o_{h:h+W}|\tau; do(a_{h:h+W-1})) = \langle (\bar{M}_h^\dagger)^\top \mathbf{m}'_t, \mathbf{q}_{\tau_h^a} \rangle$. Thus, this concludes $\mathcal{T}^K$ is a core test set. $\quad\square$

Finally, we present an important property of predictive states, which corresponds to the Bayesian filter in POMDP.

**Lemma 12** (Forward dynamics of predictive states). *We have*

$$\mathbb{P}(t \mid \tau_h^a, a, o) = \mathbf{m}_{o,a,t}^\top \mathbf{q}_{\tau_h^a} / \mathbf{m}_o^\top \mathbf{q}_{\tau_h^a}.$$

*When we define* $M_{o,a} \in \mathbb{R}^{|\mathcal{T}| \times |\mathcal{T}|}$ *where rows are* $\mathbf{m}_{o,a,t}$ *for* $t \in \mathcal{T}$, *we can express the forward update rule of predictive states as follows:*

$$\mathbf{q}_{\tau_h^a, a, o} = M_{o,a} \mathbf{q}_{\tau_h^a} / (\mathbf{m}_o^\top \mathbf{q}_{\tau_h^a}).$$

*Proof.* The proof is an application of Bayes's rule. We denote the observation part of $t$ by $t^{\mathcal{O}}$ and the action part of $t^{\mathcal{A}}$, respectively. We have

$$\mathbb{P}(t \mid \tau_h^a, a, o) = \mathbb{P}(t^{\mathcal{O}} \mid \tau_h^a, o; do(a, t^{\mathcal{A}})) \qquad \text{(by definition)}$$
$$= \frac{\mathbb{P}(o, t^{\mathcal{O}} \mid \tau_h^a; do(a, t^{\mathcal{A}}))}{\mathbb{P}(o; \tau_h^a)} \qquad \text{(Bayes rule)}$$
$$= \mathbf{m}_{o,a,t}^\top \mathbf{q}_{\tau_h^a} / \mathbf{m}_o^\top \mathbf{q}_{\tau_h^a}. \qquad \text{(by definition)}$$

This concludes the proof. $\qquad\square$

To further understand that why PSR generalizes POMDP, let us re-visit the undercomplete POMDPs (i.e., $\mathbb{O}$ being full column rank) again. Set $\mathcal{T} = \mathcal{O}$. As we see in the proof of Lemma 10, the belief state $\mathbf{s}_\tau \in \Delta(\mathcal{S})$ together with $\mathbb{O}$ defines predictive state, i.e., $\mathbf{q}_{\tau_h^a} = \mathbb{O}\mathbf{s}_{\tau_h^a}$, with $M_{o,a} = \mathbb{O}\mathbb{T}_a\mathrm{diag}(\mathbb{O}(o|\cdot))\mathbb{O}^\dagger$, and $\mathbf{m}_o^\top = \mathbf{1}^\top\mathrm{diag}(\mathbb{O}(o|\cdot))\mathbb{O}^\dagger$. Note that in POMDPs, matrix $M_{o,a}$ and vector $\mathbf{m}_o$ all contain non-negative entries. On other hand, in PSRs, $M_{a,o}$ and $\mathbf{m}_{a,o}$ could contain negative entries. This is the intuitive reason why PSRs are more expressive than POMDPs [45]. For the formal instance of a finite-dimensional PSR which cannot be expressed as a finite-dimensional POMDP, refer to [62, 31].

### E.3 Existence of future-dependent value functions

We discuss the existence and the form of future-dependent value functions. First, we define general future-dependent value functions with multi-step futures. For notational simplicity, we assume here that the tests $t \in \mathcal{T}$ have the same length, i.e., there is a $K \in \mathbb{N}^+$, such that $\mathcal{T} \subset \mathcal{O}^K \times \mathcal{A}^{K-1}$.

**Definition 8** (General future-dependent value functions in dynamical systems). *Recall $\mathcal{T} \subset \mathcal{O}^K \times \mathcal{A}^{K-1}$ is the set of tests. At time step $h$, general future-dependent value functions $g_h^\pi : \mathcal{Z}_{h-1} \times \mathcal{O}^K\mathcal{A}^{K-1} \to \mathbb{R}$ are defined as solutions to the following:*

$$\mathcal{V}_h^\pi(\tau_h^a) = \mathbb{E}[g_h^\pi(z_{h-1}, o_{h:h+K-1}, a_{h:h+K-2}) \mid \tau_h^a; (a_{h:h+K-2}) \sim \rho^{out}]. \tag{5}$$

*where $\rho^{out}$ is some distribution over the action set $\mathcal{T}^\mathcal{A}$ induced by the test set, i.e., $\{t^\mathcal{A} : t \in \mathcal{T}\}$. Here, for $t = (o_{h:h+K-1}, a_{h:h+K-2})$, we often denote $o_{h:h+K-1}$ and $a_{h:h+K-2}$ by $t^\mathcal{O}$ and $t^\mathcal{A}$, respectively.*

To show the existence of general future-dependent value functions for PSRs, we first study the format of value functions in PSRs. The following lemma states that value functions for $M$-memory policies have bilinear forms.

**Lemma 13** (Bilinear form of value functions for $M$-memory policies). *Let $\phi(\cdot) \in \mathbb{R}^{|\mathcal{Z}_{h-1}|}$ be a one-hot encoding vector over $\mathcal{Z}_{h-1}$. Suppose $\mathcal{T}$ is a core test set. Then, for any $M$-memory policy $\pi$, there exists $\mathbb{J}_h^\pi \in \mathbb{R}^{|\mathcal{Z}_{h-1}| \times |\mathcal{T}|}$ such that*

$$\mathcal{V}_h^\pi(\tau_h^a) = \phi^\top(z_{h-1})\mathbb{J}_h^\pi\mathbf{q}_{\tau_h^a}.$$

*Proof.* From Lemma 12, there exists a matrix $M_{o,a} \in \mathbb{R}^{|\mathcal{T}| \times |\mathcal{T}|}$ such that via Bayes rule:

$$\mathbf{q}_{\tau_h^a,a,o} = M_{o,a}\mathbf{q}_{\tau_h^a}/\mathbb{P}(o|\tau_h^a). \tag{6}$$

We use induction to prove the claim. Here, the base argument clearly holds. Thus, we assume

$$\mathcal{V}_{h+1}^\pi(\tau_{h+1}^a) = \phi^\top(z_h)\mathbb{J}_{h+1}^\pi\mathbf{q}_{\tau_{h+1}^a}.$$

We have

$$\begin{aligned}
\mathcal{V}_h^\pi(\tau_h^a) &= \mathbb{E}[r_h + \mathcal{V}_{h+1}^\pi(\tau_h^a, o_h, a_h) \mid \tau_h^a; a_h \sim \pi(\bar{z}_h)] \\
&= \underbrace{\sum_{o_h,a_h} \mathbb{P}(o_h \mid \tau_h^a)\pi_h(a_h \mid o_h, z_{h-1})r(o_h, a_h)}_{(a)} \\
&\quad + \underbrace{\sum_{o_h,a_h} \mathbb{P}(o_h \mid \tau_h^a)\pi_h(a_h \mid o_h, z_{h-1})\{\phi^\top(z_h)\mathbb{J}_{h+1}^\pi\mathbf{q}_{\tau_h^a,o_h,a_h}\}}_{(b)}.
\end{aligned}$$

Note we use the assumption that the reward is a function of $o_h, a_h$ conditional on $(\tau_h^a, o_h, a_h)$.

We first check the first term (a) that contains rewards. Using the fact that $\mathbb{P}(o|\tau_h^a) = \mathbf{m}_o^\top\mathbf{q}_{\tau_h^a}$, this is equal to

$$\sum_{o_h,a_h} \langle\mathbf{m}_{o_h}, \mathbf{q}_{\tau_h^a}\rangle\pi_h(a_h \mid o_h, z_{h-1})r(o_h, a_h) = \langle\sum_{o_h,a_h} \mathbf{m}_{o_h}\pi_h(a_h \mid o_h, z_{h-1})r(o_h, a_h), \mathbf{q}_{\tau_h^a}\rangle.$$

Thus, it has a bilinear form, i.e., there exists some matrix $\mathbb{J}_1^\pi$ such that

$$\langle \sum_{o_h,a_h} m_{o_h,a_h} \pi_h(a_h \mid o_h, z_{h-1}) r_h, \mathbf{q}_{\tau_h^a} \rangle = \phi^\top(z_{h-1}) \mathbb{J}_1^\pi \mathbf{q}_{\tau_h^a}$$

where $\mathbb{J}_1^\pi$ is a matrix whose row indexed by $z_{h-1}$ is equal to $\sum_{o,a} \mathbf{m}_o^\top \pi_h(a|o, z_{h-1}) r(o, a)$.

Next, we see the second term (b). Using (6), the second term is equal to

$$\sum_{o_h,a_h} \pi_h(a_h \mid o_h, z_{h-1}) \phi^\top(z_{h-1} \oplus o_h, a_h) \mathbb{J}_{h+1}^\pi M_{o_h,a_h} \mathbf{q}_{\tau_h^a}$$

where we use the notation $z_{h-1} \oplus o, a$ to represent the operation of appending $(o, a)$ pair to the memory while maintaining the proper length of the memory by truncating away the oldest observation-action pair. Thus, it has a again bilinear form $\phi(z_{h-1})^\top \mathbb{J}_2^\pi \mathbf{q}_{\tau_h^a}$ and the matrix $\mathbb{J}_2^\pi$ can be defined such that its row indexed by $z_{h-1}$ is equal to $\sum_{o,a} \pi_h(a|o, z_{h-1}) \phi^\top(z_{h-1} \oplus o, a) M_{h+1}^\pi M_{a,o}$. This concludes the proof. □

Next, we check sufficient conditions to ensure the existence of general K-step future-dependent value functions. Given $\mathcal{T}$, we define the corresponding set of action sequences $\mathcal{T}^\mathcal{A}$ as $\mathcal{T}^\mathcal{A} := \{t^\mathcal{A} : t \in \mathcal{T}\}$. We set $\rho^{out}$ in (5) to be a uniform distribution over the set $\mathcal{T}^\mathcal{A}$ denoted by $\mathcal{U}(\mathcal{T}^\mathcal{A})$. Namely, $\mathcal{U}(\mathcal{T}^\mathcal{A})$ will uniformly randomly select a sequence of test actions from $\mathcal{T}^\mathcal{A}$.

**Lemma 14** (Existence of future-dependent value functions in PSRs). *Suppose $\mathcal{T}$ is a core test. There exists $g_h^\pi : \mathcal{Z}_{h-1} \times \mathcal{T}$ such that*

$$\mathbb{E}[g_h^\pi(z_{h-1}, o_{h:h+K-1}, a_{h:h+K-2}) \mid \tau_h^a; a_{h:h+K-2} \sim \mathcal{U}(\mathcal{T}^\mathcal{A})] = \mathcal{V}_h^\pi(\tau_h^a).$$

*Proof.* We mainly need to design an unbiased estimator of the predictive state $\mathbf{q}_{\tau_h^a}$. We use importance weighting to do that. Given $a_{h:h+K-2} \sim \mathcal{U}(\mathcal{T}^\mathcal{A})$, and the resulting corresponding random observations $o_{h:h+K-1}$, we define the following estimator $\hat{\mathbf{q}}_{\tau_h^a}(o_{h:h+K-1}, a_{h:h+K-2}) \in \mathbb{R}^{|\mathcal{T}|}$, such that its entry indexed by a test $t \in \mathcal{T}$ is equal to:

$$\hat{\mathbf{q}}_{\tau_h^a}(o_{h:h+K-1}, a_{h:h+K-2})[t] = \frac{\mathbf{1}(t^\mathcal{O} = o_{h:h+K-1}, t^\mathcal{A} = a_{h:h+K-2})}{1/|\mathcal{T}^\mathcal{A}|}.$$

We can verify that

$$\mathbb{E}[\hat{\mathbf{q}}_{\tau_h^a}(o_{h:h+K-1}, a_{h:h+K-2})[t] \mid \tau_h^a; a_{h:h+K-2} \sim \mathcal{U}(\mathcal{T}^\mathcal{A})]$$
$$= 1/|\mathcal{T}^\mathcal{A}| \mathbb{E}[\mathbf{1}(t^\mathcal{O} = o_{h:h+K-1}, t^\mathcal{A} = a_{h:h+K-2}) \mid \tau_h^a; a_{h:h+K-2} \sim \mathcal{U}(\mathcal{T}^\mathcal{A})]$$
$$= \mathbb{E}[\mathbf{1}(t^\mathcal{O} = o_{h:h+K-1}, t^\mathcal{A} = a_{h:h+K-2}) \mid \tau_h^a; a_{h:h+K-2} \sim do(t^\mathcal{A})] = \mathbf{q}_{\tau_h^a}[t].$$

Then,

$$\mathbb{E}[\hat{\mathbf{q}}_{\tau_h^a}(o_{h:h+K-1}, a_{h:h+K-2}) \mid \tau_h^a; a_{h:h+K-2} \sim \mathcal{U}(\mathcal{T}^\mathcal{A})] = \mathbf{q}_{\tau_h^a}.$$

With this estimator, now we can define the future-dependent value function using the bilinear form of $\mathcal{V}_h^\pi(\tau)$, i.e.,

$$g_h^\pi(z_{h-1}, o_{h:h+K-1}, a_{h:h+K-2}) = \phi(z_{h-1})^\top \mathbb{J}_h^\pi \hat{\mathbf{q}}_{\tau_h^a}(o_{h:h+K-1}, a_{h:h+K-2}).$$

Using the fact that $\hat{\mathbf{q}}_{\tau_h^a}(o_{h:h+K-1}, a_{h:h+K-2})$ is an unbiased estimate of $\mathbf{q}_{\tau_h^a}$, we can conclude the proof. □

Since PSR models capture POMDP models, our above result directly implies the existence of the future-dependent value functions in observable POMDPs as well by using obtained facts in Example 1 and 2.

## E.4 PO-Bilinear Rank Decomoposition

Finally, we calculate the PO-bilinear rank. Here,

$$g_h \in \{ \mathcal{Z}_{h-1} \times (\mathcal{O}^K \mathcal{A}^{K-1}) \ni (z_{h-1} \times t) \mapsto \phi(z_{h-1})^\top \mathbb{J}_h^\pi \hat{\mathbf{q}}_{\tau_h^a}(t) \in \mathbb{R} : \mathbb{J}_h^\pi \in \mathbb{R}^{\mathcal{Z}_{h-1} \times |\mathcal{T}|} \}.$$

The Bellman error for $(g, \pi)$ under a roll-in $\pi'$ denoted by $\mathrm{Br}_h(g, \pi; \pi')$ is defined as

$$- \mathbb{E}[\mathbb{E}[g_{h+1}(z_h, t_{h+1}^{\mathcal{A}}, t_{h+1}^{\mathcal{O}}) \mid \tau_{h+1}^a; t_{h+1}^{\mathcal{A}} \sim U(\mathcal{T}_{h+1}^{\mathcal{A}})] + r_h; a_{1:h-1} \sim \pi', a_h \sim \pi]$$
$$+ \mathbb{E}[g_h(z_{h-1}, t_h^{\mathcal{A}}, t_h^{\mathcal{O}}) \mid \tau_h^a; t_h^{\mathcal{A}} \sim U(\mathcal{T}_h^{\mathcal{A}})]; a_{1:h-1} \sim \pi', a_h \sim \pi].$$

In fact, $\mathrm{Br}_h(g, \pi; \pi') = 0$ for any general future-dependent value functions $g^\pi$.

Our goal is to design a loss function $l_h(\cdot)$ such that we can estimate the above Bellman error $\mathrm{Br}_h(g, \pi; \pi')$ using data from a *single* policy. To do that, we design the following randomized action selection strategy.

Given a action sequence $t^{\mathcal{A}}$ from a test $t$, let us denote $\bar{t}^{\mathcal{A}}$ as a copy of $t^{\mathcal{A}}$ but starting from the second action of $t^{\mathcal{A}}$, i.e., if $t^{\mathcal{A}} = \{a_1, a_2, a_3\}$, then $\bar{t}^{\mathcal{A}} = \{a_2, a_3\}$. Denote $\bar{\mathcal{T}}^{\mathcal{A}} = \{\bar{t}^{\mathcal{A}} : t \in \mathcal{T}\}$. Our random action selection strategy first selects $a_h \sim U(\mathcal{A})$ uniformly randomly from $\mathcal{A}$, and then select a sequence of actions $\bar{\mathbf{a}}$ uniformly randomly from $\mathcal{T}^{\mathcal{A}} \cup \bar{\mathcal{T}}^{\mathcal{A}}$. Here, we remark the length of outputs is not fixed (i.e., $\bar{\mathbf{a}} \in \mathcal{T}^{\mathcal{A}}$ has length larger than the $\bar{\mathbf{a}} \in \bar{\mathcal{T}}^{\mathcal{A}}$).

As a first step, we define two unbiased estimators for $\mathbf{q}_{\tau_h^a}$ and $\mathbf{q}_{\tau_{h+1}^a}$. Conditioning on history $\tau_h^a$, given actions $a_h \sim U(\mathcal{A})$ followed by action sequence $\bar{\mathbf{a}}_{h+1} \sim U(\mathcal{T}^{\mathcal{A}} \cup \bar{\mathcal{T}}^{\mathcal{A}})$, denote the corresponding observations as $o_h, o_{h+1}, \ldots o_{h+|\bar{\mathbf{a}}_{h+1}|+1}$. We construct unbiased estimators for $\mathbf{q}_{\tau_h^a}$ and $\mathbf{q}_{\tau_{h+1}^a}$ as follows. As an unbiased estimator of $\mathbf{q}_{\tau_h^a}$, we define $\hat{\mathbf{q}}_{\tau_h^a}$ with the entry indexed by test $t' \in \mathcal{T}$ as follows:

$$\hat{\mathbf{q}}_{\tau_h^a}(a_h, \bar{\mathbf{a}}_{h+1}, o_{h:h+|\bar{\mathbf{a}}_{h+1}|+1})[t'] = \frac{\mathbf{1}(\bar{\mathbf{a}}_{h+1} \in \bar{\mathcal{T}}^{\mathcal{A}}, (a_h, \bar{\mathbf{a}}_{h+1}) = t'^{\mathcal{A}}, o_{h:h+|\bar{\mathbf{a}}_{h+1}|+1} = t'^{\mathcal{O}})}{1/(2|\mathcal{A}||\mathcal{T}^{\mathcal{A}}|)}. \quad (7)$$

Similarly, as an unbiased estimator of $\mathbf{q}_{\tau_{h+1}^a}$, we define $\hat{\mathbf{q}}_{\tau_{h+1}^a}$ with the entry indexed by test $t' \in \mathcal{T}$ as follows:

$$\hat{\mathbf{q}}_{\tau_{h+1}^a}(a_h, \bar{\mathbf{a}}_{h+1}, o_{h+1:h+|\bar{\mathbf{a}}_{h+1}|+1})[t'] = \frac{\mathbf{1}(\bar{\mathbf{a}}_{h+1} \in \mathcal{T}^{\mathcal{A}}, \bar{\mathbf{a}}_{h+1} = t'^{\mathcal{A}}, o_{h+1:h+|\bar{\mathbf{a}}_{h+1}|+1} = t'^{\mathcal{O}})}{1/(2|\mathcal{T}^{\mathcal{A}}|)} \quad (8)$$

We remark the length of $\bar{\mathbf{a}}$ in (7) and the one of (8) are different.

Then, by using importance sampling, we can verify

$$\mathbb{E}[\hat{\mathbf{q}}_{\tau_h^a}(a_h, \bar{\mathbf{a}}_{h+1}, o_{h:h+|\bar{\mathbf{a}}_{h+1}|+1})|\tau_h^a; a_h \sim U(\mathcal{A}), \bar{\mathbf{a}}_{h+1} \sim U(\mathcal{T}^{\mathcal{A}} \cup \bar{\mathcal{T}}^{\mathcal{A}})] = \mathbf{q}_{\tau_h^a},$$
$$\mathbb{E}[\hat{\mathbf{q}}_{\tau_{h+1}^a}(\bar{\mathbf{a}}_{h+1}, o_{h+1:h+|\bar{\mathbf{a}}_{h+1}|+1})|\tau_{h+1}^a; \bar{\mathbf{a}}_{h+1} \sim U(\mathcal{T}^{\mathcal{A}} \cup \bar{\mathcal{T}}^{\mathcal{A}})] = \mathbf{q}_{\tau_{h+1}^a}.$$

With the above setup, we can construct the loss function $l$ for estimating the Bellman error. We set the loss as follows:

$$l_h(z_{h-1}, a_h, r_h, \bar{\mathbf{a}}_{h+1}, o_{h:h+|\bar{\mathbf{a}}_{h+1}|+1}; \pi, g) \quad (9)$$
$$= \phi(z_{h-1})^\top \mathbb{J}_h \hat{\mathbf{q}}_{\tau_h^a}(a_h, \bar{\mathbf{a}}_{h+1}, o_{h:h+|\bar{\mathbf{a}}_{h+1}|+1})$$
$$- \frac{\mathbf{1}\{a_h = \pi_h(\bar{z}_h)\}}{1/|\mathcal{A}|} \left( r_h + \phi(z_h)^\top \mathbb{J}_{h+1} \hat{\mathbf{q}}_{\tau_{h+1}^a}(\bar{\mathbf{a}}_{h+1}, o_{h+1:h+|\bar{\mathbf{a}}_{h+1}|+1}) \right).$$

Since we have shown that $\hat{\mathbf{q}}_{\tau_h^a}$ and $\hat{\mathbf{q}}_{\tau_{h+1}^a}$ are unbiased estimators of $\mathbf{q}_\tau$ and $\mathbf{q}_{\tau_{h+1}^a}$, respectively, we can show that for any roll-in policy $\pi'$:

$$\mathrm{Br}_h(\pi, g; \pi')$$
$$= -\mathbb{E}[\mathbb{E}[g_{h+1}(z_h, t_{h+1}^{\mathcal{A}}, t_{h+1}^{\mathcal{O}}) \mid \tau_{h+1}^a; t_{h+1}^{\mathcal{A}} \sim U(\mathcal{T}_{h+1}^{\mathcal{A}})] + r_h; a_{1:h-1} \sim \pi', a_h \sim \pi]$$
$$+ \mathbb{E}[g_h(z_{h-1}, t_h^{\mathcal{A}}, t_h^{\mathcal{O}}) \mid \tau_h^a; t_h^{\mathcal{A}} \sim U(\mathcal{T}_h^{\mathcal{A}})]; a_{1:h-1} \sim \pi', a_h \sim \pi]$$
$$= \mathbb{E}[-\phi(z_h)^\top \mathbb{J}_{h+1}^\pi \mathbf{q}_{\tau_{h+1}^a} - r_h + \phi(z_{h-1})^\top \mathbb{J}_h^\pi \mathbf{q}_{\tau_h^a}; a_{1:h-1} \sim \pi', a_h \sim \pi]$$
$$= \mathbb{E}\left[ l_h(z_{h-1}, a_h, r_h, \bar{\mathbf{a}}_{h+1}, o_{h:h+|\bar{\mathbf{a}}_{h+1}|+1}; \pi, g); a_{1:h-1} \sim \pi', a_h \sim U(\mathcal{A}), \bar{\mathbf{a}}_{h+1} \sim U(\mathcal{T}^{\mathcal{A}} \cup \bar{\mathcal{T}}^{\mathcal{A}}) \right].$$

The above shows that we can use $l_h(\cdot)$ as a loss function.

**Summary** We can use the almost similar algorithm as Algorithm 1. The sole difference is we need to replace $\sigma_h^t(\pi, g)$ with

$$\mathbb{E}_{\mathcal{D}_h^t}\left[l_h(z_{h-1}, a_h, r_h, \bar{\mathbf{a}}_{h+1}, o_{h:h+|\bar{\mathbf{a}}_{h+1}|+1}; \pi, g); a_{1:h-1} \sim \pi', a_h \sim U(\mathcal{A}), \bar{\mathbf{a}}_{h+1} \sim U(\mathcal{T}^{\mathcal{A}} \cup \bar{\mathcal{T}}^{\mathcal{A}})\right]$$

where $\mathcal{D}_h^t$ is an empirical approximation when executing $a_{1:h-1} \sim \pi^t, a_h \sim U(\mathcal{A}), \bar{\mathbf{a}}_{h+1} \sim U(\mathcal{T}^{\mathcal{A}} \cup \bar{\mathcal{T}}^{\mathcal{A}})$.

**Calculation of PO-bilinear rank** Finally, we prove a PSR belongs to the PO-bilinear class.

**Lemma 15** (PO-bilinear decomposition)**.** *Let $\mathcal{Q}$ be a minimum core test set contained in $\mathcal{T}$. The PSR model has PO-bilinear rank at most $|\mathcal{O}|^M|\mathcal{A}|^M|\mathcal{Q}|$, i.e., there exists two $|\mathcal{O}|^M|\mathcal{A}|^M|\mathcal{Q}|$-dimensional mappings $W_h : \Pi \times \mathcal{G} \to \mathbb{R}^{|\mathcal{O}|^M|\mathcal{A}|^M|\mathcal{Q}|}$ and $X_h : \Pi \to \mathbb{R}^{|\mathcal{O}|^M|\mathcal{A}|^M|\mathcal{Q}|}$ such that for any tripe $(\pi, g; \pi')$, we have:*

$$\mathrm{Br}_h(\pi, g; \pi') = \mathbb{E}\left[\phi(z_{h-1})^\top \mathbb{J}_h \mathbf{q}_{\tau_h^a} - r_h - \phi(z_h)^\top \mathbb{J}_{h+1}\mathbf{q}_{\tau_{h+1}^a}; a_{1:h-1} \sim \pi', a_h \sim \pi\right]$$
$$= \langle X_h(\pi'), W_h(\pi, g)\rangle.$$

*Proof.* We first take expectation conditional on $\tau_h^a$. Then, we have

$$\phi(z_{h-1})^\top \mathbb{J}_h \mathbf{q}_{\tau_h^a} - \mathbb{E}\left[r_h + \phi(z_h)^\top \mathbb{J}_{h+1}\mathbf{q}_{\tau_{h+1}^a} \mid \tau_h^a; a_h \sim \pi\right]$$
$$= \phi(z_{h-1})^\top \mathbb{J}_h \mathbf{q}_{\tau_h^a} + \left(\phi(z_{h-1})^\top \mathbb{J}_1^\pi \mathbf{q}_{\tau_h^a} + \phi(z_{h-1})^\top \mathbb{J}_2^\pi \mathbf{q}_{\tau_h^a}\right),$$

where $\mathbb{J}_1^\pi$ and $\mathbb{J}_2^\pi$ are some two matrices as defined in the proof of Lemma 13 from where we have already known that the $\pi$-induced Bellman backup on a value function which has a bilinear form gives back a bilinear form value function. Rearrange terms, we get:

$$\phi(z_{h-1})^\top \mathbb{J}_h \mathbf{q}_{\tau_h^a} - \mathbb{E}\left[r_h + \phi(z_h)^\top \mathbb{J}_{h+1}\mathbf{q}_{\tau_{h+1}^a} \mid \tau_h^a; a_h \sim \pi\right] = \left\langle \phi(z_{h-1}), (\mathbb{J}_h + \mathbb{J}_1^\pi + \mathbb{J}_2^\pi)\mathbf{q}_{\tau_\mathbf{h}^\mathbf{a}}\right\rangle.$$

Now recall that the minimum core test set is $\mathcal{Q} \subset \mathcal{T}$. The final step is to argue that $\mathbf{q}_\tau$ lives in a subspace whose dimension is $|\mathcal{Q}|$. Since $\mathcal{Q}$ is a core test set, by definition, we can express $\mathbf{q}_{\tau_h^a}$ using $[\mathbb{P}(t|\tau_h^a)]_{t \in \mathcal{Q}}$, i.e.,

$$\exists K \in \mathbb{R}^{|\mathcal{T}| \times |\mathcal{Q}|}, \quad \mathbf{q}_{\tau_h^a} = K[\mathbb{P}(t|\tau_h^a)]_{t \in \mathcal{Q}},$$

where the row of $K$ indexed by $t \in \mathcal{T}$ is equal to $\mathbf{k}_t$, where $\mathbf{k}_t$ is the vector that is used to predict $\mathbb{P}(t|\tau_h^a) = \mathbf{k}_t^\top [\mathbb{P}(t|\tau_h^a)]_{t \in \mathcal{Q}}$ whose existences is ensured by the definition of PSRs. This implies that

$$\left\langle \phi(z_{h-1}), (\mathbb{J}_h + \mathbb{J}_1^\pi + \mathbb{J}_2^\pi)\mathbf{q}_{\tau_\mathbf{h}^\mathbf{a}}\right\rangle = \left\langle \phi(z_{h-1}), (\mathbb{J}_h + \mathbb{J}_1^\pi + \mathbb{J}_2^\pi)K[\mathbb{P}(t|\tau_h^a)]_{t \in \mathcal{Q}}\right\rangle$$
$$= (\phi(z_{h-1}) \otimes [\mathbb{P}(t|\tau_h^a)]_{t \in \mathcal{Q}}, \mathrm{vec}((\mathbb{J}_h + \mathbb{J}_1^\pi + \mathbb{J}_2^\pi)K)).$$

Finally, we take expectation with respect to $\tau_h^a$ then we get $\mathrm{Br}_h(\pi, g; \pi') = \langle X_h(\pi'), W_h(\pi, g)\rangle$ such that

$$X_h(\pi') = \phi(z_{h-1}) \otimes \mathbb{E}[[\mathbb{P}(t|\tau_h^a)]_{t \in \mathcal{Q}}; a_{1:h-1} \sim \pi'], \quad W_h(\pi, g) = \mathrm{vec}((\mathbb{J}_h + \mathbb{J}_1^\pi + \mathbb{J}_2^\pi)K).$$

$\square$

The key observation here is that the bilinear rank scales with $|\mathcal{Q}|$ but not $|\mathcal{T}|$. This is good news since we often cannot identify exact minimal core test sets; however, it is easy to find core tests including minimal core tests. Thus, even if we do not know the linear dimension of a dynamical system a priori, the resulting bilinear rank is the linear dimension of dynamical systems as long as core sets are large enough so that they include minimal core tests. This will result in the benefit of sample complexity as we will see Section N.

# F  Generalization of PO-Bilinear AC Class

We extend our previous definition of PO-Bilinear AC framework. We first present an even more general framework that captures all the previous examples that we have discussed so far. We then provide two more examples that can be covered by this framework: (1) $M$-step decodable POMDPs, and (2) observable POMDPs with low-rank latent transition. Using the result in (2), we can obtain refined results in the tabular setting compared to the result from Section 5.3.2.

The following is a general PO-Bilinear AC Class. Recall $M(h) := \max(h - M, 1)$. We consider one-step future, i.e., $K = 1$, but the extension to $K > 1$ is straightforward. Comparing to Definition 3, we introduce another class of functions termed as discriminators $\mathcal{F}$ and the loss function $l$.

**Definition 9** (General PO-Bilinear AC Class). *Consider a tuple $\langle \Pi, \mathcal{G}, l, \Pi^e, \mathcal{F} \rangle$ consisting of a policy class $\Pi$, a function class $\mathcal{G}$, a loss function $l = \{l_h\}_{h=1}^H$ where $l_h(\cdot; f, \pi, g) : \mathcal{H}_{h-1} \times \mathcal{O} \times \mathcal{A} \times \mathbb{R} \times \mathcal{O} \to \mathbb{R}$, a set of estimation policies $\Pi^e := \{\pi^e(\pi) : \pi \in \Pi\}$ where $\pi_h^e(\pi) : \bar{\mathcal{Z}}_h \to \Delta(\mathcal{A})$, and a discriminator class $\mathcal{F} = \{\mathcal{F}_h\}$ with $\mathcal{F}_h \subset [\mathcal{H}_h \to \mathbb{R}]$. Consider a non-decreasing function $\zeta : \mathbb{R}^+ \to \mathbb{R}$ with $\zeta(0) = 0$.*

*The model is a PO-bilinear class of rank $d$ if $\mathcal{G}$ is realizable, and there exist $W_h : \Pi \times \mathcal{G} \to \mathbb{R}^d$ and $X_h : \Pi \to \mathbb{R}^d$ such that for all $\pi, \pi' \in \Pi, g \in \mathcal{G}$ and $h \in [H]$,*

*(a)* $|\mathbb{E}[g_h(\bar{z}_h) - r_h - g_{h+1}(\bar{z}_{h+1}); a_{1:h} \sim \pi]| \leq |\langle W_h(\pi, g), X_h(\pi) \rangle|$,

*(b)*

$$\zeta(\max_{f \in \mathcal{F}_h} |\mathbb{E}[l_h(\tau_h, a_h, r_h, o_{h+1}; f, \pi, g); a_{1:M(h)-1} \sim \pi', a_{M(h):h} \sim \pi^e(\pi')]|) \geq |\langle W_h(\pi, g), X_h(\pi') \rangle|.$$

*(In $M$-step decodable POMDPs and POMDPs with low-rank latent transition, we set $\pi^e(\pi) = \mathcal{U}(\mathcal{A})$ and in the previous sections, we set $\pi^e(\pi') = \pi'$. )*

*(c)*

$$\max_{f \in \mathcal{F}_h} |\mathbb{E}[l_h(\tau_h, a_h, r_h, o_{h+1}; f, \pi, g^\pi); a_{1:M(h)-1} \sim \pi', a_{M(h):h} \sim \pi^e(\pi')]| = 0$$

*for any $\pi \in \Pi$ and the corresponding future-dependent value function $g^\pi$ in $\mathcal{G}$ .*

The first condition states the average Bellman error under $\pi$ is upper-bounded by the quantity in the bilinear form. The second condition states that we have a known loss function $l$ that can be used to estimate an upper bound (up to a non-decreasing transformation $\zeta$) of the value of the bilinear form. Our algorithm will use the surrogate loss $l(\cdot)$. As we will show, just being able to estimate an upper bound of the value of the bilinear form suffices for deriving a PAC algorithm. The discriminator $\mathcal{F}$ and the non-decreasing functions $\zeta$ give us additional freedom to design the loss function. For simple examples such as tabular POMDPs and LQG, as we already see, we simply set the discriminator class $\mathcal{F} = \emptyset$ (i.e., we do not use discriminators) and $\zeta$ being the identity mapping.

With this definition, we slightly modify PROVABLE to incorporate the discriminator to construct constraints. The algorithm is summarized in Algorithm 3 that is named as DISPROVABLE. There are two modifications: (1) when we collect data, we switch from the roll-in policy $\pi^t$ to the policy $\pi^e$ at time step $M(h)$; (2) the Bellman error constraint $\sigma_h^t$ is defined using the loss $l$ together with the discriminator class $\mathcal{F}_h$.

The following theorem shows the sample complexity of Algorithm 3. For simplicity, we direct consider the case where $\Pi, \mathcal{G}, \mathcal{F}$ are all discrete.

**Assumption 3** (Uniform Convergence). *Fix $h \in [H]$. Let $\mathcal{D}_h'$ be a set of $m$ i.i.d tuples by executing $a_{1:M(h)-1} \sim \pi^t, a_{M(h):h} \sim \pi^e$ With probability $1 - \delta$,*

$$\sup_{\pi \in \Pi, g \in \mathcal{G}, f \in \mathcal{F}} |(\mathbb{E}_{\mathcal{D}_h'} - \mathbb{E})[l_h(\tau_h, a_h, r_h, o_{h+1}; f, \pi, g)]| \leq \epsilon_{gen,h}(m, \Pi, \mathcal{G}, \mathcal{F}, \delta)$$

*For $h = 1$, we also require*

$$\sup_{g_1 \in \mathcal{G}_1} |\mathbb{E}_{\mathcal{D}_1'}[g_1(o_1)] - \mathbb{E}[\mathbb{E}_{\mathcal{D}_1'}[g_1(o_1)]]| \leq \epsilon_{ini,1}(m, \mathcal{G}, \delta).$$

**Theorem 8** (Sample complexity of Algorithm 3). *Suppose we have a PO-bilinear AC class with rank $d$ in Definition 9. Suppose Assumption 3, $\sup_{\pi \in \Pi} \|X_h(\pi)\| \leq B_X$ and $\sup_{\pi \in \Pi, g \in \mathcal{G}} \|W_h(\pi, g)\| \leq$*

---

**Algorithm 3** PaRtially ObserVAble BiLinEar with DIScriminators (DISPROVABLE)

---

1: **Input:** Value future-dependent value function class $\mathcal{G} = \{\mathcal{G}_h\}, \mathcal{G}_h \subset [\bar{\mathcal{Z}}_h \to \mathbb{R}]$, discriminator class $\mathcal{F} = \{\mathcal{F}_h\}, \mathcal{F}_h \subset [\mathcal{H}_h \to \mathbb{R}]$, policy class $\Pi = \{\Pi_h\}, \Pi_h \subset [\bar{\mathcal{Z}}_h \to \Delta(\mathcal{A})]$, parameters $m \in \mathbb{N}, R \in \mathbb{R}$
2: Initialize $\pi^0 \in \Pi$
3: Form the first step dataset $\mathcal{D}^0 = \{o^i\}_{i=1}^m$, with $o^i \sim \mathbb{O}(\cdot|s_1)$
4: **for** $t = 0 \to T-1$ **do**
5:     For any $h \in [H]$, define the Bellman error
$$\forall(\pi, g) \in \Pi \times \mathcal{G} : \sigma_h^t(\pi, g) := \max_{f \in \mathcal{F}_h} |\mathbb{E}_{\mathcal{D}_h^t} [l_h(\tau_h, a_h, r_h, o_{h+1}; f, \pi, g)]|$$
    where $\mathcal{D}_h^t$ is the empirical approximation by executing $a_{1:M(h)-1} \sim \pi^t, a_{M(h):h} \sim \pi^e(\pi^t)$ and collecting $m$ i.i.d tuples.
6:     Select policy optimistically as follows
$$(\pi^{t+1}, g^{t+1}) := \operatorname*{argmax}_{\pi \in \Pi, g \in \mathcal{G}} \mathbb{E}_{\mathcal{D}^0}[g_1(o)] \quad \text{s.t.} \quad \forall h \in [H], \forall i \in [t], \sigma_h^i(\pi, g) \leq R.$$
7: **end for**
8: **Output:** Randomly choose $\hat{\pi}$ from $(\pi_1, \cdots, \pi_T)$.

---

$B_W$ for any $h \in [H]$.

By setting $T = 2Hd \ln \left( 4Hd \left( \frac{B_X^2 B_W^2}{\zeta^2(\tilde{\epsilon}_{gen})} + 1 \right) \right), R = \epsilon_{gen}$ where

$$\epsilon_{gen} := \max_h \epsilon_{gen,h}(m, \Pi, \mathcal{G}, \mathcal{F}, \delta/(TH+1)), \tilde{\epsilon}_{gen} := \max_h \epsilon_{gen,h}(m, \Pi, \mathcal{G}, \mathcal{F}, \delta/H).$$

*With probability at least* $1 - \delta$, *letting* $\pi^\star = \operatorname{argmax}_{\pi \in \Pi} J(\pi^\star)$, *we have*

$$J(\pi^\star) - J(\hat{\pi}) \leq H^{1/2} \left[ 4\zeta(\epsilon_{gen})^2 + 2T\zeta(2\epsilon_{gen})^2 Hd \ln(4Hd(B_X^2 B_W^2/\zeta^2(\tilde{\epsilon}_{gen}) + 1)) \right]^{1/2} + 2\epsilon_{ini}.$$

*The total number of samples used in the algorithm is* $mTH$.

This reduces to Theorem 1 when we set $\zeta$ as an identify function and $\pi^e(\pi') = \pi'$. When $\zeta^{-1}(\cdot)$ is a strongly convex function, we can gain more refined rate results. For example, when $\zeta(x) = \sqrt{x}$, i.e., $\zeta^{-1}(x) = x^2$, with $\epsilon_{gen} = O(1/\sqrt{m})$, the above theorem implies a slow sample complexity rate $1/\epsilon^4$. However, by leverage the strong convexity of the square function $\zeta^{-1}(x) := x^2$, a refined analysis can give the fast rate $1/\epsilon^2$. We will see such two examples in the next sections.

## G   Examples for Generalized PO-Bilinear AC Class

We demonstrate that our generalized framework captures two models: (1) $M$-step decodable POMDPs, and (2) observable POMDPs with the latent low-rank transition. In this section, we assume $r_h \in [0, 1]$ for any $h \in [H]$.

### G.1   $M$-step decodable POMDPs

The example we include here is a model that involves nonlinear function approximation but has a unique assumption on the exact identifiability of the latent states.

**Example 5** ($M$-step decodable POMDPs [19]). *There exists an unknown decoder* $\iota_h : \bar{\mathcal{Z}}_h \to \mathcal{S}$, *such that for every reachable trajectory* $(s_{1:h}, a_{1:h-1}, o_{1:h})$, *we have* $s_h = \iota_h(\bar{z}_h)$ *for all* $h \in [H]$.

Note that when $M = 0$, this model is reduced to the well-known Block MDP model [17, 51, 78].

**Existence of future-dependent value functions.**   From the definition, using a value function $V_h^\pi(z_{h-1}, s_h)$ over $z_{h-1} \in \mathcal{Z}_{h-1}, s_h \in \mathcal{S}$, we can define a future-dependent value function $v_h^\pi : \mathcal{Z}_{h-1} \times \mathcal{O} \to \mathbb{R}$ as

$$v_h^\pi(z_{h-1}, o_h) = V_h^\pi(z_{h-1}, \iota_h(\bar{z}_h))$$

since it satisfies

$$\mathbb{E}_{o_h \sim \mathbb{O}(s_h)}[v_h^\pi(z_{h-1}, o_h) \mid z_{h-1}, s_h] = \mathbb{E}_{o_h \sim \mathbb{O}(s_h)}[V_h^\pi(z_{h-1}, \iota_h(\bar{z}_h)) \mid z_{h-1}, s_h] = V_h^\pi(z_{h-1}, s_h).$$

This is summarized in the following lemma.

**Lemma 16** (Existence of future-dependent value functions in $M$-step decodable POMDPs). *In $M$-step decodable POMDPs, future-dependent value functions exist.*

$M$-step decodable POMDPs showcase the *generality of future-dependent value functions*, which not only capture standard observability conditions where future observations and actions are used to replace belief states (e.g., observable tabular POMDPs and observable LQG), but also capture a model where history is used to replace latent states.

**PO-Bilinear Rank.** Next, we calculate the PO-bilinear rank based on Definition 9. In the tabular case, we can naïvely obtain the PO-bilinear decomposition with rank $|\mathcal{O}|^M |\mathcal{A}|^M |\mathcal{S}|$ following Example 1. Here, we consider the nontabular case where function approximation is used and $|\mathcal{O}|$ can be extremely large. We define the following Bellman operator associated with $\pi$ at step $h$:

$$\mathcal{B}_h^\pi : \mathcal{G} \to [\bar{\mathcal{Z}}_h \to \mathbb{R}]; \tag{10}$$
$$\forall \bar{z}_h : [\mathcal{B}_h^\pi g](\bar{z}_h) := \mathbb{E}_{a_h \sim \pi(\bar{z}_h)} \left[ r_h(\iota_h(\bar{z}_h), a_h) + \mathbb{E}_{o_{h+1} \sim \mathbb{O} \circ \mathbb{T}(\iota_h(\bar{z}_h), a_h)}[g_{h+1}(\bar{z}_{h+1})] \right].$$

Note that above we use the ground truth decoder $\iota_h$ to decode from $\bar{z}_h$ to its associated latent state $s_h$. The existence of this Bellman operator $\mathcal{B}_h^\pi$ is crucially dependent on the existence of such decoder $\iota_h$.

We show that $M$-step decodable POMDPs satisfy the definition in Definition 9. We assume that the latent state-wise transition model is low-rank. In MDPs, this assumption is widely used in [76, 36, 2, 71]. Here, we do *not* need to know $\mu, \phi$ in the algorithm.

**Assumption 4** (Low-rankness of latent transition). *Suppose $\mathbb{T}$ is low-rank, i.e., $\mathbb{T}(s' \mid s, a) = \langle \phi(s, a), \mu(s') \rangle (\forall (s, a, s'))$ where $\phi, \mu$ are (unknown) $d$-dimensional features. As technical conditions, we suppose $\|\phi(s, a)\| \leq 1$ for any $(s, a)$ and $|\int \mu(s) v(s) d(s)| \leq \sqrt{d}$ for any $\|v\|_\infty \leq 1$.*

**Lemma 17** (Bilinear decomposition of low-rank $M$-step decodable POMDPs). *Suppose Assumption 4, $\|\mathcal{G}_h\|_\infty \leq H$, $\|\mathcal{F}_h\|_\infty \leq H$, $r_h \in [0, 1]$ for any $h \in [H]$. Assume a discriminator class is Bellman complete, i.e.,*

$$\forall \pi \in \Pi, \forall g \in \mathcal{G} : (\mathcal{B}_h^\pi g) - g_h \in \mathcal{F}_h,$$

*for any $h \in [H]$. The loss function is designed as*

$$l_h(\tau_h, a_h, r_h, o_{h+1}; f, \pi, g) := \pi_h(a_h \mid \bar{z}_h)|\mathcal{A}|f(\bar{z}_h)(g_h(\bar{z}_h) - r_h - g_{h+1}(\bar{z}_{h+1})) - 0.5f(\bar{z}_h)^2. \tag{11}$$

*Then, there exist $W_h(\pi, g), X_h(\pi')$ so that the PO-bilinear rank is at most $d$ such that*

$$|\mathbb{E}[g_h(\bar{z}_h) - r_h - g_{h+1}(\bar{z}_{h+1}) : a_{1:h} \sim \pi]| = |\langle W_h(\pi, g), X_h(\pi) \rangle|, \tag{12}$$

$$\left| \max_{f \in \mathcal{F}_h} \mathbb{E}[l_h(\tau_h, a_h, r_h, o_{h+1}; f, \pi, g); a_{1:M(h)-1} \sim \pi', a_{M(h):h} \sim \mathcal{U}(\mathcal{A})] \right| \geq \frac{0.5 \langle W_h(\pi, g), X_h(\pi') \rangle^2}{|\mathcal{A}|^M}, \tag{13}$$

*and*

$$\left| \max_{f \in \mathcal{F}_h} \mathbb{E}[l_h(\tau_h, a_h, r_h, o_{h+1}; f, \pi, g^\pi); a_{1:M(h)-1} \sim \pi', a_{M(h):h} \sim \mathcal{U}(\mathcal{A})] \right| = 0. \tag{14}$$

*Proof.* The proof is deferred to Section Q.2. Note that (12), (13), (14) correspond to (a), (b), (c) in Definition 9. $\square$

We use the most general bilinear class definition from Definition 9, where $\zeta(a) = |\mathcal{A}|^{M/2} a^{1/2}$ for scalar $a \in \mathbb{R}^+$. Hence $\zeta$ is a non-decreasing function ($\zeta$ is non-decreasing in $\mathbb{R}^+$). The proof of the above lemma leverages the novel trick of the so-called moment matching policy introduced by [19]. When the latent state and action space are discrete, it states that the bilinear rank is $|\mathcal{S}||\mathcal{A}|$, which is much smaller than $|\mathcal{O}|^M |\mathcal{A}|^M |\mathcal{S}|$. Note we here introduce $-0.5f(\bar{z}_h)^2$ in the loss function (11) to induce strong convexity w.r.t $f$ as in [70, 14, 10], which is important to obtain the fast rate later.

The concrete sample complexity of PROVABLE (Algorithm 3) for this model is summarized in the following. Recall that the bilinear rank is $d$ where $d$ is the rank of the transition matrix. We set $\mathcal{G}_h \subset [\bar{\mathcal{Z}}_h \to [0, H]]$. Then, we have the following result.

**Theorem 9** (Sample complexity for $M$-step decodable POMDPs (Informal)). *Suppose Assumption 4, Bellman completeness, $\|\mathcal{G}_h\|_\infty \leq H, \|\mathcal{F}_h\|_\infty \leq H, r_h \in [0,1]$ for any $h \in [H]$. With probability $1 - \delta$, we can achieve $J(\pi^\star) - J(\hat{\pi}) \leq \epsilon$ when we use samples at most*

$$\tilde{O}\left(\frac{d^2 H^6 |\mathcal{A}|^{2+M}\ln(|\Pi_{\max}||\mathcal{F}_{\max}||\mathcal{G}_{\max}|/\delta)}{\epsilon^2}\right).$$

*Here,* $\mathrm{polylog}(d, H, |\mathcal{A}|, 1/\epsilon, \ln(|\Pi_{\max}|), \ln(|\mathcal{F}_{\max}|), \ln(|\mathcal{G}_{\max}|), \ln(1/\delta))$ *are omitted.*

The followings are several implications. First, the error rate scales with $O(1/\epsilon^2)$. As we promised, by leveraging the strong convexity of loss functions, we obtain a rate $O(1/\epsilon^2)$, which is faster than $O(1/\epsilon^4)$ that are attained when we naively invoke Theorem 8 with $\xi(x) \propto \sqrt{x}$. Secondly, the error bound incurs $|\mathcal{A}|^M$. As showed in [19], this is inevitable in $M$-step decodable POMDPs. Thirdly, in the tabular case, when we use the naïve function classes for $\mathcal{G}, \mathcal{F}, \Pi$, i.e., $\mathcal{G}_h = \{\bar{\mathcal{Z}}_h \to [0, H]\}$, $\mathcal{F}_h = \{\bar{\mathcal{Z}}_h \to [0, H]\}, \Pi_h = \{\bar{\mathcal{Z}}_h \to \Delta(\mathcal{A})\}$, the bound could incur additional $|\mathcal{O}|^M$ since the complexity of the function classes can scale with respect to $(|\mathcal{O}||\mathcal{A}|)^M$ (e.g., $\log(|\mathcal{G}_h|)$ can be in the order of $O(|\mathcal{O}|^M|\mathcal{A}^M)$, and similarly for $\log(|\mathcal{F}_h|), \ln(\Pi_h))$. However, when we start form a realizable model class that captures the ground truth transition and omission distribution, we can remove $|\mathcal{O}|^M$. See Section G.2.4 for an example.

Note that [19] uses a different function class setup where they assume one has an M memory-action dependent $Q$ function class $\mathcal{Q}_h : \bar{\mathcal{Z}}_h \times \mathcal{A} \to \mathbb{R}$ which contains $Q_h^\star(\bar{z}_h, a)$ while we use the actor-critic framework $v_h^\pi, \pi$. The two function class setups are not directly comparable. Generally, we mention that such optimal $Q^\star$ with truncated history does not exist when the exact decodability does not hold (e.g., such $Q^\star$ with truncated history does not exist in LQG). This displays the potential generality of the actor-critic framework we propose here.

## G.2   Observable POMDPs with Latent Low-rank Transition: a model-based perspective

The final example we include in this work is a POMDP with the latent low-rank transition. We first introduce the model, and then we introduce our function approximation setup and show the sample complexity. Finally, we revisit the sample complexity for observable tabular POMDPs and $M$-step decodable tabular POMDPs using the improved algorithm that elaborates on the model-based approach in this section.

**Example 6** (Observable POMDPs with latent low-rank transition). *The latent transition $\mathbb{T}(s'|s, a)$ is factorized as $\mathbb{T}(s'|s, a) = \mu^\star(s')^\top \phi^\star(s, a), \forall s, a, s'$ where $\mu^\star : \mathcal{S} \to \mathbb{R}^d$ and $\phi^\star : \mathcal{S} \times \mathcal{A} \to \mathbb{R}^d$. The observation $|\mathcal{O}| \times |\mathcal{S}|$ matrix $\mathbb{O}$ has full-column rank.*

In the tabular POMDP example, we have $d \leq |\mathcal{S}|$. However in general $d$ can be much smaller than $|\mathcal{S}|$. Note that in this section, we will focus on the setting where $\mathcal{S}, \mathcal{O}$ are discrete to avoid using measure theory languages, but their size could be extremely large. Particularly, our sample complexity will not have explicit polynomial or logarithmic dependence on $|\mathcal{O}|, |\mathcal{S}|$, instead it will only scale polynomially with respect to the complexity of the hypothesis class and the rank $d$.

**Model-based function approximation.**   Our function approximation class consists of a set of models $\mathcal{M} = \{(\mu, \phi, O)\}$ where $\mu, \phi$ together models latent transition as $\mu(\cdot)^\top \phi(s, a) \in \Delta(\mathcal{S})$, and $O : \mathcal{S} \to \Delta(\mathcal{O})$ models $\mathbb{O}$, and $O$ is full column rank. For notation simplicity, we often use $\theta := (\mu, \phi, O) \in \mathcal{M}$ to denote a model $(\mu, \phi, O)$. We impose the following assumption.

**Assumption 5** (Realizability). *We assume realizability, i.e., $(\mu^\star, \phi^\star, \mathbb{O}) \in \mathcal{M}$.*

We assume $\mathcal{M}$ is discrete, but $|\mathcal{M}|$ can be large such that a linear dependence on $|\mathcal{M}|$ in the sample complexity is not acceptable. Our goal is to get a bound that scales polynomially with respect to $\ln(|\mathcal{M}|)$, which is the standard statistical complexity of the discrete hypothesis class $\mathcal{M}$.

Next, we construct $\Pi, \mathcal{G}, \mathcal{F}$ using the model class $\mathcal{M}$. Given $\theta := (\mu, \phi, O)$, we denote $\pi^\theta$ as the optimal $M$-memory policy, i.e., the $M$-memory policy that maximizes the total expected reward. We set

$$\Pi = \{\pi^\theta : \theta \in \mathcal{M}\}.$$

We consider the value function class for $\theta := (\mu, \phi, O)$ with $O$ being full column rank. For each $\theta$, we can define the corresponding value function of the policy $\pi$ at $h \in [H]$: $V_{\theta;h}^\pi(z_{h-1}, s_h)$ :

$\mathcal{Z}_{h-1} \times \mathcal{S} \to \mathbb{R}$. Then, since $O$ is full column rank, as we see in the proof of Lemma 1, a corresponding future-dependent value function is

$$g_{\theta;h}^{\pi}(z,o) = \langle f_{\theta,h}^{\pi}, \mathbf{1}(z) \otimes \mathbb{O}^{\dagger}\mathbf{1}(o)\rangle$$

where $V_{\theta;h}^{\pi}(z_{h-1}, s_h) = \langle f_{\theta;h}^{\pi}, \mathbf{1}(z) \otimes \mathbf{1}(s)\rangle$. Then, we construct $\mathcal{G} = \{\mathcal{G}_h\}$ as:

$$\forall h \in [H]: \mathcal{G}_h = \{\bar{\mathcal{Z}}_h \ni \bar{z}_{h-1} \mapsto g_{\theta;h}^{\pi}(\bar{z}_{h-1}) \in \mathbb{R} : \pi \in \Pi, \theta \in \mathcal{M}\}. \tag{15}$$

By construction, since $\theta^{\star} := (\mu^{\star}, \phi^{\star}, \mathbb{O}) \in \mathcal{M}$, we must have $g^{\pi} \in \mathcal{G}, \forall \pi \in \Pi$, which implies $\mathcal{G}$ is realizable (note $g_h^{\pi} = g_{\theta^{\star};h}^{\pi}$). Here, from the construction and the assumption $r_h \in [0,1]$ for any $h \in [H]$, we have $|\mathcal{G}_h| \le |\mathcal{M}|^2$ and $\|\mathcal{G}_h\|_{\infty} \le H/\sigma_1$, which can be seen from

$$\forall (z,o); \langle f_{\theta;h}^{\pi}, \mathbf{1}(z) \otimes O^{\dagger}\mathbf{1}(o)\rangle \le \|f_{\theta;h}^{\pi}\|_{\infty}\|\mathbf{1}(z) \otimes O^{\dagger}\mathbf{1}(o)\|_1 \le H \times \|O^{\dagger}\mathbf{1}(o)\|_1 \le H/\sigma_1$$

by assuming $\|O^{\dagger}\|_1 \le 1/\sigma_1$ and $\|f_{\theta;h}^{\pi}\|_{\infty} \le H$.

To construct a discriminator class $\mathcal{F}$, we first define the Bellman operator $\mathcal{B}_{\theta;h}^{\pi}$ for $\pi \in \Pi, h \in [H], \theta \in \mathcal{M}$:

$$\mathcal{B}_{\theta;h}^{\pi} : \mathcal{G} \to [\mathcal{H}_h \to \mathbb{R}];$$

$$\forall \tau_h; \left(\mathcal{B}_{\theta;h}^{\pi}g\right)(\tau_h) = \mathbb{E}_{a_h \sim \pi_h(\bar{z}_h)}\left[r_h + \mathbb{E}_{o_{h+1} \sim \mathbb{P}_{\theta}(\cdot|\tau_h, a_h)}g_{h+1}(\bar{z}_{h+1})\right],$$

where $\mathcal{H}_h$ is the whole history space up to $h$ ($\tau_h = (a_{1:h-1}, o_{1:h})$, and $\bar{z}_h$ is just part of this history) and $\mathbb{P}_{\theta}(o_{h+1}|\tau_h, a_h)$ is the probability of generating $o_{h+1}$ conditioned on $\tau_h, a_h$ under model $\theta$. Then, we construct $\mathcal{F} = \{\mathcal{F}_h\}$ such that

$$\forall h \in [H]: \mathcal{F}_h = \{\mathcal{H}_h \ni \tau_h \mapsto \{g_h - \mathcal{B}_{\theta;h}^{\pi}g\}(\tau_h) \in \mathbb{R} : \pi \in \Pi, g \in \mathcal{G}, \theta \in \mathcal{M}\}. \tag{16}$$

so that we can ensure the Bellman completeness:

$$-(\mathcal{B}_h^{\pi}\mathcal{G}) + \mathcal{G}_h \subset \mathcal{F}_h.$$

noting $\mathcal{B}_{\theta^{\star};h}^{\pi} = \mathcal{B}_h^{\pi}$. Here, from the construction, $|\mathcal{F}_h| \le |\mathcal{M}|^2 \times |\mathcal{M}|^2 \times |\mathcal{M}|^2 = |\mathcal{M}|^6$ and $\|\mathcal{F}_h\|_{\infty} \le 3H/\sigma_1$.

We define the loss as the same as the one we used in $M$-step decodable POMDPs, except that our discriminators now take the entire history as input:

$$l_h(\tau_h, a_h, r_h, o_{h+1}; f, \pi, g) := \pi_h(a_h \mid \bar{z}_h)|\mathcal{A}|f(\tau_h)(g_h(\bar{z}_h) - r_h - g_{h+1}(\bar{z}_{h+1})) - 0.5f(\tau_h)^2. \tag{17}$$

Finally, as in the case of $M$-step decodable POMDPs (Lemma 17), we get the following lemma that states that our model is a PO-bilinear AC class (Definition 9) under the following model assumption.

**Assumption 6.** *We assume $\|O^{\dagger}\|_1 \le 1/\sigma_1$ for any $O$ in the model. Suppose $\mu(\cdot)^{\top}\phi(s,a) \in \Delta(\mathcal{S})$ for any $(s,a)$, $\mu(\cdot)$ and $\phi(\cdot)$ in the model. Suppose $\|\phi(s,a)\| \le 1$ for any $\phi$ in the model and $(s,a) \in \mathcal{S} \times \mathcal{A}$. Suppose for any $v : \mathcal{S} \to [0,1]$ and for any $\mu$ in the model, we have $\|\int v(s)\mu(s)\mathrm{d}(s)\|_2 \le \sqrt{d}$.*

**Lemma 18** (PO-bilinear decomposition for Observable POMDPs with low-rank transition)**.** *Suppose Assumption 5, 6. Consider observable POMDPs with latent low-rank transition. Set $\mathcal{G}$ as in (15), $\mathcal{F}$ as in (16) and $l$ as in (17). Then, there exist $W_h(\pi, g), X_h(\pi')$ that admits the PO-bilinear rank decomposition in Definition 9 with rank $d$.*

The above lemma ensures that the PO-bilinear rank only depends on $d$, and is independent of the length of the memory. For example, in the tabular case, it is $|\mathcal{S}|$.

Next, we show the output from DISPROVABLEcan search for the best in class $M$-memory policy as follows.

**Theorem 10** (Sample complexity of DISPROVABLE for observable POMDPs with latent low-rank transition)**.** *Consider observable POMDPs with latent low-rank transition. Suppose Assumption 5, 6. With probability $1 - \delta$, we can achieve $J(\pi^{\star}) - J(\hat{\pi}) \le \epsilon$ when we use samples at most*

$$\tilde{O}\left(\frac{d^2 H^6 |\mathcal{A}|^{2+M} \ln(|\mathcal{M}|/\delta)}{\epsilon^2 \sigma_1^2}\right).$$

*Here, we omit* $\mathrm{polylog}(d, H, |\mathcal{A}|, \ln(1/\delta), \ln(|\mathcal{M}|), 1/\sigma_1, 1/\epsilon)$.

Here, we emphasize that there is no explicit polynomial or logarithmic dependence on $|\mathcal{S}|$ and $|\mathcal{O}|$, which permits learning for large state and observation spaces. We also do not have any explicit polynomial dependence on $|\mathcal{O}|^M$, as we construct $\Pi$ and $\mathcal{G}$ from the model class $\mathcal{M}$ which ensures the complexities of $\pi$ and $\mathcal{G}$ are in the same order as that of $\mathcal{M}$.

### G.2.1 Global Optimality

We show a quasi-polynomial sample complexity bound for competing against the globally optimal policy $\pi_{\text{gl}}^\star$. To compete against the globally optimal policy $\pi_{\text{gl}}^\star$, we need to set $M$ properly. We use the following lemma. The proof is given in Section R.

**Lemma 19** (Near global optimaltiy of $M$-memoruy policy). *Consider $\epsilon \in (0, H]$, and a POMDP with low-rank latent transition and $\mathbb{O}$ being full column rank with $\|\mathbb{O}^\dagger\|_1 \le 1/\sigma_1$. When $M = \Theta(C(1/\sigma_1)^{-4} \ln(dH/\epsilon))$ (with $C$ being some absolute constant), there must exists an $M$-memory policy $\pi^\star$, such that $J(\pi_{\text{gl}}^\star) - J(\pi^\star) \le \epsilon$*

Note that the memory $M$ above is independent of $|\mathcal{S}|$ instead it only depends on the rank $d$. To prove the above lemma, we first show a new result on belief contraction for low-rank POMDPs under the $\ell_1$-based observability. The proof of the belief contraction borrows some key lemma from [23] but extends the original result for small-size tabular POMDPs to low-rank POMDPs. We leverage the linear structure of the problem and the G-optimal design to construct an initial distribution over $\mathcal{S}$ that can be used as a starting point for belief propagation along the memory.

We conclude the study on the POMDPs with low-rank latent transition by the following theorem, which demonstrates a quasi-polynomial sample complexity for learning the globally optimal policy.

**Theorem 11** (Sample complexity of DISPROVABLE for POMDPs with low-rank latent transition — competing against $\pi_{\text{gl}}^\star$). *Consider observable POMDPs with latent low-rank transition. Fix some $\epsilon \in (0, H), \delta \in (0, 1)$. Suppose Assumption 5, 6. We construct $\Pi, \mathcal{G}, \mathcal{F}$, and the loss $l$ as we described above. With probability at least $1 - \delta$, when $M = \Theta(C\sigma_1^{-4} \ln(dH/\epsilon))$, DISPROVABLE outputs a $\hat{\pi}$ such that $J(\pi_{\text{gl}}^\star) - J(\hat{\pi}) \le \epsilon$, with number of samples scaling*

$$\tilde{O}\left(\frac{d^2 H^6 |\mathcal{A}|^2 \ln(|\mathcal{M}|/\delta)}{\epsilon^2 \sigma_1^2} \cdot |\mathcal{A}|^{\ln(dH/\epsilon)/\sigma_1^4}\right).$$

**Remark 6** (Comparison to [73]). *We compare our results to the very recent work [73] that studies POMDPs with the low-rank latent transition. The results are in general not directly comparable, but we state several key differences here. First, [73] considers a special instance of low-rank transition, i.e., [73] assumes $\mathbb{T}$ has low non-negative rank, which could be exponentially larger than the usual rank [2]. Second, [73] additionally assumes short past sufficiency, a condition which intuitively says that for any roll-in policy, the sufficient statistics of a short memory is enough to recover the belief over the latent states, and their sample complexity has an exponential dependence on the length of the memory. While our result also relies on the fact that the globally optimal policy can be approximated by an $M$-memory policy with small $M$, this fact is derived directly from the standard observability condition.*

### G.2.2 Revisiting Observable Undercomplete Tabular POMDPs

We reconsider the sample complexity of undercomplete tabular POMDPs using Theorem 10. In this case, we will start from a model class that captures the ground truth latent transition $\mathbb{T}$ and omission distribution $\mathbb{O}$. By constructing $\epsilon$-nets over the model class,we can set $\ln(|\mathcal{M}|) = \tilde{O}(|\mathcal{S}|^3 |\mathcal{O}||\mathcal{A}|)$ since $\mathbb{T}, \mathbb{O}$ have $|\mathcal{S}|^2|\mathcal{A}|$ and $|\mathcal{O}||\mathcal{S}|$ many parameters, respectively. Besides, the PO-bilinear rank is $d = |\mathcal{S}|$. Therefore, the sample complexity is

$$\tilde{O}\left(\frac{|\mathcal{S}|^5 |\mathcal{O}| H^6 |\mathcal{A}|^{2+M} \ln(1/\delta)}{\epsilon^2 \sigma_1^2}\right).$$

We leave the formal analysis to future works.

Compared to results in Section 5.3.2, there is no $|\mathcal{O}|^M$ term. This is due to two improvements. The first improvement is that we refine the rank from $|\mathcal{O}|^M|\mathcal{A}|^M|\mathcal{S}|$ to $|\mathcal{S}|$. The second improvement is we model the future-dependent value function class and policy class starting from the model class whose complexity has nothing to do with the length of memory $M$ (note that previously, from a pure model-free perspective, the statistical complexity of $\mathcal{G}$ can scale as $|\mathcal{O}|^M|\mathcal{A}|^M|\mathcal{S}|$ in the worst case).

### G.2.3 Revisiting Observable Overcomplete POMDPs

We reconsider the sample complexity of overcomplete tabular POMDPs using Theorem 10 with slight modification to incorporate multi-step future. Suppose $\|\{\mathbb{O}^K\}^\dagger\|_1 \le 1/\sigma_1$ (recall $\mathbb{O}^K$ is defined in

Lemma 5 in Section C.2). Then, we can achieve a sample complexity

$$\tilde{O}\left(\frac{|\mathcal{S}|^5|\mathcal{O}|H^6|\mathcal{A}|^{2+M}\ln(1/\delta)}{\epsilon^2\sigma_1^2}\right)$$

since the PO-bilinear rank is $|\mathcal{S}|$. Note that there is no $|\mathcal{O}|^{M+K}$ dependence, since both the policy class and the future-dependent value function class are built from the model class whose complexity has nothing to do with $M, K$.

Note that due to our definition of $\mathbb{O}^K$, there is no $|\mathcal{A}|^K$ term. However, when we use a different definition, for instance, $\min_{a'_{h:h+K-2}\in\mathcal{A}^{K-1}}\|\{\mathbb{O}^K(a'_{h:h+K-2})\}^\dagger\|_1 \le 1/\alpha_1$ (recall $\mathbb{O}^K(a'_{h:h+K-2})$ is defined in Section C.2), we would incur $|\mathcal{A}|^K$. This is because if we only know that there is an unknown sequence of actions $a'_{h:h+K-2}$ such that $\mathbb{O}^K(a'_{h:h+K-2})$ is full column rank, we need to use uniform samples $|\mathcal{A}|^K$ in the importance sampling step to identify such a sequence. More formally, we can see that

$$|\mathcal{A}|^K \min_{a'_{h:h+K-2}\in\mathcal{A}^{K-1}} \|\{\mathbb{O}^K(a'_{h:h+K-2})\}^\dagger\|_1 \ge \|\{\mathbb{O}^K\}^\dagger\|_1. \tag{18}$$

### G.2.4 Revisiting $M$-step Decodable Tabular POMDPs

We reconsider the sample complexity of tabular $M$-step decodable POMDPs by constructing $\mathcal{F}, \mathcal{G}, \Pi$ from the model class $\mathcal{M}$ as we did for the low-rank POMDP. In this case, by constructing $\epsilon$-nets, we can set $\ln(|\mathcal{M}|) = \tilde{O}(|\mathcal{S}|^3|\mathcal{O}||\mathcal{A}|)$ since $\mathbb{T}, \mathbb{O}$ have $|\mathcal{S}|^2|\mathcal{A}|$ and $|\mathcal{O}||\mathcal{S}|$ parameters, respectively. Therefore, the sample complexity is

$$\tilde{O}\left(\frac{H^6|\mathcal{S}|^5|\mathcal{O}||\mathcal{A}|^{2+M}\ln(1/\delta)}{\epsilon^2}\right).$$

Again, we leave the formal analysis to future works. Compared to the naive result mentioned after Theorem 9 where $\ln(\mathcal{G}), \ln(\Pi)$ could scale in the order of $|\mathcal{O}|^M$ in the tabular case, we do not have $|\mathcal{O}|^M$ dependence here.

## H  Proof of Theorem 1

We fix the parameters as in Theorem 1. Let

$$l_h(\bar{z}_h, a_h, r_h, o_{h+1}) = |\mathcal{A}|\pi_h(a_h \mid \bar{z}_h)\{r_h + g_{h+1}(\bar{z}_{h+1})\} - g_h(\bar{z}_h).$$

We define

$$\epsilon_{gen} = \max_h \epsilon_{gen,h}(m, \Pi, \mathcal{G}, \delta/(TH+1)), \quad \epsilon_{ini} = \epsilon_{ini}(\mathcal{G}, \delta/(TH+1)),$$
$$\tilde{\epsilon}_{gen} = \max_h \epsilon_{gen}(m, \Pi, \mathcal{G}, \delta/H).$$

Then, by our assumption 2 with probability $1 - \delta$, we $\forall t \in [T], \forall h \in [H]$

$$\sup_{\pi\in\Pi, g\in\mathcal{G}} |\mathbb{E}_{\mathcal{D}_h^t}[l_h(\bar{z}_h, a_h, r_h, o_{h+1}; \pi, g)] - \mathbb{E}[\mathbb{E}_{\mathcal{D}_h^t}[l_h(\bar{z}_h, a_h, r_h, o_{h+1}; \pi, g)]]| \le \epsilon_{gen}, \tag{19}$$

$$\sup_{g_1\in\mathcal{G}_1} |\mathbb{E}_{\mathcal{D}^0}[g_1(o_1)] - \mathbb{E}[\mathbb{E}_{\mathcal{D}^0}[g_1(o_1)]]| \le \epsilon_{ini}. \tag{20}$$

Hereafter, we condition on the above events.

We first show the following lemma. Recall

$$\pi^\star = \operatorname*{argmax}_{\pi\in\Pi} J(\pi).$$

**Lemma 20** (Optimism). *Set $R := \epsilon_{gen}^2$. For all $t \in [T]$, $(\pi^\star, g^{\pi^\star})$ is a feasible solution of the constrained program. Furthermore, we have $J(\pi^\star) \le \mathbb{E}[g_1^t(o_1)] + 2\epsilon_{ini}$ for any $t \in [T]$, where $g^t$ is the future-dependent value function selected by the algorithm in iteration $t$.*

*Proof.* For any $\pi$, we have

$$\mathbb{E}[\mathbb{E}_{\mathcal{D}_h^t}[l_h(\bar{z}_h, a_h, r_h, o_{h+1}; \pi, g^\pi)] = 0$$

since $g^\pi$ is a future-dependent value function in $\mathcal{G}$. This is because

$$\begin{aligned}
&\mathbb{E}[\mathbb{E}_{\mathcal{D}_h^t}[l_h(\bar{z}_h, a_h, r_h, o_{h+1}; \pi, g^\pi)] \\
&= \mathbb{E}[g_h(\bar{z}_h) - r_h - g_{h+1}(\bar{z}_{h+1}); a_{1:h-1} \sim \pi^t, a_h \sim \pi] &&\text{(IS sampling)} \\
&= \langle W_h(\pi, g^\pi), X_h(\pi^t) \rangle &&\text{(First assumption in Definition 3)} \\
&= 0. &&\text{(Second assumption in Definition 3)}
\end{aligned}$$

Thus,

$$|\mathbb{E}_{\mathcal{D}_h^t}[l_h(\bar{z}_h, a_h, r_h, o_{h+1}; \pi^\star, g^{\pi^\star})]| \leq \epsilon_{gen}.$$

using (19) noting $\pi^\star \in \Pi, g^{\pi^\star} \in \mathcal{G}$. This implies

$$\forall t \in [T], \forall h \in [H]; (\mathbb{E}_{\mathcal{D}_h^t}[l_h(\bar{z}_h, a_h, r_h, o_{h+1}; \pi^\star, g^{\pi^\star})])^2 \leq \epsilon_{gen}^2.$$

Hence, $(\pi^\star, g^{\pi^\star})$ is a feasible set for any $t \in [T]$.

Then, we have

$$\begin{aligned}
J(\pi^\star) = \mathbb{E}[g_1^{\pi^\star}(o_1)] &\leq \mathbb{E}_{\mathcal{D}^0}[g_1^{\pi^\star}(o_1)] + \epsilon_{ini} &&\text{(Uniform convergence result)} \\
&\leq \mathbb{E}_{\mathcal{D}^0}[g_1^t(o_1)] + \epsilon_{ini} &&\text{(Using the construction of algorithm)} \\
&\leq \mathbb{E}[g_1^t(o_1)] + 2\epsilon_{ini}. &&\text{(Uniform convergence)}
\end{aligned}$$

$\square$

**Remark 7.** *Note that*

$$\mathbb{E}[\mathbb{E}_{\mathcal{D}_h^t}[l_h(\bar{z}_h, a_h, r_h, o_{h+1}; \pi, g^\pi)]] = 0$$

*holds for general future-dependent value functions $g^\pi$ in Definition 6 . Thus, the statement goes through even if we use Definition 6.*

Next, we prove the following lemma to upper bound the per step regret.

**Lemma 21.** *For any $t \in [T]$, we have*

$$J(\pi^\star) - J(\hat{\pi}) \leq \sum_{h=1}^{H} |\langle W_h(\pi^t, g^t), X_h(\pi^t) \rangle| + 2\epsilon_{ini}.$$

*Proof.*

$$\begin{aligned}
&J(\pi^\star) - J(\hat{\pi}) \\
&\leq 2\epsilon_{ini} + \mathbb{E}[g_1^t(o_1)] - J(\pi^t) &&\text{(From optimism)} \\
&= 2\epsilon_{ini} + \sum_{h=1}^{H} \mathbb{E}[g_h^t(\bar{z}_h) - \{r_h + g_{h+1}^t(\bar{z}_{h+1})\}; a_{1:h} \sim \pi^t] &&\text{(Performance difference lemma)} \\
&\leq 2\epsilon_{ini} + \sum_{h=1}^{H} |\mathbb{E}[g_h^t(\bar{z}_h) - \{r_h + g_{h+1}^t(\bar{z}_{h+1})\}; a_{1:h} \sim \pi^t]| \\
&= 2\epsilon_{ini} + \sum_{h=1}^{H} |\langle W_h(\pi^t, g^t), X_h(\pi^t) \rangle|. &&\text{(First assumption in Definition 3)}
\end{aligned}$$

$\square$

**Lemma 22.** *Let $\Sigma_{t,h} = \lambda I + \sum_{\tau=0}^{t-1} X_h(\pi^\tau) X_h(\pi^\tau)^\top$. We have*

$$\frac{1}{T} \sum_{t=0}^{T-1} \sum_{h=1}^{H} \|X_h(\pi^t)\|_{\Sigma_{t,h}^{-1}} \leq H\sqrt{\frac{d}{T} \ln\left(1 + \frac{TB_X^2}{d\lambda}\right)}.$$

*Proof.* We fix $h \in [H]$. Here, we have $\Sigma_{t,h} = \lambda I + \sum_{\tau=0}^{t-1} X_h(\pi^\tau) X_h(\pi^\tau)^\top$. From the elliptical potential lemma in [1, Lemma G.2], we have

$$\frac{1}{T} \sum_{t=0}^{T-1} \|X_h(\pi^t)\|_{\Sigma_{t,h}^{-1}} \leq \sqrt{\frac{1}{T} \sum_{t=0}^{T-1} \|X_h(\pi^t)\|_{\Sigma_{t,h}^{-1}}^2} \leq \sqrt{\frac{1}{T} \ln \frac{\det(\Sigma_{t,h})}{\det(\lambda I)}} \leq \sqrt{\frac{d}{T} \ln \left(1 + \frac{TB_X^2}{d\lambda}\right)}.$$

Then,

$$\frac{1}{T} \sum_{t=0}^{T-1} \sum_{h=0}^{H} \|X_h(\pi^t)\|_{\Sigma_{t,h}^{-1}}^2 \leq H \sqrt{\frac{d}{T} \ln \left(1 + \frac{TB_X^2}{d\lambda}\right)}.$$

$\square$

**Lemma 23.**

$$\|W_h(\pi^t, g^t)\|_{\Sigma_{t,h}}^2 \leq 2\lambda B_W^2 + 4T\epsilon_{gen}^2.$$

*Proof.* We have

$$\|W_h(\pi^t, g^t)\|_{\Sigma_{t,h}}^2 = \lambda \|W_h(\pi^t, g^t)\|_2^2 + \sum_{\tau=0}^{t-1} \langle W_h(\pi^t, g^t), X_h(\pi^\tau)\rangle^2.$$

The first term is upper-bounded by $\lambda B_W^2$. The second term is upper-bounded by

$$\sum_{\tau=0}^{t-1} \langle W_h(\pi^t, g^t), X_h(\pi^\tau)\rangle^2$$
$$= \sum_{\tau=0}^{t-1} \left(\mathbb{E}[l_h(\bar{z}_h, a_h, r_h, o_{h+1}; \pi^t, g^t); a_{1:h-1} \sim \pi^\tau, a_h \sim U(\mathcal{A})]\right)^2$$

(First assumption in Definition 3)

$$\leq 2 \sum_{\tau=0}^{t-1} \mathbb{E}_{\mathcal{D}_h^\tau}[l_h(\bar{z}_h, a_h, r_h, o_{h+1}; \pi^t, g^t)]^2 + 2t\epsilon_{gen}^2 \leq 4T\epsilon_{gen}^2.$$

From the first line to the second line, we use the definition of bilinear rank models. From the second line to the third line, we use $(a+b)^2 \leq 2a^2 + 2b^2$. In the last line, we use the constraint on $(\pi^t, g^t)$.

$\square$

Combining lemmas so far, we have

$$J(\pi^\star) - J(\hat{\pi}) \leq \frac{1}{T} \sum_{t=0}^{T-1} \sum_{h=1}^{H} |\langle W_h(\pi^t, g^t), X_h(\pi^t)\rangle| + 2\epsilon_{ini} \qquad \text{(Use Lemma 21)}$$

$$\leq \frac{1}{T} \sum_{t=0}^{T-1} \sum_{h=1}^{H} \|W_h(\pi^t, g^t)\|_{\Sigma_{t,h}} \|X_h(\pi^t)\|_{\Sigma_{t,h}^{-1}} + 2\epsilon_{ini} \qquad \text{(CS inequality)}$$

$$\leq H^{1/2} \left[2\lambda B_W^2 + 4T\epsilon_{gen}^2\right]^{1/2} \left(\frac{dH}{T} \ln \left(1 + \frac{TB_X^2}{d\lambda}\right)\right)^{1/2} + 2\epsilon_{ini}.$$

(Use Lemma 22 and Lemma 23)

We set $\lambda$ such that $B_X^2/\lambda = B_W^2 B_X^2/\epsilon_{gen}^2 + 1$ and $T = \lceil 2Hd \ln(4Hd(B_X^2 B_W^2/\tilde{\epsilon}_{gen} + 1)) \rceil$. Then,

$$\frac{Hd}{T} \ln \left(1 + \frac{TB_X^2}{d\lambda}\right) \leq \frac{Hd}{T} \ln \left(1 + \frac{T}{d} \left(\frac{B_W^2 B_X^2}{\epsilon_{gen}^2} + 1\right)\right)$$
$$\leq \frac{Hd}{T} \ln \left(1 + \frac{T}{d} \left(\frac{B_W^2 B_X^2}{\tilde{\epsilon}_{gen}^2} + 1\right)\right)$$
$$\leq \frac{Hd}{T} \ln \left(\frac{2T}{d} \left(\frac{B_W^2 B_X^2}{\tilde{\epsilon}_{gen}^2} + 1\right)\right) \leq 1$$

since $a \ln(bT)/T \leq 1$ when $T = 2a \ln(2ab)$.

Finally, the following holds

$$
\begin{aligned}
J(\pi^\star) - J(\pi^T) &\leq H^{1/2} \left[ 4\lambda B_W^2 + 8T\epsilon_{gen}^2 \right]^{1/2} + 2\epsilon_{ini} \\
&\leq H^{1/2} \left[ 4\lambda B_W^2 + 16\epsilon_{gen}^2 Hd \ln(4Hd(B_X^2 B_W^2/\tilde{\epsilon}_{gen} + 1)) \right]^{1/2} + 2\epsilon_{ini} \\
&\qquad\qquad\qquad\qquad\qquad\qquad\qquad\qquad\qquad\qquad\qquad\qquad \text{(Plug in } T) \\
&\leq H^{1/2} \left[ 8\epsilon_{gen}^2 + 16\epsilon_{gen}^2 Hd \ln(4Hd(B_X^2 B_W^2/\tilde{\epsilon}_{gen} + 1)) \right]^{1/2} + 2\epsilon_{ini} \\
&\qquad\qquad\qquad\qquad\qquad\qquad\qquad\qquad\qquad\qquad\qquad\qquad \text{(Plug in } \epsilon_{gen}) \\
&\leq 5\epsilon_{gen} \left[ H^2 d \ln(4Hd(B_X^2 B_W^2/\tilde{\epsilon}_{gen} + 1)) \right]^{1/2} + 2\epsilon_{ini}.
\end{aligned}
$$

## I  Sample Complexity for Finite Function Classes

Consider cases where $\Pi$ and $\mathcal{G}$ are finite and the PO-bilinear rank assumption is satisfied. When $\Pi$ and $\mathcal{G}$ are infinite hypothesis classes, $|\mathcal{F}|$ and $|\mathcal{G}|$ are replaced with their $L^\infty$-covering numbers, respectively.

**Theorem 12** (Sample complexity for discrete $\Pi$ and $\mathcal{G}$). *Let $\|\mathcal{G}_h\|_\infty \leq C_{\mathcal{G}}, r_h \in [0, 1]$ for any $h \in [H]$ and the PO-bilinear rank assumption holds with PO-bilinear rank $d$. By letting $|\Pi_{\max}| = \max_h |\Pi_h|, |\mathcal{G}_{\max}| = \max_h |\mathcal{G}_h|$, with probability $1 - \delta$, we can achieve $J(\pi^\star) - J(\hat{\pi}) \leq \epsilon$ when we use samples at most*

$$
\tilde{O}\left( d_b H^4 \max(C_{\mathcal{G}}, 1)^2 |\mathcal{A}|^2 \ln(|\mathcal{G}_{\max}||\Pi_{\max}|/\delta) \ln^2(B_X B_W)(1/\epsilon)^2 \right).
$$

*Here,* $\text{polylog}(d, H, |\mathcal{A}|, \ln(|\mathcal{G}_{\max}|), \ln(|\Pi_{\max}|), \ln(1/\delta), \ln(B_X), \ln(B_W), \ln(1/\delta), (1/\epsilon))$ *are omitted.*

**Proof.** We derive the above result. First, we check the uniform convergence result. Then,

$$
\epsilon_{gen} = c \max(C_{\mathcal{G}}, 1) |\mathcal{A}| \sqrt{\ln(|\mathcal{G}_{\max}||\Pi_{\max}|TH/\delta)/m}.
$$

Thus, we need to set $m$ such that

$$
J(\pi^\star) - J(\hat{\pi}) \leq c \max(C_{\mathcal{G}}, 1) |\mathcal{A}| \sqrt{\ln(|\mathcal{G}_{\max}||\Pi_{\max}|TH/\delta)/m} \sqrt{dH^2 \ln(H^3 dB_X^2 B_W^2 m + 1)} \leq \epsilon
$$

where $c$ is some constant and

$$
T = cHd \ln(HdB_X^2 B_W^2 m + 1).
$$

By organizing the term, the following $m$ is sufficient

$$
c\sqrt{\frac{dH^2 \max(C_{\mathcal{G}}, 1)^2 |\mathcal{A}|^2 \ln(|\mathcal{G}_{\max}||\Pi_{\max}|H^2 d/\delta) \ln(H^3 dB_X^2 B_W^2 m)}{m}} \leq \epsilon
$$

Using Lemma 44, the following $m$ satisfies the condition:

$$
m = c\frac{B_1 (\ln B_1 B_2)^2}{\epsilon^2}, B_1 = dH^2 \max(C_{\mathcal{G}}, 1)^2 |\mathcal{A}|^2 \ln(|\mathcal{G}_{\max}||\Pi_{\max}|H^2 d/\delta), B_2 = H^3 dB_X^2 B_W^2.
$$

Combining all together, the sample complexity is $mTH$, i.e.,

$$
\tilde{O}\left( \frac{d^2 H^4 \max(C_{\mathcal{G}}, 1)^2 |\mathcal{A}|^2 \ln(|\mathcal{G}_{\max}||\Pi_{\max}|/\delta) \ln^2(B_X B_W)}{\epsilon^2} \right). \qquad \blacksquare
$$

## J  Sample Complexity in Observable HSE POMDPs

We revisit the existence of future-dependent value functions by taking the norm constraint into account. Then, we consider the PO-bilinear decomposition with certain $B_X \in \mathbb{R}$ and $B_W \in \mathbb{R}$. Next, we calculate the uniform convergence result. Finally, we show the sample complexity result.

We use the following assumptions

**Assumption 7.** *For any $h \in [H]$, the following holds:*

1. *$V_h^\pi(z_{h-1}, s) = \langle \theta_h^\pi, \phi_h(z_{h-1}, s) \rangle$.*

2. *There exists a matrix $K_h$ such that $\mathbb{E}_{o \sim \mathbb{O}(s)}[\psi_h(z_{h-1}, o)] = K_h \phi_h(z_{h-1}, s)$ (i.e., conditional embedding of the omission distribution),*

3. *$\|\phi_h(\cdot)\| \leq 1, \|\psi_h(\cdot)\| \leq 1, \|\theta_h^\pi\| \leq \Theta_V, 0 \leq r_h \leq 1$,*

4. *There exists a matrix $T_{\pi;h}$ such that $\mathbb{E}[\phi_h(z_h, s_{h+1}) \mid z_{h-1}, s_h; a_h \sim \pi] = T_{\pi;h}\phi_h(z_{h-1}, s_h)$ (i.e., conditional embedding of the transition)*

5. *$\Pi$ is finite.*

We define

$$\sigma_{\min}(K) = \min_{h \in [H]} 1/\|K_h^\dagger\|, \quad \sigma_{\max}(K) = \max_{h \in [H]} \|K_h\|, \quad \sigma_{\max}(T) = \max_{h \in [H]} \|T_{\pi:h}\|,$$

$$d_\phi = \max_{h \in [H]} d_{\phi_h}, \quad d_\psi = \max_{h \in [H]} d_{\psi_h}.$$

**Existence of future-dependent value functions.** We show future-dependent value functions exist. This is proved by noting

$$\mathbb{E}_{o \sim \mathbb{O}(s)}[\langle (K_h^\dagger)^\top \theta_h^\pi, \psi_h(\bar{z}_h) \rangle] = \langle (K_h^\dagger)^\top \theta_h^\pi, K_h\phi_h(z_{h-1}, s_h) \rangle = \langle \theta_h^\pi, \phi_h(z_{h-1}, s_h) \rangle = V_h^\pi(z_{h-1}, s_h).$$

Thus, $\langle (K_h^\dagger)^\top \theta_h^\pi, \psi_h(\bar{z}_h) \rangle$ is a future-dependent value function. The radius of the parameter space is upper-bounded by $\Theta_V/\sigma_{\min}(K)$. Hence, we set

$$\mathcal{G}_h = \{\langle \theta, \psi_h(\cdot) \rangle : \|\theta\| \leq \Theta_V/\sigma_{\min}(K)\}.$$

Then, the realizability holds.

**PO-bilinear decomposition.** Recall we derive the PO-bilinear decomposition in Section C.4. Consider a triple $(\pi', \pi, g)$ with $g_h(\cdot) = \theta_h^\top \psi_h(\cdot)$ and $g_h^\pi = \langle \theta_h^\star, \psi_h(\cdot) \rangle$, we have:

$$\mathbb{E}\left[\theta_h^\top \psi_{h+1}(\bar{z}_h) - r_h - \theta_{h+1}^\top \psi(\bar{z}_{h+1}); a_{1:h-1} \sim \pi', a_h \sim \pi\right]$$
$$= \left\langle \mathbb{E}[\phi_h(z_{h-1}, s_h); a_{1:h-1} \sim \pi'], \quad K_h^\top(\theta_h - \theta_h^\star) - T_{\pi;h}^\top K_{h+1}^\top(\theta_{h+1} - \theta_{h+1}^\star) \right\rangle,$$

which verifies the PO-bilinear structure, i.e.,

$$X_h(\pi') = \mathbb{E}[\phi_h(z_{h-1}, s_h); a_{1:h-1} \sim \pi'], \quad W_h(\pi, g) = K_h^\top(\theta_h - \theta_h^\star) - T_{\pi;h}^\top K_{h+1}^\top(\theta_{h+1} - \theta_{h+1}^\star),$$

and shows that the PO-bilinear rank is at most $d_\phi = \max_h d_{\phi_h}$. Thus, based on the above PO-bilinear decomposition, we set $\|B_X\| = 1, \|B_W\| = 2(1 + \sigma_{\max}(T))\sigma_{\max}(K)\Theta_V/\sigma_{\min}(K)$. This is because

$$\|K_h^\top(\theta_h - \theta_h^\star) - T_{\pi;h}^\top K_{h+1}^\top(\theta_{h+1} - \theta_{h+1}^\star)\|$$
$$\leq \|K_h^\top\|(\|\theta_h\| + \|\theta_h^\star\|) + \|T_{\pi;h}^\top\|\|K_{h+1}^\top\|(\|\theta_{h+1}\| + \|\theta_{h+1}^\star\|))$$
$$\leq 2(1 + \sigma_{\max}(T))\sigma_{\max}(K)\Theta_V/\sigma_{\min}(K).$$

and

$$\|\mathbb{E}[\phi_h(z_{h-1}, s_h); a_{1:h-1} \sim \pi']\| \leq \mathbb{E}[\|\phi_h(z_{h-1}, s_h)\|; a_{1:h-1} \sim \pi'] \leq 1.$$

In the above, we use Jensen's inequality.

**Uniform convergence.** To invoke Theorem 1, we show the uniform convergence result.

**Lemma 24** (Uniform convergence of loss functions). *Let $C = \Theta_V/(\sigma_{\min}(K))$. Then, with probability $1 - \delta$,*

$$\sup_{\pi \in \Pi, g \in \mathcal{G}} |\{\mathbb{E}_\mathcal{D} - \mathbb{E}\}[|\mathcal{A}|\pi_h(a_h \mid \bar{z}_h)\{g_h(\bar{z}_h) - r_h - g_{h+1}(\bar{z}_{h+1})\}]|$$

$$\leq 5|\mathcal{A}|\{1 + 2C\}\sqrt{\frac{\{2d_\psi \ln(1 + Cm) + \ln(|\Pi_{\max}|/\delta)\}}{m}}$$

*and*

$$\sup_{g_1 \in \mathcal{G}_1} |\{\mathbb{E}_\mathcal{D} - \mathbb{E}\}[g_1(\bar{z}_1)]| \leq 5C\sqrt{\frac{\{d_\psi \ln(1 + Cm) + \ln(|\Pi_{\max}|/\delta)\}}{m}}.$$

*Proof.* Let $C = \Theta_V/\sigma_{\min}(K)$. Define $\mathcal{N}_{\epsilon,h}$ as an $\epsilon$-net for $\mathcal{G}_h$. Then, $|\mathcal{N}_{\epsilon,h}| \leq (1 + C/\epsilon)^d$. Then,

$$|l_h(\cdot; \pi, g) - l_h(\cdot; \pi^\diamond, g^\diamond)| \leq |\mathcal{A}|\{\|g_h - g_h^\diamond\|_\infty + \|g_{h+1} - g_{h+1}^\diamond\|_\infty\}$$
$$\leq |\mathcal{A}|\{\|\theta_h - \theta_h^\diamond\|_2 + \|\theta_{h+1} - \theta_{h+1}^\diamond\|_2\} \leq 2|\mathcal{A}|\epsilon.$$

Besides, for fixed $\pi \in \Pi, \theta_h \in \mathcal{N}_{\epsilon,h}, \theta_{h+1} \in \mathcal{N}_{\epsilon,h+1}$, we have

$$|\{\mathbb{E}_\mathcal{D} - \mathbb{E}\}[|\mathcal{A}|\pi_h(a_h \mid \bar{z}_h)\{g_h(\bar{z}_h; \theta_h) - r_h - g_{h+1}(\bar{z}_{h+1}; \theta_{h+1})\}]| \leq |\mathcal{A}|(1 + 2C)\sqrt{\frac{\ln(|\Pi_h|/\delta)}{m}}.$$

Then, for $\forall \pi \in \Pi, \forall \theta_h \in \mathcal{N}_{\epsilon,h}, \forall \theta_{h+1} \in \mathcal{N}_{\epsilon,h+1}$, we have

$$|\{\mathbb{E}_\mathcal{D} - \mathbb{E}\}[|\mathcal{A}|\pi_h(a_h \mid \bar{z}_h)\{g_h(\bar{z}_h; \theta_h) - r_h - g_{h+1}(\bar{z}_{h+1}; \theta_{h+1})\}]| \leq |\mathcal{A}|(1 + 2C)\sqrt{\frac{\ln(|\Pi_h||\mathcal{N}_{\epsilon,h}||\mathcal{N}_{\epsilon,h+1}|/\delta)}{m}}.$$

Hence, for any $g_h = \langle\theta_h, \psi_h\rangle \in \mathcal{G}_h, g_{h+1} = \langle\theta_{h+1}, \psi_{h+1}\rangle \in \mathcal{G}_{h+1}$,

$$|\{\mathbb{E}_\mathcal{D} - \mathbb{E}\}[|\mathcal{A}|\pi_h(a_h \mid \bar{z}_h)\{g_h(\bar{z}_h; \theta_h) - r_h - g_{h+1}(\bar{z}_{h+1}; \theta_{h+1})\}]|$$

$$\leq |\mathcal{A}|(1 + 2C)\sqrt{\frac{\ln(|\Pi_h||\mathcal{N}_{\epsilon,h}||\mathcal{N}_{\epsilon,h+1}|/\delta)}{m}} + 4|\mathcal{A}|\epsilon.$$

By taking $\epsilon = 1/m$, we have $\forall \pi \in \Pi, \forall g_h \in \mathcal{G}_h, \forall g_{h+1} \in \mathcal{G}_{h+1}$:

$$|\{\mathbb{E}_\mathcal{D} - \mathbb{E}\}[|\mathcal{A}|\pi_h(a_h \mid \bar{z}_h)\{g_h(\bar{z}_h) - r_h - g_{h+1}(\bar{z}_{h+1})\}]|$$

$$\leq |\mathcal{A}|\{1 + 2C\}\sqrt{\frac{\{2d\ln(1 + Cm) + \ln(|\Pi_h|/\delta)\}}{m}} + \frac{4|\mathcal{A}|}{m}$$

$$\leq 5|\mathcal{A}|\{1 + 2C\}\sqrt{\frac{\{2d\ln(1 + Cm) + \ln(|\Pi_h|/\delta)\}}{m}}.$$

Similarly,

$$\forall g_1 \in \mathcal{G}_1; |\{\mathbb{E}_\mathcal{D} - \mathbb{E}\}[g_1(\bar{z}_1)]| \leq C\sqrt{\frac{\{d\ln(1 + Cm) + \ln(|\Pi_h|/\delta)\}}{m}} + \frac{4}{m}$$

$$\leq 5C\sqrt{\frac{\{d\ln(1 + Cm) + \ln(|\Pi_h|/\delta)\}}{m}}.$$

$\square$

Finally, we obtain the PAC bound, we need to find $m$ such that

$$c|\mathcal{A}|\max(C, 1)\sqrt{\frac{d_\psi \ln(\max(C, 1)m) + \ln(|\Pi_{\max}|TH/\delta)}{m}}\sqrt{d_\phi H^2 \ln(Hd_\phi B_X^2 B_W^2 m + 1)} \leq \epsilon.$$

where $c$ is some constant and

$$T = cHd_\phi \ln(HdB_X^2 B_W^2 m + 1).$$

By organizing the term, the following $m$ is sufficient:

$$c\sqrt{\frac{\{d_\psi + \ln(d_\phi|\Pi_{\max}|H^2/\delta)\}d_\phi H^2 |\mathcal{A}|^2 \max(C, 1)^2 \ln(\{C + Hd_\phi B_X^2 B_W^2 + 1\})m)^2}{m}} \leq \epsilon.$$

By using Lemma 44, we can set

$$m = \frac{B_1}{\epsilon^2}\ln(mB_1B_2)^2,$$
$$B_1 = \{d_\psi + \ln(d_\phi|\Pi_{\max}|H^2/\delta)\}d_\phi H^2 |\mathcal{A}|^2 \max(C, 1)^2, B_2 = C + Hd_\phi B_X^2 B_W^2 + 1.$$

Thus, the final sample complexity is

$$\tilde{O}\left(\frac{d_\phi^2\{d_\psi + \ln(|\Pi_{\max}|/\delta)\}H^4 |\mathcal{A}|^2 \max(C, 1)^2}{\epsilon^2}\right)$$

where $C = \Theta_V/\sigma_{\min}(K)$.

# K  Sample Complexity in Observable Undercomplete Tabular POMDPs

We revisit the existence of future-dependent value functions. Then, we show the PO-bilinear rank decomposition. After showing the uniform convergence lemma, we calculate the sample complexity.

**Existence of future-dependent value functions.**   In the tabular case, by setting

$$\psi_h(z,o) = \mathbf{1}(z) \otimes \mathbf{1}(o), \phi_h(z,s) = \mathbf{1}(z) \otimes \mathbf{1}(s), K_h = \mathbb{I}_{|\mathcal{Z}_{h-1}|} \otimes \mathbb{O}.$$

where $\mathbf{1}(z)$ is a one-hot encoding vector over $\mathcal{Z}_{h-1}$, we can regard the tabular model as an HSE-POMDP. Here is our assumption.

**Assumption 8.** *(a) $0 \le r_h \le 1$, (b) $\mathbb{O}$ is full-column rank and $\|\mathbb{O}^\dagger\|_1 \le 1/\sigma_1$ for any $h \in [H]$.*

Note we use the 1-norm since this choice is more amenable in the tabular setting. However, even if the norm bound is given in terms of 2-norm, we can still ensure the PAC guarantee  (this is because $\|\mathbb{O}^\dagger\|_1/\sqrt{|\mathcal{S}|} \le \|\mathbb{O}^\dagger\|_2 \le \|\mathbb{O}^\dagger\|_1\sqrt{|\mathcal{O}|}$).

Here, since we assume the reward lies in $[0,1]$, value functions on the latent state belong to $\{\langle \theta, \phi_h(\cdot) \rangle : \|\theta\|_\infty \le H\}$. Here, letting $V_h^\pi = \langle \theta_h^\pi, \phi_h \rangle$, future-dependent value functions exist by taking $\langle \theta_h^\pi, \mathbf{1}(z) \times \mathbb{O}^\dagger \mathbf{1}(o) \rangle$. Hence, we take

$$\mathcal{G}_h = \left\{ (z,o) \mapsto \langle \theta, \mathbf{1}(z) \otimes \mathbb{O}^\dagger \mathbf{1}(o) \rangle; \|\theta\|_\infty \le H \right\}$$

so that the realizability holds. Importantly, we can ensure $\|\mathcal{G}_h\|_\infty \le H/\sigma_1$ since

$$|\langle \theta, \mathbf{1}(z) \otimes \mathbb{O}^\dagger \mathbf{1}(o) \rangle| \le \|\theta\|_\infty \|\mathbf{1}(z) \otimes \mathbb{O}^\dagger \mathbf{1}(o)\|_1 \le \|\theta\|_\infty \|\mathbb{O}^\dagger \mathbf{1}(o)\|_1 \le H/\sigma_1$$

for any $(z,o) \in \mathcal{Z}_{h-1} \times \mathcal{O}$. Note $\mathcal{G}_h$ is contained in

$$\left\{ \langle \theta, \mathbf{1}(z) \otimes \mathbf{1}(o) \rangle; \|\theta\|_2 \le H|\mathcal{O}|^{M+1}|\mathcal{A}|^M/\sigma_1 \right\} \tag{21}$$

This is because each $\langle \theta, \mathbf{1}(z) \otimes \mathbb{O}^\dagger \mathbf{1}(o) \rangle$ is equal to $\langle \theta', \mathbf{1}(z) \otimes \mathbf{1}(o) \rangle$ for some vector $\theta' \in \mathbb{R}^{|\mathcal{Z}_{h-1}| \times |\mathcal{O}|}$. Here, denoting the component of $\theta$ corresponding to $z \in \mathcal{Z}_{h-1}$ by $\theta_z \in \mathbb{R}^{|\mathcal{O}|}$, $\theta'$ is a vector stacking $\mathbb{O}^\dagger \theta_z$ for each $z \in \mathcal{Z}_{h-1}$. Then, we have

$$\|\mathbb{O}^\dagger \theta_z\|_2 \le \|\mathbb{O}^\dagger\|_2 \|\theta_z\|_2 \le \|\mathbb{O}^\dagger\|_1 \sqrt{|\mathcal{O}|} H\sqrt{|\mathcal{O}|} \le H|\mathcal{O}|/\sigma_1.$$

Hence, $\|\theta'\|_2 \le |\mathcal{O}|^M |\mathcal{A}|^M \times H|\mathcal{O}|/\sigma_1$.

**PO-Bilinear decomposition.**   Next, recall we derive the PO-bilinear decomposition:

$$\mathbb{E}[\theta_h^\top \phi_h(\bar{z}_h) - r_h - \theta_{h+1}^\top \phi_{h+1}(\bar{z}_{h+1}); a_{1:h-1} \sim \pi', a_h \sim \pi]$$
$$= \langle K_h^\top \{\theta_h - \theta_h^\pi\} - \{T_{\pi:h}\}^\top K_{h+1}^\top \{\theta_{h+1} - \theta_{h+1}^\pi\}, \mathbb{E}[\phi_h(z_{h-1}, s_h); a_{1:h-1} \sim \pi'] \rangle.$$

Then, $B_X = 1$ and $B_W = 4H|\mathcal{O}|^{M+1}|\mathcal{A}|^M/\sigma_1$. We use $\|K_h^\top\|_2 = \|\mathbb{O}_h\|_2 \le 1, \|T_{\pi:h}^\top\|_2 \le 1$. This is because

$$\|K_h^\top \{\theta_h - \theta_h^\pi\} - \{T_{\pi:h}\}^\top K_{h+1}^\top \{\theta_{h+1} - \theta_{h+1}^\pi\}\|_2$$
$$\le \|\theta_h\|_2 + \|\theta_h^\pi\|_2 + \|\theta_{h+1}\|_2 + \|\theta_{h+1}^\pi\|_2 \le 4H|\mathcal{O}|^{M+1}|\mathcal{A}|^M/\sigma_1.$$

In the last line, we use (21).

**Uniform convergence.**   Then, we can obtain the following uniform convergence lemma.

**Lemma 25.** *Let $C = H/\sigma_1$ and $d_\psi = |\mathcal{O}|^{M+1}|\mathcal{A}|^M$. Then, with probability $1 - \delta$,*

$$\sup_{\pi \in \Pi, g \in \mathcal{G}} |\{\mathbb{E}_\mathcal{D} - \mathbb{E}\}[|\mathcal{A}|\pi_h(a_h \mid \bar{z}_h) \{g_h(\bar{z}_h) - r_h - g_{h+1}(\bar{z}_{h+1})\}]|$$

$$\le 5|\mathcal{A}|\{1 + 2C\}\sqrt{\frac{\{d_\psi^2 \ln(1 + Cm) + \ln(|\Pi_h|/\delta)\}}{m}}$$

*and*

$$\sup_{g_1 \in \mathcal{G}_1} |\{\mathbb{E}_\mathcal{D} - \mathbb{E}\}[g_1(\bar{z}_1)]| \le 5C\sqrt{\frac{\{d_\psi \ln(1 + Cm) + \ln(|\Pi_h|/\delta)\}}{m}}.$$

*Proof.* Let $d_\phi = |\mathcal{S}||\mathcal{O}|^M|\mathcal{A}|^M, d_\psi = |\mathcal{O}|^{M+1}|\mathcal{A}|^M$.

Define $\mathcal{N}_{\epsilon,h}$ as an $\epsilon$-net for $\{\theta : \|\theta\|_2 \le C\}$ with respect to $L^2$-norm. Define $\mathcal{N}'_{\epsilon,h}$ as an $\epsilon$-net for $\Pi_h : \bar{\mathcal{Z}}_h \to \Delta(\mathcal{A})$ with respect to the following norm:
$$d(\pi, \pi') = \max_{\bar{z}_{h-1} \in \bar{\mathcal{Z}}_{h-1}} \|\pi(\cdot \mid \bar{z}_{h-1}) - \pi'(\cdot \mid \bar{z}_{h-1})\|_1.$$
Then, $|\mathcal{N}_{\epsilon,h}| \le (1 + C/\epsilon)^d, |\mathcal{N}'_{\epsilon,h}| \le (1 + 1/\epsilon)^{d_\psi|\mathcal{A}|}$.

Let $g_h = \langle\theta_h, \psi_h\rangle, g_h^\diamond = \langle\theta_h^\diamond, \psi_h\rangle$ where $\psi_h$ is a one-hot encoding vector over $\bar{\mathcal{Z}}_h$. Then, when $\|\theta_h - \theta_h^\diamond\|_2 \le \epsilon, \|\theta_{h+1} - \theta_{h+1}^\diamond\|_2 \le \epsilon, \|\pi_h - \pi_h^\diamond\|_1 \le \epsilon$, we have
$$|l_h(\cdot; \pi, g) - l_h(\cdot; \pi^\diamond, g^\diamond)| \le |\mathcal{A}|\{\|\pi_h - \pi_h^\diamond\|_\infty C + \|g_h - g_h^\diamond\|_\infty + \|g_{h+1} - g_{h+1}^\diamond\|_\infty\}$$
$$\le |\mathcal{A}|\{\epsilon C + \|\theta_h - \theta_h^\diamond\|_2 + \|\theta_{h+1} - \theta_{h+1}^\diamond\|_2\}$$
$$\le 3|\mathcal{A}|C\epsilon.$$
Besides, for fixed $\pi \in \mathcal{N}'_{\epsilon,h}, \theta_h \in \mathcal{N}_{\epsilon,h}, \theta_{h+1} \in \mathcal{N}_{\epsilon,h+1}$, we have

$$|\{\mathbb{E}_\mathcal{D} - \mathbb{E}\}[|\mathcal{A}|\pi_h(a_h \mid \bar{z}_h)\{g_h(\bar{z}_h; \theta_h) - r_h - g_{h+1}(\bar{z}_{h+1}; \theta_{h+1})\}]| \le |\mathcal{A}|(1 + 2C)\sqrt{\frac{\ln(1/\delta)}{m}}.$$
Then, for $\forall \pi \in \mathcal{N}'_{\epsilon,h}, \forall \theta_h \in \mathcal{N}_{\epsilon,h}, \forall \theta_{h+1} \in \mathcal{N}_{\epsilon,h+1}$, we have

$$|\{\mathbb{E}_\mathcal{D} - \mathbb{E}\}[|\mathcal{A}|\pi_h(a_h \mid \bar{z}_h)\{g_h(\bar{z}_h; \theta_h) - r_h - g_{h+1}(\bar{z}_{h+1}; \theta_{h+1})\}]| \le |\mathcal{A}|(1 + 2C)\sqrt{\frac{\ln(|\mathcal{N}'_{\epsilon,h}||\mathcal{N}_{\epsilon,h}||\mathcal{N}_{\epsilon,h+1}|/\delta)}{m}}.$$
Hence, for any $\pi_h \in \Pi_h, g_h = \langle\theta_h, \psi_h\rangle \in \mathcal{G}_h, g_{h+1} = \langle\theta_{h+1}, \psi_{h+1}\rangle \in \mathcal{G}_{h+1}$,
$$|\{\mathbb{E}_\mathcal{D} - \mathbb{E}\}[|\mathcal{A}|\pi_h(a_h \mid \bar{z}_h)\{g_h(\bar{z}_h; \theta_h) - r_h - g_{h+1}(\bar{z}_{h+1}; \theta_{h+1})\}]|$$

$$\le |\mathcal{A}|(1 + 2C)\sqrt{\frac{\ln(|\mathcal{N}'_{\epsilon,h}||\mathcal{N}_{\epsilon,h}||\mathcal{N}_{\epsilon,h+1}|/\delta)}{m}} + 3|\mathcal{A}|C\epsilon.$$
By taking $\epsilon = 1/m$, we have $\forall \pi \in \Pi, \forall g_h \in \mathcal{G}_h, \forall g_{h+1} \in \mathcal{G}_{h+1}$;
$$|\{\mathbb{E}_\mathcal{D} - \mathbb{E}\}[|\mathcal{A}|\pi_h(a_h \mid \bar{z}_h)\{g_h(\bar{z}_h) - r_h - g_{h+1}(\bar{z}_{h+1})\}]|$$

$$\le |\mathcal{A}|\{1 + 2C\}\sqrt{\frac{\{2d_\psi \ln(1 + Cm) + d_\psi|\mathcal{A}|\ln(1 + m) + \ln(1/\delta)\}}{m}} + \frac{3|\mathcal{A}|C}{m}$$
$$\le 10|\mathcal{A}|C\sqrt{\frac{\{d_\psi|\mathcal{A}|\ln(1 + Cm) + \ln(1/\delta)\}}{m}}.$$

$\square$

**Sample Complexity.** Finally, we obtain the PAC bound. We need to find $m$ such that
$$c|\mathcal{A}|C\sqrt{\frac{\{d_\psi|\mathcal{A}|\ln(1 + Cm) + \ln(TH/\delta)\}}{m}}\sqrt{d_\phi H^2 \ln(Hd_\phi B_X^2 B_W^2 m + 1)} \le \epsilon.$$
where $c$ is some constant and
$$T = cHd_\phi \ln(HdB_X^2 B_W^2 m + 1).$$
By organizing terms, we get
$$\sqrt{\frac{|\mathcal{A}|^3 C^2 d_\phi d_\psi H^2 \ln(H^2 d_\phi/\delta)\ln(\{C + d_\psi + Hd_\phi B_X^2 B_W^2\}m)^2}{m}} \le \epsilon.$$
Thus, we need to set
$$m = \tilde{O}\left(\frac{|\mathcal{A}|^3 C^2 d_\phi d_\psi H^2 \ln(1/\delta)}{\epsilon^2}\right)$$
Hence, the sample complexity is
$$\tilde{O}\left(\frac{|\mathcal{A}|^3 C^2 d_\phi^2 d_\psi H^4 \ln(1/\delta)}{\epsilon^2}\right).$$
By some algebra, it is
$$\tilde{O}\left(\frac{|\mathcal{A}|^{3M+3}|\mathcal{O}|^{3M+1}|\mathcal{S}|^2 H^6 \ln(1/\delta)}{\epsilon^2 \sigma_1^2}\right).$$
Later, we prove we can remove $|\mathcal{O}|^M$ using the more refined analysis in Section Q.

**Global optimality.** We use a result in the proof of [23, Theorem 1.2]. We just set $M = C(1/\sigma_1)^4 \ln(SH/\epsilon)$. Note their assumption 1 is satisfied when $\|\mathbb{O}^\dagger\|_1 \leq (1/\sigma_1)$. More specifically, assumption 1 in [23] requires for any $b$ and $b'$, we have

$$\|\mathbb{O}b - \mathbb{O}b'\|_1 \geq 1/\sigma_1 \|b - b'\|_1.$$

This is proved as follows. Note for any $e, e'$,

$$\|\mathbb{O}^\dagger e - \mathbb{O}^\dagger e'\|_1 \leq \|\mathbb{O}^\dagger\|_1 \|e - e'\|_1.$$

Then, by setting $e = \mathbb{O}b$ and $e' = \mathbb{O}b'$, the assumption 1 is ensured. Here, we use $\mathbb{O}^\dagger \mathbb{O} = I$.

# L  Sample Complexity in Observable Overcomplete Tabular POMDPs

We first gave an overview of the result. Then, we move to the detail.

## L.1  Summary

We consider obvercomplete tabular POMDPs. In this case, the PO-bilinear rank is at most $|\mathcal{O}|^M |\mathcal{A}|^M |\mathcal{S}|$. We suppose $r_h \in [0, 1]$ for any $h \in [H]$. Assuming $\mathbb{O}^K$ is full-column rank, to satisfy the realizability, we set $\mathcal{G}_h = \{\langle \theta, \mathbf{1}(z) \otimes \{\mathbb{O}^K\}^\dagger \mathbf{1}(t^K) \rangle \mid \|\theta\|_\infty \leq H\}$ where $\|\mathbb{O}^K\|_1 \leq 1/\sigma_1$ and $\mathbf{1}(z), \mathbf{1}(t^K)$ are one-hot encoding vectors over $\mathcal{Z}_{h-1}$ and $\mathcal{O}^K \times \mathcal{A}^{K-1}$, respectively. We set $\Pi_h = [\mathcal{Z}_h \to \Delta(\mathcal{A})]$. Then, the following holds.

**Theorem 13** (Sample complexity for overcomplete tabular models). *With probability* $1 - \delta$, *we can achieve* $J(\pi^\star) - J(\hat{\pi}) \leq \epsilon$ *when we use samples at most* $\tilde{O}\left(|\mathcal{S}|^2 |\mathcal{A}|^{3M+K+2} |\mathcal{O}|^{3M+K} H^6 (1/\epsilon)^2 (1/\sigma_1)^2 \ln(1/\delta)\right).$
*Here,* $\mathrm{polylog}(|\mathcal{S}|, |\mathcal{O}|, |\mathcal{A}|, H, 1/\sigma_1, \ln(1/\delta))$ *are omitted.*

When we use K-step futures, in the above theorem, we additionally incur $|\mathcal{A}|^K |\mathcal{O}|^K$, which is coming from a naive parameterization of $\mathcal{G}_h$. In Section G.2.3, we will see that under the model-based learning perspective (i.e., we parameterize $\mathbb{T}, \mathbb{O}$ first and then construct $\Pi$ and $\mathcal{G}$ using the model class), we will get rid of the dependence $|\mathcal{O}|^{M+K}$ and $|\mathcal{A}|^K$. This is because the complexity of the model class is independent of $M$ or $K$ (i.e., number of parameters in $\mathbb{T}, \mathbb{O}$ are $O(|\mathcal{S}|^2|\mathcal{A}||\mathcal{O}|)$).

## L.2  Detail

To simplify the presentation, we focus on the case when $\pi^{out} = U(\mathcal{A})$.

**Existence of future-dependent value functions.** In the tabular case, by setting

$$\psi_h(z, t^K) = \mathbf{1}(z) \otimes \mathbf{1}(t^K), \phi_h(z, s) = \mathbf{1}(z) \otimes \mathbf{1}(s), K_h = \mathbb{I}_{|\mathcal{Z}_{h-1}|} \otimes \mathbb{O}^K.$$

where $\mathbf{1}(z)$ is a one-hot encoding vector over $\mathcal{Z}_{h-1}$ and $\mathbf{1}(t^K)$ is a one-hot encoding vector over $\mathcal{Z}^K = \mathcal{O}^K \times \mathcal{A}^{K-1}$, we can regard the tabular model as an HSE-POMDP. Here is our assumption.

**Assumption 9.** *(a)* $0 \leq r_h \leq 1$, *(b)* $\mathbb{O}^K$ *is full-column rank and* $\|\{\mathbb{O}^K\}^\dagger\|_1 \leq 1/\sigma_1$.

Recall we define $\mathbb{O}^K$ in Lemma 5. Since we assume the reward lies in $[0, 1]$, value functions on the latent state belong to $\{\langle \theta, \phi_h(\cdot) \rangle : \|\theta\|_\infty \leq H\}$. Here, letting $V_h^\pi(\cdot) = \langle \theta_h^\pi, \phi_h(\cdot) \rangle$, future-dependent value functions exist by taking $\langle \theta_h^\pi, \mathbf{1}(z) \otimes \{\mathbb{O}^K\}^\dagger \mathbf{1}(t^K) \rangle$. Hence, we take

$$\mathcal{G}_h = \{(z, t^K) \mapsto \langle \theta_h^\pi, \mathbf{1}(z) \otimes \{\mathbb{O}^K\}^\dagger \mathbf{1}(t^K) \rangle; \|\theta_h^\pi\|_\infty \leq H\}$$

so that the realizability holds. Importantly, we can ensure $\|\mathcal{G}_h\|_\infty \leq H/\sigma_1$ as in Section K. Then, as in Section K, $\mathcal{G}_h$ is contained in

$$\left\{\langle \theta, \mathbf{1}(z) \otimes \mathbf{1}(o) \rangle; \|\theta\|_2 \leq H|\mathcal{O}|^{M+1}|\mathcal{A}|^M/\sigma_1\right\}.$$

**PO-bilinear decomposition.** Next, we derive the PO-bilinear decomposition:

$$\mathbb{E}[\theta_h^\top \phi(z_{h-1}, t_h^K) - r_h - \theta_{h+1}^\top \phi(z_{h-1}, t_h^K); a_{1:h-1} \sim \pi', a_h \sim \pi, a_{h+1:h+K-1} \sim U(\mathcal{A})]$$
$$= \langle \{K_h\}^\top \{\theta_h - \theta_h^\pi\} - \{T_{\pi:h}\}^\top \{K_{h+1}\}^\top \{\theta_{h+1} - \theta_{h+1}^\pi\}, \mathbb{E}[\phi_h(z_{h-1}, s_h); a_{1:h-1} \sim \pi'] \rangle.$$

Then, $B_X = 1$ and $B_W = 4H|\mathcal{O}|^{M+1}|\mathcal{A}|^M/\sigma_1$. We use $\|K_h\|_2 = \|\mathbb{O}^K\|_2 \leq 1, \|T_{\pi:h}^\top\|_2 \leq 1$.

**Sample Complexity.** We can follow the same procedure in the proof of Section K. Let $d_\phi = |\mathcal{S}||\mathcal{O}|^M|\mathcal{A}|^M, d_\psi = |\mathcal{O}|^{M+K}|\mathcal{A}|^{M+K-1}$. Hence, the sample complexity is

$$\tilde{O}\left(\frac{|\mathcal{A}|^3 C^2 d_\phi^2 d_\psi H^4 \ln(1/\delta)}{\epsilon^2 \sigma_1^2}\right).$$

By some algebra, the above is

$$\tilde{O}\left(\frac{|\mathcal{A}|^{3M+K+2}|\mathcal{O}|^{3M+K}|\mathcal{S}|^2 H^6 \ln(1/\delta)}{\epsilon^2 \sigma_1^2}\right).$$

Using the more refined analysis later, we show we can remove $|\mathcal{O}|^{3M+K}$ in Section Q.

## M  Sample Complexity in LQG

In this section, we derive the sample complexity in LQG. We first explain the setting. Then, we prove the existence of future-dependent value functions. Lemma 3 is proved there. Furthermore, we show the PO-bilinear rank decomposition in LQG. We prove Lemma 3 there. Next, we show the uniform convergence result in LQG. Finally, by invoking Theorem 1, we calculate the sample complexity.

We study a finite-horizon discrete time LQG governed by the following equation:

$$s_1 = \epsilon_1, s_{h+1} = As_h + Ba_h + \epsilon_h, r_h = s_h^\top Q s_h + a_h^\top R a_h, o_h = Os_h + \tau_h.$$

where $\epsilon_h$ is Gaussian noise with mean $0$ and noise $\Sigma_\epsilon$ and $\tau_h$ is a Gaussian noise with mean $0$ and $\Sigma_\tau$. We use a matrix $O$ instead of $C$ to avoid notational confusion later. With a linear policy $\pi_h(a_h \mid o_h, z_{h-1}) = \delta(a_h = \mathbf{U}_{1h}o_h + \mathbf{U}_{2h}z_{h-1})$, this induces the following system:

$$\begin{bmatrix} z_h' \\ o_h \\ a_h \\ s_{h+1} \end{bmatrix} = \Xi_{1h}(\pi)\begin{bmatrix} z_{h-1} \\ s_h \end{bmatrix} + \Xi_{2h}(\pi), \Xi_{2h}(\pi) = \begin{bmatrix} 0 \\ \tau \\ \mathbf{U}_{1h}\tau \\ B\mathbf{U}_{1h}\tau + \epsilon \end{bmatrix}, \Xi_{1h}(\pi) = \begin{bmatrix} I' & 0 \\ 0 & O \\ \mathbf{U}_{2h} & \mathbf{U}_{1h}O \\ B\mathbf{U}_{2h} & A + B\mathbf{U}_{1h}O \end{bmatrix}$$

where $z_h'$ is the vector removing $(o_h, a_h)$ from $z_h$ and $I'$ is a matrix mapping $z_h$ to $z_h'$. This is derived by

$$\begin{aligned} s_{h+1} &= As_h + Ba_h + \epsilon = As_h + B\{\mathbf{U}_{1h}o_h + \mathbf{U}_{2h}z_{h-1}\} + \epsilon \\ &= (A + B\mathbf{U}_{1h}O)s_h + B\mathbf{U}_{2h}z_{h-1} + \epsilon + B\mathbf{U}_{1h}\tau, \\ a_h &= \mathbf{U}_{1h}o_h + \mathbf{U}_{2h}z_{h-1} = \mathbf{U}_{1h}Os_h + \mathbf{U}_{2h}z_{h-1} + \mathbf{U}_{1h}\tau, \\ o_h &= Os_h + \tau. \end{aligned}$$

We suppose the system is always stable in the sense that the operator norm of $\Xi_{1h}(\pi)$ is upper-bounded by $1$. Here is the assumption we introduce throughout this section.

**Assumption 10.** *Suppose* $\max(\|A\|, \|B\|, \|O\|, \|Q\|, \|R\|) \leq \mathbb{C}$. *Suppose* $\|\Xi_{1h}(\pi)\| \leq 1$ *for any* $\pi$. *$O$ is full-column rank.*

We present the form of linear mean embedding operators in LQGs.

**Lemma 26** (Linear mean embedding operator). *Let* $z \in \mathcal{Z}_{h-1}, o \in \mathcal{O}, s \in \mathcal{S}$. *We have*

$$\mathbb{E}_{o\sim\mathbb{O}(s)}\left[\begin{bmatrix} \begin{bmatrix} 1 \\ z \\ o \end{bmatrix} \otimes \begin{bmatrix} 1 \\ z \\ o \end{bmatrix} \end{bmatrix}\right] = K_h\begin{bmatrix} \begin{bmatrix} 1 \\ z \\ s \end{bmatrix} \otimes \begin{bmatrix} 1 \\ z \\ s \end{bmatrix} \end{bmatrix}, K_h = \begin{bmatrix} 1 & \mathbf{0} \\ \text{vec}\left(\begin{bmatrix} 0 & 0 \\ 0 & \Sigma_\tau \end{bmatrix}\right) & \begin{bmatrix} \mathbb{I} & 0 \\ 0 & O \end{bmatrix} \otimes \begin{bmatrix} \mathbb{I} & 0 \\ 0 & O \end{bmatrix} \end{bmatrix}.$$

*Proof.* Here, we have

$$\begin{aligned} \mathbb{E}_{o\sim\mathbb{O}(s)}\left[\begin{bmatrix} z \\ o \end{bmatrix} \otimes \begin{bmatrix} z \\ o \end{bmatrix}\right] &= \text{Vec}\left[\begin{bmatrix} zz^\top & zo^\top \\ oz^\top & oo^\top \end{bmatrix}\right] = \text{Vec}\left[\begin{bmatrix} zz^\top & zs^\top O^\top \\ Osz^\top & Oss^\top O^\top + \Sigma_r \end{bmatrix}\right] \\ &= \text{Vec}\left[\begin{bmatrix} 0 & 0 \\ 0 & \Sigma_r \end{bmatrix}\right] + \begin{bmatrix} \mathbb{I} & 0 \\ 0 & O \end{bmatrix} \otimes \begin{bmatrix} \mathbb{I} & O \\ 0 & O \end{bmatrix} \times \text{Vec}\left[\begin{bmatrix} zz^\top & zs^\top \\ sz^\top & ss^\top \end{bmatrix}\right]. \end{aligned}$$

From the second line to the third line, we use formula $\text{vec}[A_1 A_2 A_3] = (A_3^\top \otimes A_1)\text{vec}(A_2)$. This immediately concludes the result.

$\square$

Thus, the matrix $K_h$ has the left inverse when $O$ is full-column rank as follows:

$$K_h^\dagger = \left[ -\begin{bmatrix} \mathbb{I} & 0 \\ 0 & O^\dagger \end{bmatrix} \otimes \begin{bmatrix} \mathbb{I} & 0 \\ 0 & O^\dagger \end{bmatrix} \text{vec} \left( \begin{bmatrix} 0 & 0 \\ 0 & \Sigma_\tau \end{bmatrix} \right) \quad \begin{matrix} \mathbf{0} \\ \begin{bmatrix} \mathbb{I} & 0 \\ 0 & O^\dagger \end{bmatrix} \otimes \begin{bmatrix} \mathbb{I} & 0 \\ 0 & O^\dagger \end{bmatrix} \end{matrix} \right].$$

We use a block matrix inversion formula:

$$\begin{bmatrix} A_1^{-1} & 0 \\ -A_3^\dagger A_2 A_1^{-1} & A_3^\dagger \end{bmatrix} \begin{bmatrix} A_1 & 0 \\ A_2 & A_3 \end{bmatrix} = I.$$

## M.1 Existence of Link Functions

**Lemma 27** (Value functions in LQGs). *Let $\pi_h(a \mid o, z) = \delta(a = \mathbf{U}_{1h}o + \mathbf{U}_{2h}z)$ for $z \in \mathcal{Z}_{h-1}, o \in \mathcal{O}$. Then, a value function has a bilinear form:*

$$V_h^\pi(z, s) = [z^\top, s^\top]\Lambda_h[z^\top, s^\top]^\top + \Gamma_h.$$

*For any $h \in [H]$, these parameters $\Lambda_h, \Gamma_h$ are recursively defined inductively by*

$$\Lambda_H = \begin{bmatrix} \mathbf{U}_{2h}^\top R \mathbf{U}_{2h} & \mathbf{U}_{2h}^\top R \mathbf{U}_{1h} O \\ \{\mathbf{U}_{2h}^\top R \mathbf{U}_{1h} O\}^\top & Q + O^\top \mathbf{U}_{1h}^\top R \mathbf{U}_{1h} O \end{bmatrix}, O_H = \text{tr}(\mathbf{U}_{1h}^\top R \mathbf{U}_{1h} \Sigma_\tau),$$

$$\Lambda_h = \Xi_{1h}(\pi)\Lambda_{h+1}\Xi_{1h}^\top(\pi) + \Sigma_{\Lambda_h}, \quad \Sigma_{\Lambda_{h1}} = \begin{bmatrix} \mathbf{U}_{2h}^\top R \mathbf{U}_{2h} & \mathbf{U}_{2h}^\top R \mathbf{U}_{1h} O \\ \{\mathbf{U}_{2h}^\top R \mathbf{U}_{1h} O\}^\top & Q + O^\top \mathbf{U}_{1h}^\top R \mathbf{U}_{1h} O \end{bmatrix},$$

$$\Gamma_h = \text{tr}\left(\Lambda_{h+1}\Sigma_{\Lambda_{h2}}(\pi)\right) + \Gamma_{h+1}, \quad \Sigma_{\Lambda_{h2}}(\pi) = \begin{bmatrix} 0 & 0 & 0 & 0 \\ 0 & \Sigma_\tau & \Sigma_\tau \mathbf{U}_{1h}^\top & \Sigma_\tau \mathbf{U}_{1h}^\top B^\top \\ 0 & \mathbf{U}_{1h}\Sigma_\tau & \mathbf{U}_{1h}\Sigma_\tau \mathbf{U}_{1h}^\top & \mathbf{U}_{1h}\Sigma_\tau \mathbf{U}_{1h}^\top B^\top \\ 0 & B\mathbf{U}_{1h}\Sigma_\tau & B\mathbf{U}_{1h}\Sigma_\tau \mathbf{U}_{1h}^\top & B\mathbf{U}_{1h}\Sigma_\tau \mathbf{U}_{1h}^\top B^\top + \Sigma_\epsilon \end{bmatrix}.$$

*Proof.* The proof is completed by backward induction regarding $h$, starting from level $H$. First, we have

$$V_H^\pi(z, s) = s^\top Q s + \mathbb{E}_{o \sim O(s)}[\{\mathbf{U}_{1h}o + \mathbf{U}_{2h}z\}^\top R\{\mathbf{U}_{1h}o + K_2 z\}]$$

$$= s^\top Q s + \mathbb{E}_{o \sim O(s)}[\{\mathbf{U}_{1h}Os + \mathbf{U}_{1h}\tau + \mathbf{U}_{2h}z\}^\top R\{\mathbf{U}_{1h}Os + \mathbf{U}_{1h}\tau + \mathbf{U}_{2h}z\}]$$

$$= s^\top\{Q + O^\top \mathbf{U}_{1h}^\top R \mathbf{U}_{1h} O\}s + z^\top \mathbf{U}_{2h}^\top R \mathbf{U}_{2h} z + 2z^\top \mathbf{U}_{2h}^\top R \mathbf{U}_{1h} Os + \text{tr}(\mathbf{U}_{1h}^\top R \mathbf{U}_{1h} \Sigma_\tau)$$

$$= [z^\top, s^\top] \begin{bmatrix} \mathbf{U}_{2h}^\top R \mathbf{U}_{2h} & \mathbf{U}_{2h}^\top R \mathbf{U}_{1h} O \\ \{\mathbf{U}_{2h}^\top R \mathbf{U}_{1h} O\}^\top & Q + O^\top \mathbf{U}_{1h}^\top R \mathbf{U}_{1h} O \end{bmatrix} [z^\top, s^\top]^\top + \text{tr}(\mathbf{U}_{1h}^\top R \mathbf{U}_{1h} \Sigma_\tau).$$

Here, we use induction. Thus, supposing the statement is true at horizon $h + 1$, we have

$$V_h^\pi(z, s) = \Gamma_{h+1} + s^\top Q s + \mathbb{E}_{o \sim O(s)}[\{\mathbf{U}_{1h}o + \mathbf{U}_{2h}z\}^\top R\{\mathbf{U}_{1h}o + K_2 z\}]$$

$$+ \mathbb{E}_{o \sim O(s), a \sim \pi(o,z), s' \sim \mathbb{T}(s,a)}[[z_{-1}^\top, o^\top, a^\top, s'^\top]\Lambda_{h+1}[z_{-1}^\top, o^\top, a^\top, s'^\top]^\top]$$

where $z'$ is a vector that removes the last component $(o, a)$ from $z$ and $s'$ is a state at $h + 1$. Here, recall we have

$$[(z')^\top, o^\top, a^\top, s'^\top]^\top = \Xi_{1h}(\pi)[z^\top, s^\top]^\top + \Xi_{2h}(\pi).$$

Then, the statement is concluded some algebra.

$\square$

**Lemma 28** (Norm constraints on value functions). *We can set $\|\Lambda_h\| \leq \mathbb{C}_{\Lambda,h}, \|\Gamma_h\| \leq \mathbb{C}_{\Gamma,h}$ such that*

$$\mathbb{C}_{\Lambda,h} = \text{poly}(\mathbb{C}, H), \mathbb{C}_{\Gamma,h} = \text{poly}(d_o, d_s, d_a, \mathbb{C}, H).$$

*Proof.* We have

$$\|\Lambda_H\| \leq \text{poly}(\mathbb{C}, H), \quad \|\Gamma_H\| \leq \text{poly}(\mathbb{C}, H).$$

Then,
$$\|\Lambda_h\| \le \|\Xi_{1h}(\pi)\|\|\Lambda_{h+1}\|\|\Xi_{1h}(\pi)\| + \text{poly}(\mathbb{C}, H).$$

Since we assume $\|\Xi_{1h}(\pi)\| \le 1$, this immediately leads to
$$\|\Lambda_h\| \le \text{poly}(\mathbb{C}, H).$$

Besides,
$$\|\Gamma_h\| \le \text{poly}(H, d_o, d_s, d_a, \mathbb{C})\|\Lambda_{h+1}\| + \|\Gamma_{h+1}\|.$$

Thus,
$$\|\Gamma_h\| \le \text{poly}(H, d_o, d_s, d_a, \mathbb{C}).$$

$\square$

Next, we set the norm on the function class $\mathcal{G}_h$.

**Lemma 29** (Realizability on LQGs). *We set*
$$\mathcal{G}_h = \left\{ \bar{\Gamma}_h + (z^\top, o^\top)\bar{\Lambda}_h(z^\top, o^\top)^\top \mid \|\bar{\Lambda}_h\| \le C_{\bar{\Lambda},h}, |\bar{\Gamma}_h| \le C_{\bar{\Gamma},h}, z \in Z_{h-1}, o \in \mathcal{O} \right\},$$
$$C_{\bar{\Lambda},h} = \text{poly}(H, d_o, d_s, d_a, \mathbb{C}, \|O^\dagger\|), C_{\bar{\Gamma},h} = \text{poly}(H, d_o, d_s, d_a, \mathbb{C}, \|O^\dagger\|).$$

*A function class $\mathcal{G}_h$ includes at least one value future-dependent value function for any linear policy $\pi = \delta(a = \mathbf{U}_{1h}o + \mathbf{U}_{2h}z)$ for $\|\mathbf{U}_{1h}\| \le \mathbb{C}, \|\mathbf{U}_{2h}\| \le \mathbb{C}$.*

*Proof.* Here, we have

$$V_h^\pi(\cdot) = \Gamma_h + \text{tr}\left\{\Lambda_h \begin{bmatrix} zz^\top & zs^\top \\ sz^\top & ss^\top \end{bmatrix}\right\}$$

$$= \Gamma_h + \mathbb{E}_{o \sim \mathbb{O}(s)}\left[\text{tr}\left\{\Lambda_h \begin{bmatrix} zz^\top & zo^\top\{O^\dagger\}^\top \\ O^\dagger oz^\top & O^\dagger\{oo^\top - \Sigma_\tau\}\{O^\dagger\}^\top \end{bmatrix}\right\}\right]$$

$$= \Gamma_h - \text{tr}\left\{\Lambda_h \begin{bmatrix} 0 & 0 \\ 0 & O^\dagger\Sigma_\tau\{O^\dagger\}^\top \end{bmatrix}\right\}$$

$$+ \mathbb{E}_{o \sim \mathbb{O}(s)}\left[[z^\top, o^\top]\begin{bmatrix} I & 0 \\ 0 & \{O^\dagger\}^\top \end{bmatrix}\Lambda_h \begin{bmatrix} I & 0 \\ 0 & O^\dagger \end{bmatrix}\begin{bmatrix} z \\ o \end{bmatrix}\right].$$

The norm constraint on $\bar{\Lambda}_h$ is decided by the following calculation:

$$\left\|\begin{bmatrix} I & 0 \\ 0 & \{O^\dagger\}^\top \end{bmatrix}\Lambda_h \begin{bmatrix} I & 0 \\ 0 & O^\dagger \end{bmatrix}\right\| \le \|O^\dagger\|_2^2\|\Lambda_h\| = \text{poly}(H, d_o, d_s, d_a, \mathbb{C}, \|O^\dagger\|).$$

Then, the norm on $\bar{\Gamma}_h$ is decided by the following calculation:

$$\left|\Gamma_h - \text{tr}\left\{\Lambda_h \begin{bmatrix} 0 & 0 \\ 0 & O^\dagger\Sigma_\tau\{O^\dagger\}^\top \end{bmatrix}\right\}\right| \le |\Gamma_h| + \left|\text{tr}\left\{\Lambda_h \begin{bmatrix} 0 & 0 \\ 0 & O^\dagger\Sigma_\tau\{O^\dagger\}^\top \end{bmatrix}\right\}\right|$$

$$\le |\Gamma_h| + \|\Sigma_h\|_2\|O^\dagger\|_2^2\text{Tr}(\Sigma_\tau)$$

$$\le |\Gamma_h| + \|\Sigma_h\|_2\|O^\dagger\|_2^2\mathbb{C}d_o$$

$$= \text{poly}(H, d_o, d_s, d_a, \mathbb{C}, \|O^\dagger\|).$$

From the first line to the second line, we use Lemma 43.

$\square$

## M.2 PO-bilinear Rank Decomposition

**Lemma 30** (Bilinear rank decomposition for LQG). *For any $g_{h+1} \in \mathcal{G}_{h+1}, g_h \in \mathcal{G}_h$, we have the following bilinear rank decomposition:*
$$\mathbb{E}[g_{h+1}(z_h, o_{h+1}) + r_h - g_h(z_{h-1}, o_h); a_{1:h-1} \sim \pi', a_h \sim \pi] = \langle X(\pi'), W(\pi) \rangle$$

*where*

$$X_h(\pi') = (1, \mathbb{E}[[z_{h-1}^\top, s_h^\top] \otimes [z_{h-1}^\top, s_h^\top]; a_{1:h-1} \sim \pi'])^\top,$$

$$W_h(\pi) = \begin{bmatrix} \operatorname{tr}\left(\{\bar{\Lambda}_h - \bar{\Lambda}_h^\star\}\begin{bmatrix} 0 & 0 \\ 0 & \Sigma_\tau \end{bmatrix} + \begin{bmatrix} I & 0 \\ 0 & O^\top \end{bmatrix}\{\bar{\Lambda}_{h+1} - \bar{\Lambda}_{h+1}^\star\}\begin{bmatrix} I & 0 \\ 0 & O \end{bmatrix}\Sigma_{\Lambda_{h2}}(\pi)\right) \\ \operatorname{vec}\left[\begin{bmatrix} I & 0 \\ 0 & O^\top \end{bmatrix}\{\bar{\Lambda}_h - \bar{\Lambda}_h^\star\}\begin{bmatrix} I & 0 \\ 0 & O \end{bmatrix} + \Xi_{1h}^\top(\pi)\begin{bmatrix} I & 0 \\ 0 & O^\top \end{bmatrix}\{\bar{\Lambda}_{h+1}^\star - \bar{\Lambda}_{h+1}\}\begin{bmatrix} I & 0 \\ 0 & O \end{bmatrix}\Xi_{1h}(\pi)\right] \end{bmatrix}.$$

*Here, $\Xi_{1h}(\pi)$ and $\Sigma_{\Lambda_{h2}}(\pi)$ depend on a policy $\pi$. The following norm constraints hold:*

$$\|X_h(\pi')\|_2 \le \operatorname{poly}(H, d_o, d_s, d_a, \mathbb{C}, \|O^\dagger\|), \quad \|W_h(\pi)\|_2 \le \operatorname{poly}(H, d_o, d_s, d_a, \mathbb{C}, \|O^\dagger\|).$$

*Proof.* We have

$$\mathbb{E}[g_h(z_{h-1}, o_h) - r_h(z_{h-1}, s_h) - g_{h+1}(z_h, o_{h+1}); a_{1:h-1} \sim \pi', a_h \sim \pi]$$
$$= -\mathbb{E}[r_h(z_{h-1}, s_h); a_{1:h-1} \sim \pi', a_h \sim \pi] +$$
$$+ \mathbb{E}\left[\bar{\Gamma}_h + (z_{h-1}^\top, o_h^\top)\bar{\Lambda}_h(z_{h-1}^\top, o_h^\top)^\top - \bar{\Gamma}_{h+1} - (z_h^\top, o_{h+1}^\top)\bar{\Lambda}_{h+1}(z_h^\top, o_{h+1}^\top)^\top; a_{1:h-1} \sim \pi', a_h \sim \pi\right].$$
(22)

Since we have

$$\mathbb{E}[r_h(z_{h-1}, s_h); a_{1:h-1} \sim \pi', a_h \sim \pi]$$
$$= -\mathbb{E}\left[\bar{\Gamma}_h^\star + (z_{h-1}^\top, o_h^\top)\bar{\Lambda}_h^\star(z_{h-1}^\top, o_h^\top)^\top - \bar{\Gamma}_{h+1}^\star - (z_h^\top, o_{h+1}^\top)\bar{\Lambda}_{h+1}^\star(z_h^\top, o_{h+1}^\top)^\top; a_{1:h-1} \sim \pi', a_h \sim \pi\right].$$

we focus on the term (22).

Hereafter, we suppose the expectation is always taken under $a_{1:h-1} \sim \pi', a \sim \pi$. We also denote $z = z_{h-1}, o_h = o, o_{h+1} = o', s_h = s, s_{h+1} = s'$ to simplify the presentation. Using this simplified notation, we get

$$\mathbb{E}\left[(z^\top, o^\top)\bar{\Lambda}_h(z^\top, o^\top)^\top\right] = \mathbb{E}\left[(z^\top, (Os + \tau)^\top)\bar{\Lambda}_h(z^\top, (Os + \tau)^\top)^\top\right]$$
$$= \mathbb{E}\left[[z^\top, s^\top]\begin{bmatrix} I & 0 \\ 0 & O^\top \end{bmatrix}\bar{\Lambda}_h\begin{bmatrix} I & 0 \\ 0 & O \end{bmatrix}\begin{bmatrix} z \\ s \end{bmatrix}\right] + \operatorname{tr}\left(\bar{\Lambda}_h\begin{bmatrix} 0 & 0 \\ 0 & \Sigma_\tau \end{bmatrix}\right).$$

Besides,

$$\mathbb{E}\left[(z'^\top, o'^\top)\bar{\Lambda}_{h+1}(z'^\top, o'^\top)^\top\right]$$
$$= \mathbb{E}\left[[z'^\top, s'^\top]\begin{bmatrix} I & 0 \\ 0 & O^\top \end{bmatrix}\bar{\Lambda}_{h+1}\begin{bmatrix} I & 0 \\ 0 & O \end{bmatrix}\begin{bmatrix} z' \\ s' \end{bmatrix}\right] + \operatorname{tr}\left(\bar{\Lambda}_h\begin{bmatrix} 0 & 0 \\ 0 & \Sigma_\tau \end{bmatrix}\right)$$
$$= \mathbb{E}\left[[z^\top, s^\top]\Xi_{1h}^\top(\pi)\begin{bmatrix} I & 0 \\ 0 & O^\top \end{bmatrix}\bar{\Lambda}_{h+1}\begin{bmatrix} I & 0 \\ 0 & O \end{bmatrix}\Xi_{1h}(\pi)\begin{bmatrix} z \\ s \end{bmatrix}\right] + \operatorname{tr}\left(\bar{\Lambda}_h\begin{bmatrix} 0 & 0 \\ 0 & \Sigma_\tau \end{bmatrix}\right) +$$
$$+ \operatorname{tr}\left(\begin{bmatrix} I & 0 \\ 0 & O^\top \end{bmatrix}\bar{\Lambda}_{h+1}\begin{bmatrix} I & 0 \\ 0 & O \end{bmatrix}\Sigma_{\Lambda_{h2}}(\pi)\right).$$

Then, the bilinear decomposition is clear by using

$$A_2^\top A_1 A_2 = \operatorname{tr}(A_1 A_2 A_2^\top) = \operatorname{vec}(A_1^\top)^\top \operatorname{vec}(A_2 A_2^\top) = \langle \operatorname{vec}(A_1^\top), A_2 \otimes A_2 \rangle.$$

where $A_2$ is any vector and $A_1$ is any matrix.

First, we calculate the upper bounds of the norms.

$$\|X_h(\pi')\|_2^2 = 1 + \left\|\mathbb{E}_{(z,s) \sim d_h^{\pi'}}\left[\begin{bmatrix} zz^\top & zs^\top \\ sz^\top & ss^\top \end{bmatrix}\right]\right\|_F^2$$
$$= 1 + \left\|\mathbb{E}_{(z,s) \sim d_{h-1}^{\pi'}}\left[\Xi_{1h}(\pi)\begin{bmatrix} zz^\top & zs^\top \\ sz^\top & ss^\top \end{bmatrix}\Xi_{1h}^\top(\pi)\right] + \Sigma_{\Lambda_{h2}}(\pi)\right\|_F^2$$
$$\le 1 + \|\Xi_{1h}(\pi)\|_2^4 \left\|\mathbb{E}_{(z,s) \sim d_{h-1}^{\pi'}}\left[\begin{bmatrix} zz^\top & zs^\top \\ sz^\top & ss^\top \end{bmatrix}\right]\right\|_F^2 + \|\Sigma_{\Lambda_{h2}}(\pi)\|_F^2$$
$$\le 1 + \|X_{h-1}(\pi')\|_2^2 + \|\Sigma_{\Lambda_{h2}}(\pi)\|_F^2.$$

From the third line to the fourth line, we use $\|\Xi_{1h}(\pi)\|_2 \leq 1$. Thus, $\|X_h(\pi')\|_2 \leq \text{poly}(H, d_o, d_s, d_a, \|O^\dagger\|, \mathbb{C})$.

Next, we consider $W(\pi)$. By some algebra, we can see

$$\|W(\pi)\|_2 \leq \text{poly}(\|\bar{\Lambda}_h\|, \|\bar{\Lambda}_{h+1}\|, \mathbb{C}, d_o, d_s, d_a, \|\Xi_{1h}(\pi)\|))$$
$$\leq \text{poly}(H, d_o, d_s, d_a, \|O^\dagger\|, \mathbb{C})$$

$\square$

**Lemma 31** (Variance of marginal distribution). *Recall $d_h^\pi(z_{h-1}, s_h)$ is a marginal distribution over $\mathcal{Z}_{h-1} \times \mathcal{S}$ at $h$ when we execute $a_{1:h-1} \sim \pi$. The distribution $d_h^\pi(z_{h-1}, s_h)$ is a Gaussian distribution with mean $0$. The operator norm on the variance of $d_h^\pi(z_{h-1}, s_h)$ is upper-bounded by $\text{poly}(H, d_o, d_a, d_s, \mathbb{C})$.*

*Proof.* We first calculate the operator norm of the variance of $d_h^\pi(z_{h-1}, o_h)$. The variance is

$$\sum_{i=1}^h \left( \prod_{t=i+1}^h \Xi_{1t}(\pi) \right) \Sigma_{\Lambda_{i2}}(\pi) \left( \prod_{t=i+1}^h \Xi_{1t}^\top(\pi) \right).$$

The statement is immediately concluded. $\square$

Let $u_h(\bar{z}_h, r_h, a_h, o_{h+1}; \theta) = \theta_h^\top \psi_h(\bar{z}_h) - r_h - \theta_{h+1}^\top \psi_{h+1}(\bar{z}_{h+1})$. Recall $\psi_h(\bar{z}_h) = [1, \bar{z}_h^\top \otimes \bar{z}_h^\top]^\top$. We define

$$\hat{y}_h(a^{[i]}) = \mathbb{E}_\mathcal{D}\{\alpha_i(\pi(\bar{z}_h))\mathbb{I}(\|\bar{z}_h\| \leq Z_1)\mathbb{I}(\|r_h\| \leq Z_2)\mathbb{I}(\|o_{h+1}\| \leq Z_3)\mathbb{I}(a_h = a^{[i]})(1 + d^\diamond)$$
$$u_h(\bar{z}_h, r_h, a_h, o_{h+1}; \theta); a_{1:h-1} \sim \pi', a_h \sim U(1 + d^\diamond)\}$$
$$\hat{y}_h(a^{[0]}) = \mathbb{E}_\mathcal{D}\{\{1 - \sum_i \alpha_i(\pi(\bar{z}_h))\}\mathbb{I}(a = 0)\mathbb{I}(\|\bar{z}_h\| \leq Z_1)\mathbb{I}(\|r_h\| \leq Z_2)\mathbb{I}(\|o_{h+1}\| \leq Z_3)$$
$$(1 + d^\diamond)u_h(\bar{z}_h, r_h, a_h, o_{h+1}; \theta); a_{1:h-1} \sim \pi', a_h \sim U(1 + d^\diamond)\}.$$

Then, the final estimator is constructed by

$$\hat{y}_h(a^{[0]}) + \sum_{i=1}^{d^\diamond} \hat{y}_h(a^{[i]}).$$

This is equal to

$$\mathbb{E}_\mathcal{D}[l_h(\bar{z}_h, a_h, r_h, o_{h+1}; \theta, \pi)]$$

where

$$l_h(\bar{z}_h, a_h, r_h, o_{h+1}; \theta, \pi) = \left[ \sum_i \alpha_i(\pi(\bar{z}_h))\mathbb{I}(a_h = a^{[i]}) + \{1 - \sum_i \alpha_i(\pi(\bar{z}_h))\}\mathbb{I}(a_h = 0) \right] \times$$

$$\mathbb{I}(\|\bar{z}_h\| \leq Z_1)\mathbb{I}(\|r_h\| \leq Z_2)\mathbb{I}(\|o_{h+1}\| \leq Z_3)(1 + d^\diamond)u_h(\bar{z}_h, r_h, a_h, o_{h+1}; \theta).$$

We set

$$Z_i = \text{poly}(\ln(m), d_s, d_o, d_a, \mathbb{C}, H, \|O^\dagger\|).$$

for any $i \in [3]$.

### M.3 Uniform Convergence

Recall that

$$\Pi = \{\delta(a = \mathbf{U}_{1h}z + \mathbf{U}_{2h}o) \mid \|\mathbf{U}_{1h}\| \leq \mathbb{C}, \|\mathbf{U}_{2h}\| \leq \mathbb{C}\}.$$

Besides, $\mathcal{G}_h$ is included in

$$\{\langle \theta, \psi_h(\cdot) \rangle \mid \|\theta\| \leq \text{poly}(H, d_o, d_s, d_a, \mathbb{C}, \|O^\dagger\|)\}.$$

**Lemma 32** (Concentration of loss functions). *With probability $1 - \delta$,*

$$\sup_{\pi \in \Pi, \theta \in \Theta} |(\mathbb{E}_{\mathcal{D}} - \mathbb{E})\{l_h(\bar{z}_h, a_h, r_h, o_{h+1}; \theta, \pi)\}|$$

*is upper-bounded by*

$$\operatorname{poly}(\ln(m), d_s, d_o, d_a, \mathbb{C}, H, \|O^\dagger\|) \times \sqrt{\ln(1/\delta)/m}.$$

*Proof.* Due to indicator functions, $l_h(\bar{z}_h, a_h, o_h, o_{h+1}; \theta, \pi)$ is bounded for any $\pi, \theta$ by

$$\operatorname{poly}(\ln(m), d_s, d_a, d_o, \mathbb{C}, H, \|O^\dagger\|).$$

Thus, for fixed $\pi$ and $\theta$, we can say that with high probability $1 - \delta$

$$\operatorname{poly}(\ln(m), d_s, d_o, d_a, \mathbb{C}, H, \|O^\dagger\|, \ln(1/\delta)) \times \sqrt{1/m}.$$

Besides, we can consider a covering number with respect to $l^\infty$-norm for the space of $K$ and $\theta$ since both are bounded. The radius of each space is upper-bounded by

$$\operatorname{poly}(\ln(m), d_s, d_o, d_a, \mathbb{C}, H, \|O^\dagger\|).$$

Thus, by taking uniform bound and considering the bias term due to the discretization as in the proof of Lemma 24, the statement is concluded. □

**Lemma 33** (Bias terms 1). *Expectation of $\hat{y}_h(a^{[i]})$ and $\hat{y}_h(a^{[0]})$ are equal to*

$$y_h(a^{[i]}) + \mathrm{Error}_1, \quad y_h(a^{[0]}) + \mathrm{Error}_2.$$

*where*

$$y_h(a^{[i]}) = \mathbb{E}\left[\alpha_i(\pi(\bar{z}_h))\mathbb{I}(\|\bar{z}_h\| \le Z_1)u_h(\bar{z}_h, a_h, r_h, o_{h+1}; \theta); a_{1:h-1} \sim \pi', a_h \sim do(a^{[i]})\right],$$

$$y_h(a^{[0]}) = \mathbb{E}\left[\{1 - \sum_i \alpha_i(\pi(\bar{z}_h))\}\mathbb{I}(\|\bar{z}_h\| \le Z_1)u_h(\bar{z}_h, a_h, r_h, o_{h+1}; \theta); a_{1:h-1} \sim \pi', a_h \sim do(0)\right],$$

$$\mathrm{Error}_1 = m^{-1}\operatorname{poly}(\ln(m), d_s, d_o, d_a, \mathbb{C}, H, \|O^\dagger\|), \quad \mathrm{Error}_2 = m^{-1}\operatorname{poly}(\ln(m), d_s, d_o, d_a, \mathbb{C}, H, \|O^\dagger\|).$$

*Proof.* We want to upper bound the difference of

$$\mathbb{E}\left[\alpha_i(\pi(\bar{z}_h))\mathbb{I}(\|\bar{z}_h\| \le Z_1)u_h(\bar{z}_h, a_h, r_h, o_{h+1}; \theta); a_{1:h-1} \sim \pi', a_h \sim do(a^{[i]})\right]$$

and

$$\mathbb{E}\left[\alpha_i(\pi(\bar{z}_h))\mathbb{I}(\|\bar{z}_h\| \le Z_1)\mathbb{I}(\|r_h\| \le Z_2)\mathbb{I}(\|o_{h+1}\| \le Z_3)u_h(\bar{z}_h, a_h, r_h, o_{h+1}; \theta); a_{1:h-1} \sim \pi', a_h \sim do(a^{[i]})\right].$$

By CS inequality, we have

$$|\mathbb{E}\left[\alpha_i(\pi(\bar{z}_h))\mathbb{I}(\|\bar{z}_h\| \le Z_1)\{\mathbb{I}(\|r_h\| \le Z_2)\mathbb{I}(\|o_{h+1}\| \le Z_3) - 1\}u_h(\bar{z}_h, a_h, r_h, o_{h+1}; \theta); a_{1:h-1} \sim \pi', a_h \sim do(a^{[i]})\right]|$$

$$\le \underbrace{\left|\mathbb{E}\left[\{\mathbb{I}(\|r_h\| \le Z_2)\mathbb{I}(\|o_{h+1}\| \le Z_3) - 1\}^2; a_{1:h-1} \sim \pi', a_h \sim do(a^{[i]})\right]\right|}_{(a)}$$

$$\times \underbrace{\left|\mathbb{E}\left[\alpha_i^2(\pi(\bar{z}_h))u_h^2(\bar{z}_h, a_h, r_h, o_{h+1}); a_{1:h-1} \sim \pi', a_h \sim do(a^{[i]})\right]\right|^{1/2}}_{(b)}.$$

We analyze the term (a) and the term (b). Before starting analysis, note $(\bar{z}_h^\top, a_h^\top, r_h^\top, o_{h+1}^\top)$ follows Gaussian distribution with mean 0 and variance upper-bounded by

$$\operatorname{poly}(\mathbb{C}, d_s, d_o, d_a, H)$$

using Lemma 31. Besides, $\alpha_i^2(\pi_h(\bar{z}_h)) \le \operatorname{poly}(d_s, d_o, d_a, H)$ from Lemma 7. Note we can use a G-optimal design since we have a norm constraint on $\bar{z}_1$.

Regarding the term (a), by setting $Z_2 = \operatorname{poly}(\mathbb{C}, d_s, d_o, d_a, \ln(m), H, \|O^\dagger\|)$ and $Z_3 = \operatorname{poly}(\mathbb{C}, d_s, d_o, d_a, \ln(m), H, \|O^\dagger\|)$ properly, we can ensure it is upper-bounded by

$$\frac{\operatorname{poly}(\mathbb{C}, d_s, d_o, d_a, H, \|O^\dagger\|, \ln(m))}{m}.$$

Regarding the term (b), noting high order moments of Gaussian distributions can be always upper-bounded, the term (b) is upper-bounded by $\operatorname{poly}(\mathbb{C}, d_s, d_o, d_a, H, \|O^\dagger\|, \ln(m))$. This concludes the statement. □

**Lemma 34** (Bias terms 2). *Recall we define $y_h(a^{[i]})$ and $y_h(a^{[0]})$ in Lemma 33. Then, we have*

$$\mathbb{E}[\mathbb{I}(\|\bar{z}_h\| \leq Z_1)u_h(\bar{z}_h, a_h, r_h, o_{h+1}; \theta); a_{1:h-1} \sim \pi', a_h \sim \pi(\bar{z}_j)] = y_h(a^{[0]}) + \sum_i y_h(a^{[i]}).$$

*Thus,*

$$\mathbb{E}[l_h(\bar{z}_h, a_h, r_h, o_{h+1}; \theta, \pi)a_{1:h-1} \sim \pi', a_h \sim U(1 + d^\diamond)]$$

$$= \mathbb{E}[u_h(\bar{z}_h, a_h, r_h, o_{h+1}; \theta); a_{1:h-1} \sim \pi', a_h \sim \pi(\bar{z}_j)] + \frac{\text{poly}(\mathbb{C}, d_s, d_a, d_o, H, \|O^\dagger\|)}{m}.$$

*Proof.*

**First Statement**  We have

$$\mathbb{E}[\mathbb{I}(\|\bar{z}_h\| \leq Z_1)u_h(\bar{z}_h, a_h, r_h, o_{h+1}; \theta); a_{1:h-1} \sim \pi', a_h \sim \pi(\bar{z}_j)]$$
$$= \mathbb{E}[\mathbb{I}(\|\bar{z}_h\| \leq Z_1)\mathbb{E}[u_h(\bar{z}_h, a_h, r_h, o_{h+1}; \theta) \mid \bar{z}_h, s_h, a_h]; a_{1:h-1} \sim \pi', a_h \sim \pi(\bar{z}_j)]$$
$$= \mathbb{E}[\mathbb{I}(\|\bar{z}_h\| \leq Z_1)\mathbb{E}[u_h(\bar{z}_h, \pi_h(\bar{z}_h), r_h, o_{h+1}; \theta) \mid \bar{z}_h, s_h, a_h = \pi_h(\bar{z}_h)]; a_{1:h-1} \sim \pi'].$$

Here, by some algebra, there exists a vector $c_2$

$$\mathbb{E}[u_h(\bar{z}_h, a_h, r_h, o_{h+1}; \theta) \mid \bar{z}_h, s_h, a_h] = \langle c_2, [1, [\bar{z}_h^\top, s_h^\top, a_h^\top] \otimes [\bar{z}_h^\top, s_h^\top, a_h^\top]]^\top \rangle.$$

Thus, there exists $c_0$ and a vector $c_1$ such that

$$\mathbb{E}[u_h(\bar{z}_h, a_h, r_h, o_{h+1}; \theta) \mid \bar{z}_h, s_h, a_h] = c_0(\bar{z}_h, s_h) + c_1^\top(\bar{z}_h, s_h)\kappa(a_h)$$

Recall we can write

$$\kappa(\pi_h(\bar{z}_h)) = \sum_{i=1}^{d^\diamond} \alpha_i(\pi_h(\bar{z}_h))\kappa(a^{[i]})$$

Using the above,

$$\mathbb{E}[\mathbb{I}(\|\bar{z}_h\| \leq Z_1)u_h(\bar{z}_h, a_h, r_h, o_{h+1}; \theta); a_{1:h-1} \sim \pi', a_h \sim \pi(\bar{z}_j)]$$
$$= \mathbb{E}[\mathbb{I}(\|\bar{z}_h\| \leq Z_1)\{c_0(\bar{z}_h, s_h) + c_1^\top(\bar{z}_h, s_h)\}\kappa(\pi_h(\bar{z}_h)); a_{1:h-1} \sim \pi']$$
$$= \mathbb{E}\mathbb{I}(\|\bar{z}_h\| \leq Z_1)\{c_0(\bar{z}_h, s_h) + \sum_i c_1^\top(\bar{z}_h, s_h)\alpha_i(\pi_h(\bar{z}_h))\kappa(a^{[i]})\}; a_{1:h-1} \sim \pi']$$
$$= \mathbb{E}[\mathbb{I}(\|\bar{z}_h\| \leq Z_1)[c_0(\bar{z}_h, s_h) +$$
$$+ \sum_i \alpha_i(\pi_h(\bar{z}_h))\{\mathbb{E}[\kappa(\bar{z}_h, \pi_h(\bar{z}_h), r_h, o_{h+1}) \mid \bar{z}_h, s_h, a_h = a^{[i]}] - c_0(\bar{z}_h, s_h)\}]; a_{1:h-1} \sim \pi']$$
$$= \mathbb{E}\left[\mathbb{I}(\|\bar{z}_h\| \leq Z_1)\left[c_0(\bar{z}_h, s_h) - \sum_i \alpha_i(\pi_h(\bar{z}_h))c_0(\bar{z}_h, s_h)\right]\right] + \sum_i y_h(a^{[i]}).$$

Besides,

$$c_0(\bar{z}_h, s_h) = \mathbb{E}[u_h(\bar{z}_h, a_h, r_h, o_{h+1}; \theta) \mid \bar{z}_h, s_h, a_h = do(0)].$$

Thus,

$$\mathbb{E}\left[\mathbb{I}(\|\bar{z}_h\| \leq Z_1)\left[c_0(\bar{z}_h, s_h) - \sum_i \alpha_i(\pi_h(\bar{z}_h))c_0(\bar{z}_h, s_h)\right]\right]$$

$$= \mathbb{E}[\mathbb{I}(\|\bar{z}_h\| \leq Z_1)\{1 - \sum_i \alpha_i(\pi_h(\bar{z}_h))\}\mathbb{E}[u_h(\bar{z}_h, a_h, r_h, o_{h+1}; \theta) \mid \bar{z}_h, s_h, a_h = 0]; a_{1:h-1} \sim \pi']$$

$$= \mathbb{E}[\mathbb{I}(\|\bar{z}_h\| \leq Z_1)\{1 - \sum_i \alpha_i(\pi_h(\bar{z}_h))\}u_h(\bar{z}_h, a_h, r_h, o_{h+1}; \theta); a_{1:h-1} \sim \pi', a_h = do(0)]$$

$$= y_h(a^{[0]}).$$

In conclusion,

$$\mathbb{E}[\mathbb{I}(\|\bar{z}_h\| \leq Z_1)u_h(\bar{z}_h, a_h, r_h, o_{h+1}; \theta); a_{1:h-1} \sim \pi', a_h \sim \pi(\bar{z}_j)] = y_h(a^{[0]}) + \sum_i y_h(a^{[i]}).$$

**Second Statement** As we see in the proof of Lemma 33, the following term

$$\mathbb{E}[\{\mathbb{I}(\|\bar{z}_h\| \leq Z_1) - 1\}u_h(\bar{z}_h, a_h, r_h, o_{h+1}; \theta); a_{1:h-1} \sim \pi', a_h \sim \pi(\bar{z}_j)]$$

is upper-bounded by $\mathrm{poly}(\mathbb{C}, d_s, d_o, d_a, H, \ln(m))/m$. Hence,

$\mathbb{E}[l_h(\bar{z}_h, a_h, r_h, o_{h+1}; \theta, \pi)]$

$$= \mathbb{E}[\hat{y}_h(a^{[0]})] + \sum_{i=1}^{d^\diamond} \mathbb{E}[\hat{y}_h(a^{[i]})] \qquad\qquad\qquad \text{(Definition)}$$

$$= y_h(a^{[0]}) + \sum_{i=1}^{d^\diamond} y_h(a^{[i]}) + \mathrm{poly}(\mathbb{C}, d_s, d_o, d_a, H, \ln(m))/m \qquad \text{(Statement of Lemma 33)}$$

$$= \mathbb{E}[\mathbb{I}(\|\bar{z}_h\| \leq Z_1)u_h(\bar{z}_h, a_h, r_h, o_{h+1}; \theta); a_{1:h-1} \sim \pi', a_h \sim \pi_h(\bar{z}_h)] + \mathrm{poly}(\mathbb{C}, d_s, d_o, d_a, H, \ln(m))/m$$
$$\qquad\qquad\qquad\qquad\qquad\qquad\qquad\qquad\qquad\qquad\qquad \text{(First statement)}$$

$$= \mathbb{E}[u_h(\bar{z}_h, a_h, r_h, o_{h+1}; \theta); a_{1:h-1} \sim \pi', a_h \sim \pi_h(\bar{z}_h)] + \mathrm{poly}(\mathbb{C}, d_s, d_o, d_a, H, \ln(m))/m$$

$$= \mathrm{Br}_h(\pi, \theta; \pi') + \mathrm{poly}(\mathbb{C}, d_s, d_o, d_a, H, \ln(m))/m.$$

$\square$

## M.4 Sample Complexity

Summarizing results so far, we have

$$\sup_{\pi \in \Pi, \theta \in \Theta} |\mathbb{E}_{\mathcal{D}}[l_h(\bar{z}_h, a_h, r_h, o_{h+1}; \theta, \pi)\}] - \mathrm{Br}_h(\pi, \theta; \pi')|$$

$$\leq \mathrm{poly}(\ln(m), d_s, d_o, d_a, \mathbb{C}, H, \|O^\dagger\|) \times \sqrt{\ln(1/\delta)/m}.$$

This is enough to invoke Theorem 1. Here, recall we have

$$\|X_h(\pi)\| \leq \mathrm{poly}(H, d_o, d_a, d_s, \mathbb{C}, \Theta, \|O^\dagger\|), \quad \|W_h(\pi)\| \leq \mathrm{poly}(H, d_o, d_a, d_s, \mathbb{C}, \Theta, \|O^\dagger\|).$$

for any $\pi \in \Pi$ using Lemma 30. In addition, we showed the PO-bilinear rank is

$$\mathrm{poly}(H, d_o, d_a, d_s).$$

Then, using Theorem 1, the sample complexity is

$$\tilde{O}\left(\mathrm{poly}(\ln(m), d_s, d_o, d_a, \mathbb{C}, \Theta, H, \|O^\dagger\|, \ln(1/\delta)) \times \frac{1}{\epsilon^2}\right).$$

# N  Sample Complexity in PSRs

To focus on the main point, we just use a one-step future. We first show the form of future-dependent value functions to set a proper class for $\mathcal{G}_h$. Next, we show the PO-bilinear decomposition.

We assume the following assumptions.

**Assumption 11.** *(a)* $\mathcal{T} \subset \mathcal{O}$ *is a core test and* $\mathcal{Q}$ *is a minimum core rest, (b)* $\|\mathrm{vec}(\mathbb{J}_h^\pi)\| \leq \Theta$ *for any* $\pi \in \Pi$ *where* $\mathbb{J}_h^\pi$ *is in* $\mathcal{V}_h^\pi(\tau_h) = \boldsymbol{I}(z_{h-1})^\top \mathbb{J}_h^\pi \mathbf{q}_{\tau_h}$.

## N.1  Existence of Link Functions

Recall $V_h^\pi(\tau_h) = \mathbf{1}(z_{h-1})^\top \mathbb{J}_h^\pi \mathbf{q}_{\tau_h}$, where we use $\mathbf{1}(z) \in \mathbb{R}^{|\mathcal{O}|^M |\mathcal{A}|^M}$ to denote the one-hot encoding vector over $\mathcal{Z}_{h-1}$, and $\mathbb{J}_h^\pi$ is a matrix in $\mathbb{R}^{|\mathcal{O}|^M |\mathcal{A}|^M \times |\mathcal{T}|}$.

Then, $g_h^\pi(z_{h-1}, o) := \mathbf{1}(z_{h-1})^\top \mathbb{J}_h^\pi [\mathbf{1}(t = o)]_{t \in \mathcal{T}}$ is a value future-dependent value function. This is because

$$\mathbb{E}[g_h(z_{h-1}, o) \mid \tau_h] = \mathbb{E}[\mathbf{1}(z_{h-1})^\top \mathbb{J}_h^\pi [\mathbf{1}(t = o)]_{t \in \mathcal{T}} \mid \tau_h]$$
$$= \mathbf{1}(z_{h-1})^\top \mathbb{J}_h^\pi \mathbf{q}_{\tau_h}.$$

Hence, we set $\mathcal{G}_h$ to be

$$\{(z_{h-1}, o) \mapsto \mathbf{1}(z_{h-1})^\top \mathbb{J}[\mathbf{1}(t = o)]_{t \in \mathcal{T}} : \|\mathrm{vec}(\mathbb{J})\| \leq \Theta\}$$

so that the realizability holds.

## N.2 PO-bilinear Rank Decomposition

We show that PSR admits PO-bilinear rank decomposition (Definition 6). Here is the Bellman loss:

$$\mathbb{E}[\{g_{h+1}(z_h, o_{h+1}) + r_h\} - g_h(z_{h-1}, o_h); a_{1:h-1} \sim \pi', a_h \sim \pi].$$

To analyze the above, we decompose the above into three terms:

$$\underbrace{\mathbb{E}[g_{h+1}(z_h, o_{h+1}); a_{1:h-1} \sim \pi', a_h \sim \pi]}_{(a)} + \underbrace{\mathbb{E}[r_h; a_{1:h-1} \sim \pi', a_h \sim \pi]}_{(b)} + \underbrace{\mathbb{E}[-g_h(z_{h-1}, o_h); a_{1:h-1} \sim \pi', a_h \sim \pi]}_{(c)}.$$

Let $\mathcal{Q}$ be a minimum core test. Here, for any future $t$, there exists $\tilde{m}_t$ such that $\mathbb{P}(t \mid \tau_h) = \langle \tilde{m}_t, \tilde{\mathbf{q}}_{\tau_h} \rangle$ where $[\mathbb{P}(\cdot \mid \tau_h)]_{|\mathcal{Q}|}$ is a $|\mathcal{Q}|$-dimensional predictive state $\tilde{\mathbf{q}}_{\tau_h}$. This satisfies

$$\mathbb{P}(o_h \mid \tau_h; a_h)\tilde{\mathbf{q}}_{\tau_h, a_h, o_h} = \tilde{M}_{o_h, a_h} \tilde{\mathbf{q}}_{\tau_h}. \tag{23}$$

where $\tilde{M}_{o_h, a_h}$ is a matrix whose $i$-th row is $\tilde{m}_{o_h, a_h}^\top$ as we see in Section E.

**Term (c).** We have

$$\mathbb{E}[g_h(z_{h-1}, o_h) \mid \tau_h] = \mathbf{1}(z_{h-1})^\top \mathbb{J}\mathbb{E}[[\mathbf{1}(t = o_h)]_{t \in \mathcal{T}} \mid \tau_h]$$
$$= \mathbf{1}(z_{h-1})^\top \mathbb{J}\mathbb{J}_1 \tilde{\mathbf{q}}_{\tau_h}$$

where $\mathbb{J}_1 \in \mathbb{R}^{|\mathcal{T}| \times |\mathcal{Q}|}$ is a matrix whose $i$-th row is $\tilde{m}_t^\top$. The existence of $\mathbb{J}_1$ is ensured since $\mathcal{Q}$ is a core test.

**Term (b).** We have

$$\mathbb{E}[r_h \mid \tau_h; a_h \sim \pi] = \sum_{o_h, a_h} \pi(a_h \mid o_h, z_{h-1})r_h(a_h, o_h)\mathbb{P}(o_h \mid \tau_h; a_h)$$
$$= \sum_{o_h, a_h} \pi(a_h \mid o_h, z_{h-1})r_h(a_h, o_h)\langle \tilde{m}_{o_h, a_h}, \tilde{\mathbf{q}}_{\tau_h} \rangle$$
$$= \mathbf{1}(z_{h-1})^\top \mathbb{J}_2^\pi \tilde{\mathbf{q}}_{\tau_h}$$

for some matrix $\mathbb{J}_2^\pi$. In the first inequality, we use the reward is a function of $o_h, a_h$ conditioning on the whole history. From the first line to the second line, we use a property of core tests.

**Term (a).** We have

$$\mathbb{E}[g_{h+1}(z_h, o_{h+1}) \mid \tau_h; a_h \sim \pi] = \mathbb{E}[\mathbf{1}(z_h)^\top \mathbb{J}[\mathbf{1}(t = o_{h+1})]_{t \in \mathcal{T}} \mid \tau_h; a_h \sim \pi]$$
$$= \mathbb{E}[\mathbf{1}(z_h)^\top \mathbb{J}\mathbb{J}_3 \tilde{q}_{\tau_h, a_h, o_h} \mid \tau_h; a_h \sim \pi]$$

for some matrix $\mathbb{J}_3$. Then, the above is further equal to

$$\sum_{a_h, o_h} \mathbf{1}(z_h)^\top \mathbb{J}\mathbb{J}_3 \pi(a_h \mid z_{h-1}, o_h)\mathbb{P}(o_h \mid \tau_h; a_h)\tilde{\mathbf{q}}_{\tau_h, a_h, o_h}$$
$$= \sum_{a_h, o_h} \mathbf{1}(z_h)^\top \mathbb{J}\mathbb{J}_3 \pi(a_h \mid z_{h-1}, o_h)\tilde{M}_{o_h, a_h}\tilde{\mathbf{q}}_{\tau_h}$$
$$= \mathbf{1}(z_{h-1})^\top \mathbb{J}_4^\pi \tilde{\mathbf{q}}_{\tau_h}$$

for some matrix $\mathbb{J}_4^\pi$. From the first line to the second line, we use $\mathbb{P}(o_h \mid \tau_h; a_h)\tilde{\mathbf{q}}_{\tau_h, a_h, o_h} = \tilde{M}_{o_h, a_h}\tilde{\mathbf{q}}_{\tau_h}$ in (23).

**Summary.** Combining all terms, there exists a matrix $\mathbb{J}_5^\pi$ such that

$$\mathbb{E}[\{g_{h+1}(z_h, o_{h+1}) + r_h\} - g_h(z_{h-1}, o_h); a_{1:h-1} \sim \pi', a_h \sim \pi]$$
$$= \mathbf{1}(z_{h-1})^\top \mathbb{J}_5^\pi \mathbb{E}[\tilde{\mathbf{q}}_{\tau_h}; a_{1:h-1} \sim \pi']$$
$$= \langle \mathrm{Vec}(\mathbb{J}_5^\pi), \mathbf{1}(z_{h-1}) \otimes \mathbb{E}[\tilde{\mathbf{q}}_{\tau_h}; a_{1:h-1} \sim \pi'] \rangle$$

Here, we suppose $\|\mathrm{Vec}(\mathbb{J}_5^\pi)\| \le \Theta_W$ for any $\pi$. Besides,

$$\|\mathbf{1}(z_{h-1}) \otimes \mathbb{E}[\tilde{\mathbf{q}}_{\tau_h}; a_{1:h-1} \sim \pi']\|_2 \le \|\mathbb{E}[\tilde{\mathbf{q}}_{\tau_h}; a_{1:h-1} \sim \pi']\|_2 \le \mathbb{E}[\|\tilde{\mathbf{q}}_{\tau_h}\|_2; a_{1:h-1} \sim \pi']$$
$$\le \mathbb{E}[\|\tilde{\mathbf{q}}_{\tau_h}\|_1; a_{1:h-1} \sim \pi'] = 1.$$

Thus, we can set $B_X = 1$.

### N.3 Sample Complexity

Suppose $\Pi, \mathcal{G}$ are finite and rewards at $h$ lie in $[0, 1]$. Assume the realizability holds. Then,

$$\epsilon_{gen} = c \max(\Theta, 1)|\mathcal{A}|\sqrt{\ln(|\mathcal{G}_{\max}||\Pi_{\max}|TH/\delta)/m}.$$

Following the calculation in Section I, the sample complexity is

$$\tilde{O}\left(\frac{|\mathcal{O}|^{2(M-1)}|\mathcal{A}|^{2(M-1)}|\mathcal{Q}|^2\max(\Theta, 1)H^4|\mathcal{A}|^2\ln(|\mathcal{G}_{\max}||\Pi_{\max}|/\delta)\ln(\Theta_W)^2}{\epsilon^2}\right).$$

Here, there is no explicit dependence on $|\mathcal{T}|$. Note the worst-case sample complexity of $\ln|\mathcal{G}_{\max}|$ is $O(|\mathcal{Z}_{h-1}||\mathcal{T}|)$ and the worse-case sample complexity of $\ln|\Pi_{\max}|$ is $O(|\mathcal{Z}_{h-1}||\mathcal{O}||\mathcal{A}|)$.

### N.4 Most General Case

We consider the general case in Section E. Let $\mathcal{G}_h$ be a function class consisting of $\mathbf{1}(z_{h-1})^\top \mathbb{J}_h \mathbf{1}(t)$ where $\mathbb{J}_h$ satisfies $\mathbb{J}_h \in \mathbb{R}^{\mathcal{Z}_{h-1}\times|\mathcal{T}|}$ and $\|\text{vec}(\mathbb{J}_h)\| \leq \Theta$. When the realizability holds, we would get

$$\tilde{O}\left(\frac{|\mathcal{O}|^{2(M-1)}|\mathcal{A}|^{2(M-1)}|\mathcal{Q}|^2|\mathcal{T}^{\mathcal{A}}|^2\max(\Theta, 1)H^4|\mathcal{A}|^2\ln(|\mathcal{G}_{\max}||\Pi_{\max}|/\delta)\ln(B_X B_W)^2}{\epsilon^2}\right).$$

Here, there is no explicit sample complexity of $|\mathcal{T}^{\mathcal{O}}|$. Note the worse-case sample complexity of $\ln|\mathcal{G}_{\max}|$ is $O(|\mathcal{Z}_{h-1}||\mathcal{T}|)$ and the worst-case sample complexity of $\ln|\Pi_{\max}|$ is $O(|\mathcal{Z}_{h-1}||\mathcal{O}||\mathcal{A}|)$.

## O  Proof of Theorem 8

We fix the parameters as in Theorem 8. Let

$$l_h(\tau_h, a_h, r_h, o_{h+1}; f, \pi, g) = |\mathcal{A}|\pi_h(a_h \mid \bar{z}_h)\{r_h + g_{h+1}(\bar{z}_{h+1}) - g_h(\bar{z}_h)\}f(\tau_h) - 0.5f(\tau_h)^2.$$

From the assumption, Then, with probability $1 - \delta$, we have $\forall t \in [T], \forall h \in [H]$

$$\sup_{\pi\in\Pi, g\in\mathcal{G}, f\in\mathcal{F}} |\mathbb{E}_{\mathcal{D}_h^t}[l_h(\tau_h, a_h, r_h, o_{h+1}; f, \pi, g)] - \mathbb{E}[\mathbb{E}_{\mathcal{D}_h^t}[l_h(\tau_h, a_h, r_h, o_{h+1}; f, \pi, g)]]| \leq \epsilon_{gen}, \tag{24}$$

$$\sup_{g_1\in\mathcal{G}_1} |\mathbb{E}_{\mathcal{D}^0}[g_1(o_1)] - \mathbb{E}[\mathbb{E}_{\mathcal{D}^0}[g_1(o_1)]]| \leq \epsilon_{ini}. \tag{25}$$

We first show the following lemma. Recall $\pi^\star = \text{argmax}_{\pi\in\Pi} J(\pi)$.

**Lemma 35** (Optimism). *Set $R := \epsilon_{gen}$. For all $t \in [T]$, $(\pi^\star, g^{\pi^\star})$ is a feasible solution of the constrained program. Furthermore, we have $J(\pi^\star) \leq \mathbb{E}[g_1^t(o_1)] + 2\epsilon_{ini}$ for any $t \in [T]$, where $g^t$ is the value link function selected by the algorithm in iteration $t$.*

*Proof.* For any $\pi$, we have

$$\max_{f\in\mathcal{F}_h} |\mathbb{E}[\mathbb{E}_{\mathcal{D}_h^t}[l_h(\tau_h, a_h, r_h, o_{h+1}; f, \pi, g^\pi)]]| = 0$$

since $g^\pi$ is a value link function in $\mathcal{G}$ noting the condition (c) in Definition 9. Thus,

$$\max_{f\in\mathcal{F}_h} |\mathbb{E}_{\mathcal{D}_h^t}[l_h(\tau_h, a_h, r_h, o_{h+1}; f, \pi^\star, g^{\pi^\star})]| \leq \epsilon_{gen}$$

using (24) noting $\pi^\star \in \Pi, g^{\pi^\star} \in \mathcal{G}$. Hence, $(\pi^\star, g^{\pi^\star})$ is a feasible set for any $t \in [T]$ and any $h \in [H]$.

Then, we have

$$
\begin{aligned}
J(\pi^\star) = \mathbb{E}[g_1^{\pi^\star}(o_1)] &\leq \mathbb{E}_{\mathcal{D}^0}[g_1^{\pi^\star}(o_1)] + \epsilon_{ini} && \text{(Uniform convergence result)} \\
&\leq \mathbb{E}_{\mathcal{D}^0}[g_1^t(o_1)] + \epsilon_{ini} && \text{(Using the construction of algorithm)} \\
&\leq \mathbb{E}[g_1^t(o_1)] + 2\epsilon_{ini}. && \text{(Uniform convergence)}
\end{aligned}
$$

$\square$

Next, we prove the following lemma to upper bound the per step regret.

**Lemma 36.** *For any $t \in [T]$, we have*

$$J(\pi^\star) - J(\hat{\pi}) \leq \sum_{h=1}^{H} \left( |\langle W_h(\pi^t, g^t), X_h(\pi^t) \rangle| \right) + 2\epsilon_{ini}.$$

*Proof.*

$$
\begin{aligned}
& J(\pi^\star) - J(\hat{\pi}) \\
& \leq 2\epsilon_{ini} + \mathbb{E}[g_1^t(o_1)] - J(\pi^t) && \text{(From optimism)} \\
& = 2\epsilon_{ini} + \sum_{h=1}^{H} \mathbb{E}[g_h^t(\bar{z}_h) - \{r_h + g_{h+1}^t(\bar{z}_{h+1})\}; a_{1:h} \sim \pi^t] && \text{(Performance difference lemma)} \\
& \leq 2\epsilon_{ini} + \sum_{h=1}^{H} |\mathbb{E}[g_h^t(\bar{z}_h) - \{r_h + g_{h+1}^t(\bar{z}_{h+1})\}; a_{1:h} \sim \pi^t]| \\
& \leq 2\epsilon_{ini} + \sum_{h=1}^{H} |\langle W_h(\pi^t, g^t), X_h(\pi^t) \rangle|. && \text{(From (a) in Definition 3)}
\end{aligned}
$$

$\square$

From Lemma 22, we have

$$\frac{1}{T} \sum_{t=0}^{T-1} \sum_{h=1}^{H} \|X_h(\pi^t)\|_{\Sigma_{t,h}^{-1}} \leq H \sqrt{\frac{d}{T} \ln \left( 1 + \frac{TB_X^2}{d\lambda} \right)}.$$

**Lemma 37.**

$$\|W_h(\pi^t, g^t)\|_{\Sigma_{t,h}}^2 \leq 2\lambda B_W^2 + T\zeta(2\epsilon_{gen}).$$

*Proof.* We have

$$\|W_h(\pi^t, g^t)\|_{\Sigma_{t,h}}^2 = \lambda \|W_h(\pi^t, g^t)\|_2^2 + \sum_{\tau=0}^{t-1} \langle W_h(\pi^t, g^t), X_h(\pi^\tau) \rangle^2.$$

The first term is upper-bounded by $\lambda B_W^2$. The second term is upper-bounded by

$$
\begin{aligned}
& \sum_{\tau=0}^{t-1} \langle W_h(\pi^t, g^t), X_h(\pi^\tau) \rangle^2 \\
& \leq \sum_{k=0}^{t-1} \zeta \left( \max_{f \in \mathcal{F}_h} \left| \mathbb{E}[l_h(\tau_h, a_h, r_h, o_{h+1}; f, \pi^t, g^t); a_{1:M(h)-1} \sim \pi^k, a_{M(h):h} \sim \pi^e(\pi)] \right| \right)^2 \\
& \leq \sum_{k=0}^{t-1} \zeta \left( \max_{f \in \mathcal{F}_h} \left| \mathbb{E}_{\mathcal{D}_h^k}[l_h(\bar{z}_h, a_h, r_h, o_{h+1}; f, \pi^t, g^t)] \right| + \epsilon_{gen} \right)^2 \\
& \leq t\zeta(2\epsilon_{gen})^2.
\end{aligned}
$$

From the first line to the second line, we use (b) in Definition 9. From the second line to the third line, we use $\xi$ is a non-decreasing function. In the last line, we use the constraint on $(\pi^t, g^t)$.

$\square$

Combining lemmas so far, we have

$$J(\pi^\star) - J(\hat{\pi}) \leq \frac{1}{T} \sum_{t=0}^{T-1} \sum_{h=1}^{H} |\langle W_h(\pi^t, g^t), X_h(\pi^t) \rangle| + 2\epsilon_{ini}$$

$$\leq \frac{1}{T} \sum_{t=0}^{T-1} \sum_{h=1}^{H} \|W_h(\pi^t, g^t)\|_{\Sigma_{t,h}} \|X_h(\pi^t)\|_{\Sigma_{t,h}^{-1}} + 2\epsilon_{ini} \qquad \text{(CS inequality)}$$

$$\leq H^{1/2} \left[ 2\lambda B_W^2 + T\zeta^2(2\epsilon_{gen}) \right]^{1/2} \left( \frac{dH}{T} \ln \left( 1 + \frac{TB_X^2}{d\lambda} \right) \right)^{1/2} + 2\epsilon_{ini}.$$

We set $\lambda$ such that $B_X^2/\lambda = B_W^2 B_X^2/\zeta^2(\epsilon_{gen})+1$ and $T = \left\lceil 2Hd \ln(4Hd(B_X^2 B_W^2/\zeta^2(\tilde{\epsilon}_{gen}) + 1)) \right\rceil$.
Then,

$$\frac{Hd}{T} \ln \left( 1 + \frac{TB_X^2}{d\lambda} \right) \leq \frac{Hd}{T} \ln \left( 1 + \frac{T}{d} \left( \frac{B_W^2 B_X^2}{\zeta^2(\epsilon_{gen})} + 1 \right) \right)$$

$$\leq \frac{Hd}{T} \ln \left( 1 + \frac{T}{d} \left( \frac{B_W^2 B_X^2}{\zeta^2(\tilde{\epsilon}_{gen})} + 1 \right) \right)$$

$$\leq \frac{Hd}{T} \ln \left( \frac{2T}{d} \left( \frac{B_W^2 B_X^2}{\zeta^2(\tilde{\epsilon}_{gen})} + 1 \right) \right) \leq 1$$

since $a \ln(bT)/T \leq 1$ when $T = 2a \ln(2ab)$.

Finally, the following holds

$$J(\pi^\star) - J(\pi^T)$$

$$\leq H^{1/2} \left[ 2\lambda B_W^2 + T\zeta^2(2\epsilon_{gen}) \right]^{1/2} + 2\epsilon_{ini}$$

$$\leq H^{1/2} \left[ 2\lambda B_W^2 + 2\zeta^2(2\epsilon_{gen})Hd \ln(4Hd(B_X^2 B_W^2/\zeta^2(\tilde{\epsilon}_{gen}) + 1)) \right]^{1/2} + 2\epsilon_{ini} \qquad \text{(Plug in $T$)}$$

$$\leq H^{1/2} \left[ 4\zeta^2(\epsilon_{gen}) + 2\zeta^2(2\epsilon_{gen})Hd \ln(4Hd(B_X^2 B_W^2/\zeta^2(\tilde{\epsilon}_{gen}) + 1)) \right]^{1/2} + 2\epsilon_{ini}. \qquad \text{(Plug in $\lambda$)}$$

# P  Sample Complexity in $M$-step Decodable POMDPs

We first give a summary of our results. Then, we show that an $M$-step decodable POMDP is a PO-bilinear rank model. After showing the uniform convergence of the loss function with fast rates, we calculate the sample complexity. Since we use squared loss functions, we need to modify the proof of Theorem 1.

## P.1  PO-bilinear Rank Decomposition (Proof of Lemma 17)

In this section, we derive the PO-bilinear decomposition of $M$-step decodable POMDPs (Lemma 17).

First, we define moment matching policies following [19]. We denote $M(h) = h - M$.

**Definition 10** (Moment Matching Policies). *For $h' \in [M(h), h]$, we define*

$$x_{h'} = (s_{M(h):h'}, o_{M(h):h'}, a_{M(h):h'-1}) \in \mathcal{X}_l$$

*where $\mathcal{X}_l = S^l \times \mathcal{O}^l \times \mathcal{A}^{l-1}$ and $l = h' - M(h) + 1$. For an $M$-step policy $\pi$ and $h \in [H]$, we define the moment matching policy $\mu^{\pi,h} = \{\mu_{h'}^{\pi,h} : \mathcal{X}_{h'-M(h)+1} \to \Delta(\mathcal{A})\}_{h'=M(h)}^{h-1}$:*

$$\mu_{h'}^{\pi,h}(a_{h'} \mid x_{h'}) := \mathbb{E}[\pi_{h'}(a_{h'} \mid \bar{z}_{h'}) \mid x_{h'}; \pi].$$

*Note the expectation in the right hand side is taken under a policy $\pi$.*

Using [19, Lemma B.2], we have

$$\text{Br}(\pi, g; \pi') = \mathbb{E}[\{g_h(\bar{z}_h) - r_h - g_{h+1}(\bar{z}_{h+1})\}; a_{1:M(h)-1} \sim \pi', a_{M(h):h} \sim \pi]$$

$$= \mathbb{E}[\{g_h(\bar{z}_h) - r_h - g_{h+1}(\bar{z}_{h+1})\}; a_{1:M(h)-1} \sim \pi', a_{M(h):h-1} \sim \mu^{\pi,h}, a_h \sim \pi]$$

$$= \mathbb{E}[\mathbb{E}[\{g_h(\bar{z}_h) - r_h - g_{h+1}(\bar{z}_{h+1})\} \mid s_{M(h)}; a_{M(h):h-1} \sim \mu^{\pi,h}, a_h \sim \pi]; a_{1:M(h)-1} \sim \pi']$$

$$= \langle X_h(\pi'), W_h(\pi, g) \rangle.$$

where

$$W_h(\pi, g) = \int \mathbb{E}[\{g_h(\bar{z}_h) - r_h - g_{h+1}(\bar{z}_{h+1})\} \mid s_{M(h)}; a_{M(h):h-1} \sim \mu^{\pi,h}, a_h \sim \pi]\mu(s_{M(h)})\mathrm{d}(s_{M(h)}),$$

$$X_h(\pi') = \mathbb{E}[\phi(s_{M(h)-1}, a_{M(h)-1}); a_{1:M(h)-1} \sim \pi'].$$

Thus, the first condition in Definition 9 ((12)) is satisfied

Next, we show the second condition in Definition 9 ((13)). This is proved as follows

$$\frac{0.5}{|\mathcal{A}|^M} \langle X_h(\pi'), W_h(\pi, g) \rangle^2 \tag{26}$$

$$= \frac{0.5}{|\mathcal{A}|^M} \left( \mathbb{E}\left[ (g_h(\bar{z}_h) - (\mathcal{B}_h^\pi g_{h+1})(\bar{z}_h)); a_{1:M(h)-1} \sim \pi', a_{M(h):h-1} \sim \mu^{\pi,h} \right] \right)^2$$

$$\leq \frac{0.5}{|\mathcal{A}|^M} \mathbb{E}\left[ (g_h(\bar{z}_h) - (\mathcal{B}_h^\pi g_{h+1})(\bar{z}_h))^2; a_{1:M(h)-1} \sim \pi', a_{M(h):h-1} \sim \mu^{\pi,h} \right] \qquad \text{(Jensen's inequality)}$$

$$\leq \frac{1}{|\mathcal{A}|^M} \max_{f \in \mathcal{F}_h} \mathbb{E}\left[ (g_h(\bar{z}_h) - (\mathcal{B}_h^\pi g_{h+1})(\bar{z}_h)) f(\bar{z}_h) - 0.5f(\bar{z}_h)^2; a_{1:M(h)-1} \sim \pi', a_{M(h):h-1} \sim \mu^{\pi,h} \right]$$

$$= \frac{1}{|\mathcal{A}|^M} \max_{f \in \mathcal{F}_h} \mathbb{E}\left[ |\mathcal{A}|\pi_h(a_h|\bar{z}_h) (g_h(\bar{z}_h) - r_h - g_{h+1}(\bar{z}_{h+1})) f(\bar{z}_h) - 0.5f(\bar{z}_h)^2; a_{1:M(h)-1} \sim \pi', a_{M(h):h-1} \sim \mu^{\pi,h}, a_h \sim \mathcal{U}(\mathcal{A}) \right]$$

$$\leq \max_{f \in \mathcal{F}_h} \mathbb{E}\left[ |\mathcal{A}|\pi_h(a_h|\bar{z}_h) (g_h(\bar{z}_h) - r_h - g_{h+1}(\bar{z}_{h+1})) f(\bar{z}_h) - 0.5f(\bar{z}_h)^2; a_{1:M(h)-1} \sim \pi', a_{M(h):h} \sim \mathcal{U}(\mathcal{A}) \right]$$

$$= \max_{f \in \mathcal{F}_h} \mathbb{E}\left[ l_h(\bar{z}_h, a_h, r_h o_{h+1}; f, \pi, g); a_{1:M(h)-1} \sim \pi', a_{M(h):h} \sim \mathcal{U}(\mathcal{A}) \right]. \tag{27}$$

From the first line to the second line, we use [19, Lemma B.2]. From the third to the fourth line, we use the Bellman completeness assumption: $-(\mathcal{B}_h^\pi \mathcal{G}) + \mathcal{G}_h \subset \mathcal{F}_h$. From the fourth line to the fifth line, we use importance sampling.

Finally, we show the third condition in Definition 9 (14):

$$\left| \max_{f \in \mathcal{F}_h} \mathbb{E}[l_h(\tau_h, a_h, r_h, o_{h+1}; f, \pi, g^\pi); a_{1:M(h)-1} \sim \pi', a_{M(h):h} \sim \mathcal{U}(\mathcal{A})] \right| = 0. \tag{28}$$

This follows since

$$\mathbb{E}[l_h(\tau_h, a_h, r_h, o_{h+1}; f, \pi, g^\pi); a_{1:M(h)-1} \sim \pi', a_{M(h):h} \sim \mathcal{U}(\mathcal{A})]$$

$$= \mathbb{E}\left[ |\mathcal{A}|\pi_h(a_h|\bar{z}_h) (g_h^\pi(\bar{z}_h) - r_h - g_{h+1}^\pi(\bar{z}_{h+1})) f(\bar{z}_h) - 0.5f(\bar{z}_h)^2; a_{1:M(h)-1} \sim \pi', a_{M(h):h} \sim \mathcal{U}(\mathcal{A}) \right]$$

$$= \mathbb{E}\left[ \mathbb{E}[|\mathcal{A}|\pi_h(a_h|\bar{z}_h) (g_h^\pi(\bar{z}_h) - r_h - g_{h+1}^\pi(\bar{z}_{h+1})) \mid \bar{z}_h]f(\bar{z}_h) - 0.5f(\bar{z}_h)^2; a_{1:M(h)-1} \sim \pi', a_{M(h):h} \sim \mathcal{U}(\mathcal{A}) \right]$$

$$= \mathbb{E}\left[ -0.5f(\bar{z}_h)^2; a_{1:M(h)-1} \sim \pi', a_{M(h):h} \sim \mathcal{U}(\mathcal{A}) \right].$$

### P.2 Uniform Convergence

We define the operator

$$(\mathcal{B}_h^\pi g)(\bar{z}_h) := \mathbb{E}[r_h + g_{h+1}(\bar{z}_{h+1}) \mid \bar{z}_h; a_h \sim \pi].$$

and

$$(\bar{\mathcal{B}}_h^\pi g)(\bar{z}_h) := -(\mathcal{B}_h^\pi g)(\bar{z}_h) + g_h.$$

**Lemma 38** (Uniform Convergence). *Let $|\mathcal{D}| = m$. Suppose $\|\mathcal{F}_h\|_\infty \leq 3H$ for $h \in [H]$. Fix $\pi' \in \Pi$.*

1. *Take a true future-dependent value function $g^\pi \in \mathcal{G}$. Then, it satisfies*

$$\max_{f_h \in \mathcal{F}_h} |\mathbb{E}_{\mathcal{D}}[|\mathcal{A}|\pi_h(a_h \mid \bar{z}_h)\{g_h^\pi(\bar{z}_h) - r_h - g_{h+1}^\pi(\bar{z}_{h+1})\}f_h(\bar{z}_h) - 0.5f_h(\bar{z}_h)^2; a_{1:M(h)-1} \sim \pi', a_{M(h):h} \sim U(\mathcal{A})]$$

$$\leq c_1 \frac{(H|\mathcal{A}|)^2 \ln(|\Pi_{\max}||\mathcal{F}_{\max}||\mathcal{G}_{\max}|/\delta)}{m}.$$

2. *Suppose $g(\pi)$ satisfies*

$$\max_{f_h \in \mathcal{F}_h} |\mathbb{E}_{\mathcal{D}}[|\mathcal{A}|\pi_h(a_h \mid \bar{z}_h)\{g_h(\pi)(\bar{z}_h) - r_h - g_{h+1}(\pi)(\bar{z}_{h+1})\}f_h(\bar{z}_h) - 0.5f_h(\bar{z}_h)^2; a_{1:M(h)-1} \sim \pi', a_{M(h):h} \sim U(\mathcal{A})]|$$

$$\leq \Lambda,$$

*and the Bellman completeness $\bar{\mathcal{B}}_h^\pi \mathcal{G} \subset \mathcal{F}_h (\forall \pi \in \Pi)$ holds. Then, with probability $1 - \delta$, we have*

$$\mathbb{E}[(\bar{\mathcal{B}}_h^\pi g(\pi))^2(\bar{z}_h); a_{1:M(h)-1} \sim \pi', a_{M(h):h-1} \sim U(\mathcal{A})]$$
$$\leq \Lambda + c_2 \frac{(H|\mathcal{A}|)^2 \ln(|\Pi_{\max}||\mathcal{F}_{\max}||\mathcal{G}_{\max}|/\delta)}{m}.$$

*Proof.* To simplify the notation, we define

$$\alpha_h(\bar{z}_h, a_h, r_h, o_{h+1}; g) = \pi_h(a_h \mid \bar{z}_h)|\mathcal{A}|\{g_h(\bar{z}_h) - r_h - g_{h+1}(\bar{z}_{h+1})\}.$$

Given $g \in \mathcal{G}$, we define $\hat{f}_h(\cdot; g)$ as the maximizer:

$$\underset{f_h \in \mathcal{F}_h}{\operatorname{argmax}} |\mathbb{E}_{\mathcal{D}}[|\mathcal{A}|\pi_h(a_h \mid \bar{z}_h)\{g_h(\bar{z}_h) - r_h - g_{h+1}(\bar{z}_{h+1})\}f_h(\bar{z}_h) - 0.5f_h(\bar{z}_h)^2; a_{1:M(h)-1} \sim \pi', a_{M(h):h} \sim U(\mathcal{A})]|.$$

In this proof, the expectation is always taken for the data generating process $\mathcal{D}$. We first observe

$$\mathbb{E}_{\mathcal{D}}[\alpha_h(\bar{z}_h, a_h, r_h, o_{h+1}; g)f_h(\bar{z}_h) - 0.5f_h(\bar{z}_h)^2]$$
$$= 0.5\mathbb{E}_{\mathcal{D}}[\alpha_h(\bar{z}_h, a_h, r_h, o_{h+1}; g)^2 - \{\alpha_h(\bar{z}_h, a_h, r_h, o_{h+1}; g) - f_h(\bar{z}_h)\}^2].$$

Then, we define

$$\mathrm{Er}_h(f, g) := 0.5\{\alpha_h(\bar{z}_h, a_h, r_h, o_{h+1}; g) - f_h(\bar{z}_h)\}^2 - 0.5\{\alpha_h(\bar{z}_h, a_h, r_h, o_{h+1}; g) - (\bar{\mathcal{B}}_h^\pi g)(\bar{z}_h)\}^2].$$

As the first step, we prove with probability $1 - \delta$

$$\forall g; |\mathbb{E}_{\mathcal{D}}[\mathrm{Er}_h(\hat{f}_h(\cdot; g), g)]| \leq \frac{12H|\mathcal{A}| \ln(2|\mathcal{F}_h|||\mathcal{G}_h||\mathcal{G}_{h+1}|/\delta)}{m}. \tag{29}$$

We first fix $g$. Then, from the definition of $\hat{f}_h(\cdot; g)$ and the Bellman completeness $\bar{\mathcal{B}}_h^\pi \mathcal{G} \subset \mathcal{F}_h$, we have

$$\mathbb{E}_{\mathcal{D}}[\mathrm{Er}_h(\hat{f}_h(\cdot; g), g)] \leq 0. \tag{30}$$

Here, we invoke Bernstein's inequality:

$$\forall f \in \mathcal{F}_h; |(\mathbb{E} - \mathbb{E}_{\mathcal{D}})\mathrm{Er}_h(f, g))| \leq \sqrt{\mathbb{E}[\mathrm{Er}_h(f, g)]\frac{\ln(2|\mathcal{F}_h|/\delta)}{m}} + \frac{(6H|\mathcal{A}|)^2 \ln(2|\mathcal{F}_h|/\delta)}{m}. \tag{31}$$

Hereafter, we condition on the above event. Then, combining (30) and (31), we have

$$\mathbb{E}[\mathrm{Er}_h(\hat{f}_h(\cdot; g), g)] \leq \mathbb{E}_{\mathcal{D}}[\mathrm{Er}_h(\hat{f}_h(\cdot; g), g)] + |(\mathbb{E} - \mathbb{E}_{\mathcal{D}})\mathrm{Er}_h(\hat{f}_h(\cdot; g), g)|$$
$$\leq \sqrt{\frac{\mathbb{E}[\mathrm{Er}_h^2(\hat{f}_h(\cdot; g), g)]\ln(2|\mathcal{F}_h|/\delta)(6H|\mathcal{A}|)^2}{m}} + \frac{(6H|\mathcal{A}|)^2 \ln(2|\mathcal{F}_h|/\delta)}{m}.$$

Here, we use

$$\mathbb{E}[\mathrm{Er}_h(\hat{f}_h(\cdot; g), g)] = 0.5\mathbb{E}[\{f_h(\bar{z}_h) - (\bar{\mathcal{B}}_h^\pi g)(\bar{z}_h)\}^2],$$
$$\mathbb{E}[\mathrm{Er}_h(\hat{f}_h(\cdot; g), g)^2] \leq \mathbb{E}[\{f_h(\bar{z}_h) - (\bar{\mathcal{B}}_h^\pi g)(\bar{z}_h)\}^2](6H|\mathcal{A}|)^2 = \mathbb{E}[\mathrm{Er}_h(\hat{f}_h(\cdot; g))](6H|\mathcal{A}|)^2.$$

Thus, by some algebra,

$$\mathbb{E}[\mathrm{Er}_h(\hat{f}_h(\cdot; g), g)] \leq \frac{(12H|\mathcal{A}|)^2 \ln(2|\mathcal{F}_h|/\delta)}{m}.$$

Besides,

$$|\mathbb{E}_{\mathcal{D}}[\mathrm{Er}_h(\hat{f}_h(\cdot; g), g)]|$$
$$\leq \mathbb{E}[\mathrm{Er}_h(\hat{f}_h(\cdot; g))] + |(\mathbb{E} - \mathbb{E}_{\mathcal{D}})[\mathrm{Er}_h(\hat{f}_h(\cdot; g), g)]|$$
$$\leq \mathbb{E}[\mathrm{Er}_h(\hat{f}_h(\cdot; g), g)] + \sqrt{\frac{\mathbb{E}[\mathrm{Er}_h(\hat{f}_h(\cdot; g), g)](6H|\mathcal{A}|)^2 \ln(2|\mathcal{F}_h|/\delta)}{m}} + \frac{(6H|\mathcal{A}|)^2 \ln(2|\mathcal{F}_h|/\delta)}{m}$$
$$\leq \frac{3(12H|\mathcal{A}|)^2 \ln(2|\mathcal{F}_h|/\delta)}{m} + \frac{27H|\mathcal{A}| \ln(2|\mathcal{F}_h|/\delta)}{m}.$$

Lastly, by union bounds over $\mathcal{G}_h, \mathcal{G}_{h+1}$, the statement (29) is proved. Note $\bar{\mathcal{B}}_h^\pi g^\pi = 0$.

**First Statement.**

$$|\mathbb{E}_{\mathcal{D}}[\alpha_h(\bar{z}_h, a_h, r_h, o_{h+1}; g^\pi)\hat{f}_h(\bar{z}_h; g^\pi) - 0.5\hat{f}_h(\bar{z}_h; g^\pi)^2]|$$

$$= |0.5\mathbb{E}_{\mathcal{D}}[\alpha_h(\bar{z}_h, a_h, r_h, o_{h+1}; g^\pi)] - 0.5\mathbb{E}_{\mathcal{D}}[\{\alpha_h(\bar{z}_h, a_h, r_h, o_{h+1}; g^\pi) - f_h(\bar{z}_h)\}^2]|$$

$$\leq c\frac{H|\mathcal{A}|\ln(|\mathcal{F}_h|\mathcal{G}_h||\mathcal{G}_{h+1}/\delta)}{m}.$$

From the second line to the third line, we use (29).

**Second Statement.** Now, we use the assumption on $g(\pi)$:

$$\mathbb{E}_{\mathcal{D}}[\alpha_h(\bar{z}_h, a_h, r_h, o_{h+1}; g(\pi))\hat{f}_h(\bar{z}_h; g(\pi)) - 0.5\hat{f}_h(\bar{z}_h; g(\pi))^2] \leq \Lambda.$$

From what we showed in (29), this implies

$$\mathbb{E}_{\mathcal{D}}[\alpha_h(\bar{z}_h, a_h, r_h, o_{h+1}; g(\pi))(\bar{\mathcal{B}}_h^\pi g(\pi))(\bar{z}_h) - 0.5(\bar{\mathcal{B}}_h^\pi g(\pi))^2(\bar{z}_h)] \leq \Lambda + \frac{3(12H|\mathcal{A}|)^2\ln(|\mathcal{F}_h||\mathcal{G}_h||\mathcal{G}_{h+1}|/\delta)}{m}.$$

Recall we want to upper-bound the error for $\mathbb{E}[0.5(\bar{\mathcal{B}}_h^\pi g(\pi))^2(\bar{z}_h)]$. Here, we use the following observation later:

$$\mathbb{E}[\alpha_h(\bar{z}_h, a_h, r_h, o_{h+1}; g(\pi))(\bar{\mathcal{B}}_h^\pi g(\pi))(\bar{z}_h) - 0.5(\bar{\mathcal{B}}_h^\pi g(\pi))^2(\bar{z}_h)] = \mathbb{E}[0.5(\bar{\mathcal{B}}_h^\pi g(\pi))^2(\bar{z}_h)].$$

We use Bernstein's inequality: with probability $1 - \delta$, for any $g \in \mathcal{G}$,

$$|(\mathbb{E} - \mathbb{E}_{\mathcal{D}})[\alpha_h(\bar{z}_h, a_h, r_h, o_{h+1}; g)(\mathcal{B}_h^\pi g)(\bar{z}_h) - 0.5(\mathcal{B}_h^\pi g)^2(\bar{z}_h)]|$$

$$\leq \sqrt{\frac{\mathbb{E}[(3|\mathcal{A}|H)^2(\mathcal{B}_h^\pi g)^2(\bar{z}_h)]\ln(2|\mathcal{G}_h||\mathcal{G}_{h+1}|/\delta)}{m}} + \frac{(3|\mathcal{A}|H)^2\ln(|\mathcal{G}_h||\mathcal{G}_{h+1}|/\delta)}{m}.$$

Here, we use

$$\mathbb{E}[\{\alpha_h(\bar{z}_h, a_h, r_h, o_{h+1}; g)(\mathcal{B}_h^\pi g)(\bar{z}_h) - 0.5(\mathcal{B}_h^\pi g)^2(\bar{z}_h)\}^2]$$

$$\leq \mathbb{E}[\{\alpha_h(\bar{z}_h, a_h, r_h, o_{h+1}; g)(\mathcal{B}_h^\pi g)(\bar{z}_h) - 0.5(\mathcal{B}_h^\pi g)^2(\bar{z}_h)\}](6|\mathcal{A}|H)^2.$$

Hereafter, we condition on the above event.

Finally, we have

$$\mathbb{E}[0.5(\bar{\mathcal{B}}_h^\pi g(\pi))^2(\bar{z}_h)]$$

$$\leq \mathbb{E}_{\mathcal{D}}[\alpha_h(\bar{z}_h, a_h, r_h, o_{h+1}; g(\pi))(\bar{\mathcal{B}}_h^\pi g(\pi))(\bar{z}_h) - 0.5(\bar{\mathcal{B}}_h^\pi g(\pi))^2(\bar{z}_h)]+$$

$$+ |(\mathbb{E} - \mathbb{E}_{\mathcal{D}})[\alpha_h(\bar{z}_h, a_h, r_h, o_{h+1}; g(\pi))(\bar{\mathcal{B}}_h^\pi g(\pi))(\bar{z}_h) - 0.5(\bar{\mathcal{B}}_h^\pi g(\pi))^2(\bar{z}_h)]|$$

$$\leq \Lambda + \frac{3(12H|\mathcal{A}|)^2\ln(4|\mathcal{F}_h||\mathcal{G}_h||\mathcal{G}_{h+1}|/\delta)}{m}$$

$$+ |(\mathbb{E} - \mathbb{E}_{\mathcal{D}})[\alpha_h(\bar{z}_h, a_h, r_h, o_{h+1}; g(\pi))(\mathcal{B}_h^\pi g(\pi))(\bar{z}_h) - 0.5(\mathcal{B}_h^\pi g(\pi))^2(\bar{z}_h)]|$$

$$\leq \Lambda + \frac{3(12H|\mathcal{A}|)^2\ln(4|\mathcal{F}_h||\mathcal{G}_h||\mathcal{G}_{h+1}|/\delta)}{m} + \sqrt{\frac{\mathbb{E}[0.5(\mathcal{B}_h^\pi g(\pi))^2(\bar{z}_h)]\ln(4|\mathcal{G}_h||\mathcal{G}_{h+1}|/\delta)}{m}} + \frac{\ln(|\mathcal{G}_h||\mathcal{G}_{h+1}|/\delta)}{m}.$$

Hence,

$$\forall \pi \in \Pi, \forall g(\pi); \mathbb{E}[0.5(\mathcal{B}_h^\pi g(\pi))^2(\bar{z}_h)] \leq \Lambda + c\frac{(H|\mathcal{A}|)^2\ln(|\mathcal{F}_h||\mathcal{G}_h||\mathcal{G}_{h+1}|/\delta)}{m}.$$

$\square$

## P.3  Proof of Main Statement

We define

$$|\mathcal{F}_{\max}| = \max_{h \in [H]}|\mathcal{F}_h|, \; |\Pi_{\max}| = \max_{h \in [H]}|\Pi_h|, \; |\mathcal{G}_{\max}| = \max_{h \in [H]}|\mathcal{G}_h|.$$

Let

$$\epsilon_{gen}^2 = c_1 \frac{(H|\mathcal{A}|)^2 \ln(|\Pi_{\max}||\mathcal{F}_{\max}||\mathcal{G}_{\max}|T(H+1)/\delta)}{m},$$

$$\tilde{\epsilon}_{gen}^2 = c_1 \frac{(H|\mathcal{A}|)^2 \ln(|\Pi_{\max}||\mathcal{F}_{\max}||\mathcal{G}_{\max}|(H+1)/\delta)}{m},$$

$$\epsilon_{ini} = c_3 \sqrt{\frac{(H|\mathcal{A}|)^2 \ln(|\mathcal{G}_1|T(H+1)/\delta)}{m}},$$

$$T = 2Hd \ln\left(4Hd\left(\frac{B_X^2 B_W^2}{\tilde{\epsilon}_{gen}} + 1\right)\right), \quad R = \epsilon_{gen}^2.$$

Then, from the first statement in Lemma 38, with probability $1 - \delta$, $\forall t \in [T], \forall h \in [H], \forall \pi \in \Pi$

$$\max_{f_h \in \mathcal{F}_h} |\mathbb{E}_{\mathcal{D}_h^t}[|\mathcal{A}|\pi_h(a_h \mid \bar{z}_h)\{g_h^\pi(\bar{z}_h) - r_h - g_{h+1}^\pi(\bar{z}_{h+1})\}f_h(\bar{z}_h) - 0.5f_h(\bar{z}_h)^2; a_{1:M(h)-1} \sim \pi^t, a_{M(h):h} \sim U(\mathcal{A})]| \tag{32}$$

$$\leq c_1 \frac{(H|\mathcal{A}|)^2 \ln(|\Pi_{\max}||\mathcal{F}_{\max}||\mathcal{G}_{\max}|T(H+1)/\delta)}{m}.$$

Besides, from the second statement in Lemma 38, for $\pi \in \Pi, \forall t \in [T], \forall h \in [H]$, when $g(\pi)$ satisfies

$$\max_{f_h \in \mathcal{F}_h} |\mathbb{E}_{\mathcal{D}_h^t}[|\mathcal{A}|\pi_h(a_h \mid \bar{z}_h)\{g_h(\pi)(\bar{z}_h) - r_h - g_{h+1}(\pi)(\bar{z}_{h+1})\}f_h(\bar{z}_h) - 0.5f_h(\bar{z}_h)^2; a_{1:M(h)-1} \sim \pi^t, a_{M(h):h} \sim U(\mathcal{A})]|$$

$$\leq c_1 \frac{(H|\mathcal{A}|)^2 \ln(|\Pi_{\max}||\mathcal{F}_{\max}||\mathcal{G}_{\max}|T(H+1)/\delta)}{m},$$

we have

$$\mathbb{E}[(\bar{\mathcal{B}}_h^\pi g(\pi))^2(\bar{z}_h); a_{1:M(h)-1} \sim \pi^t, a_{M(h):h-1} \sim U(\mathcal{A})] \leq (c_1 + c_2) \frac{(H|\mathcal{A}|)^2 \ln(|\Pi_{\max}||\mathcal{F}_{\max}||\mathcal{G}_{\max}|TH/\delta)}{m}. \tag{33}$$

We first show the optimism. Recall $\pi^\star = \mathrm{argmax}_{\pi \in \Pi} J(\pi)$.

**Lemma 39** (Optimism). *Set $R = \epsilon_{gen}^2$. For all $t \in [T]$, $(\pi^\star, g^{\pi^\star})$ is a feasible solution of the constrained program. Furthermore, we have $J(\pi^\star) \leq \mathbb{E}[g_1^t(o_1)] + 2\epsilon_{ini}$ for any $t \in [T]$.*

*Proof.* For any $\pi \in \Pi$, letting $g^\pi \in \mathcal{G}$ be a corresponding value future-dependent value function, we have

$$\max_{f \in \mathcal{F}_h} |\mathbb{E}_{\mathcal{D}_h^t}[l_h(\bar{z}_h, a_h, r_h, o_{h+1}; f, \pi, g^\pi)]| \leq \epsilon_{gen}^2.$$

using (32). This implies

$$\forall t \in [T], \forall h \in [H], \max_{f \in \mathcal{F}_h} |\mathbb{E}_{\mathcal{D}_h^t}[l_h(\bar{z}_h, a_h, r_h, o_{h+1}; f, \pi^\star, g^{\pi^\star})]| \leq \epsilon_{gen}^2.$$

Hence, $(\pi^\star, g^{\pi^\star})$ is a feasible set for any $t \in [T]$. Then, we have

$$J(\pi^\star) = \mathbb{E}[g_1^{\pi^\star}(o_1)] \leq \mathbb{E}_{\mathcal{D}_1^t}[g_1^{\pi^\star}(o_1)] + \epsilon_{ini}$$

$$\leq \mathbb{E}_{\mathcal{D}_1^t}[g_1^t(o_1)] + \epsilon_{ini} \leq \mathbb{E}[g_1^t(o_1)] + 2\epsilon_{ini}.$$

$\square$

Next, recall the following two statements. The following statements are proved as before in the proof of Theorem 1.

- For any $t \in [T]$,

$$J(\pi^\star) - J(\pi^t) \leq \sum_{h=1}^{H} |\langle W_h(\pi^t, g^t), X_h(\pi^t)\rangle| + 2\epsilon_{ini}.$$

- Let $\Sigma_{t,h} = \lambda I + \sum_{\tau=0}^{t-1} X_h(\pi^\tau) X_h(\pi^\tau)^\top$. We have

$$\frac{1}{T} \sum_{t=0}^{T-1} \sum_{h=1}^{H} \|X_h(\pi^t)\|_{\Sigma_{t,h}^{-1}}^2 \leq H\sqrt{\frac{d}{T} \ln\left(1 + \frac{TB_X^2}{d\lambda}\right)}.$$

**Lemma 40.**

$$\|W_h(\pi^t, g^t)\|_{\Sigma_{t,h}}^2 \leq 2\lambda B_W^2 + T|\mathcal{A}|^M \epsilon_{gen}^2.$$

*Proof.* We have

$$\|W_h(\pi^t, g^t)\|_{\Sigma_{t,h}}^2 = \lambda \|W_h(\pi^t, g^t)\|_2^2 + \sum_{\tau=0}^{t-1} \langle W_h(\pi^t, g^t), X_h(\pi^\tau) \rangle^2.$$

The first term is upper-bounded by $\lambda B_W^2$. The second term is upper-bounded by

$$\sum_{\tau=0}^{t-1} \langle W_h(\pi^t, g^t), X_h(\pi^\tau) \rangle^2$$

$$\leq |\mathcal{A}|^M \sum_{\tau=0}^{t-1} \mathbb{E}[\mathbb{E}[|\mathcal{A}|\pi_h^t(a_h \mid \bar{z}_h) g_h(\bar{z}_h) - r_h - g_{h+1}(\bar{z}_{h+1}) \mid \bar{z}_h; a_h \sim U(\mathcal{A})]^2; a_{1:M(h)-1} \sim \pi^\tau, a_{M(h):h-1} \sim U(\mathcal{A})]$$

$$= |\mathcal{A}|^M \sum_{\tau=0}^{t-1} \mathbb{E}[(\bar{\mathcal{B}}_h^{\pi^t} g(\pi))^2(\bar{z}_h); a_{1:M(h)-1} \sim \pi^\tau, a_{M(h):h-1} \sim U(\mathcal{A})]$$

$$\leq |\mathcal{A}|^M T(c_1 + c_2) \frac{(H|\mathcal{A}|)^2 \ln(|\Pi_{\max}||\mathcal{F}_{\max}||\mathcal{G}_{\max}|TH/\delta)}{m} \leq T|\mathcal{A}|^M \epsilon_{gen}^2.$$

From the first line to the second line, we use (26). Here, from the third line to the fourth line, we use (33). □

The rest of the argument is the same as the proof in Theorem 1. Finally, the following holds

$$J(\pi^\star) - J(\hat{\pi}) \leq 5\epsilon_{gen}|\mathcal{A}|^{M/2} \left[H^2 d \ln(4Hd(B_X^2 B_W^2/\tilde{\epsilon}_{gen} + 1))\right]^{1/2} + 2\epsilon_{ini}.$$

**Sample Complexity Result.** We want to find $m$ such that

$$\sqrt{\frac{H^2|\mathcal{A}|^{2+M} \ln(|\Pi_{\max}||\mathcal{F}_{\max}||\mathcal{G}_{\max}|TH/\delta)}{m}} [H^2 d \ln(HdB_X^2 B_W^2 m)]^{1/2} \leq \epsilon.$$

where

$$T = Hd \ln(HdB_X^2 B_W^2 m).$$

By organizing terms, we have

$$\sqrt{\frac{H^4 d|\mathcal{A}|^{2+M} \ln(|\Pi_{\max}||\mathcal{F}_{\max}||\mathcal{G}_{\max}|Hd/\delta) \ln(HdB_X^2 B_W^2 m)}{m}} \leq \epsilon.$$

Thus, setting the following $m$ is enough:

$$m = \tilde{O}\left(\frac{H^4 d|\mathcal{A}|^{2+M} \ln(|\Pi_{\max}||\mathcal{F}_{\max}||\mathcal{G}_{\max}|/\delta)}{\epsilon^2}\right).$$

The total sample we use $mTH$ is

$$\tilde{O}\left(\frac{d^2 H^6 |\mathcal{A}|^{2+M} \ln(|\Pi_{\max}||\mathcal{F}_{\max}||\mathcal{G}_{\max}|/\delta)}{\epsilon^2}\right).$$

# Q   Sample Complexity in Observable POMDPs with Latent Low-rank Transition

This section largely follows the one in Section P.

## Q.1 Existence of Value Future-Dependent Value Functions

Since we consider the discrete setting, we can set the value future-dependent value function class as Section K. Hence, we set

$$\mathcal{G}_h = \left\{ \langle \theta, \mathbf{1}(z) \otimes \mathbb{O}^\dagger \mathbf{1}(o) \rangle; \|\theta\|_\infty \le H \right\}.$$

Then, we can ensure $\|\mathcal{G}_h\| \le H/\sigma_1$. Then, from the construction of $\mathcal{F}_h$, we can also ensure $\|\mathcal{F}_h\| \le 4H/\sigma_1$.

## Q.2 PO-bilinear Rank Decomposition (Proof of Lemma 18)

In this section, we derive the PO-bilinear decomposition of observable POMDPs with the latent low-rank transition. We want to prove Lemma 18. Recall $M(h) = \max(h - M, 1)$.

Using [19, Lemma B.2], we have

$$
\begin{aligned}
\mathrm{Br}(\pi, g; \pi') &= \mathbb{E}[\{g_h(\bar{z}_h) - r_h - g_{h+1}(\bar{z}_{h+1})\}; a_{1:M(h)-1} \sim \pi', a_{M(h):h} \sim \pi] \\
&= \mathbb{E}[\{g_h(\bar{z}_h) - r_h - g_{h+1}(\bar{z}_{h+1})\}; a_{1:M(h)-1} \sim \pi', a_{M(h):h-1} \sim \mu^{\pi,h}, a_h \sim \pi] \\
&= \mathbb{E}[\mathbb{E}[\{g_h(\bar{z}_h) - r_h - g_{h+1}(\bar{z}_{h+1})\} \mid s_{M(h)}; a_{M(h):h-1} \sim \mu^{\pi,h}, a_h \sim \pi]; a_{1:M(h)-1} \sim \pi'] \\
&= \langle X_h(\pi'), W_h(\pi, g) \rangle
\end{aligned}
$$

where

$$
\begin{aligned}
W_h(\pi, g) &= \int \mathbb{E}[\{g_h(\bar{z}_h) - r_h - g_{h+1}(\bar{z}_{h+1})\} \mid s_{M(h)}; a_{M(h):h-1} \sim \mu^{\pi,h}, a_h \sim \pi] \mu(s_{M(h)}) \mathrm{d}(s_{M(h)}), \\
X_h(\pi') &= \mathbb{E}[\phi(s_{M(h)-1}, a_{M(h)-1}); a_{1:M(h)-1} \sim \pi'].
\end{aligned}
$$

Thus, the first condition in Definition 9 is satisfied

Next, we show the second condition in Definition 9. This is proved as follows:

$$
\begin{aligned}
&\frac{0.5}{|\mathcal{A}|^M} \langle X_h(\pi'), W_h(\pi, g) \rangle^2 \\
&= \frac{0.5}{|\mathcal{A}|^M} \mathbb{E}\left[ (g_h(\bar{z}_h) - (\mathcal{B}_h^\pi g_{h+1})(\tau_h)); a_{1:M(h)-1} \sim \pi', a_{M(h):h-1} \sim \mu^{\pi,h} \right]^2 \\
&\le \frac{0.5}{|\mathcal{A}|^M} \mathbb{E}\left[ (g_h(\bar{z}_h) - (\mathcal{B}_h^\pi g_{h+1})(\tau_h))^2; a_{1:M(h)-1} \sim \pi', a_{M(h):h-1} \sim \mu^{\pi,h} \right] \\
&\le \frac{1}{|\mathcal{A}|^M} \max_{f \in \mathcal{F}_h} \mathbb{E}\left[ (g_h(\bar{z}_h) - (\mathcal{B}_h^\pi g_{h+1})(\tau_h)) f(\tau_h) - 0.5 f(\tau_h)^2; a_{1:M(h)-1} \sim \pi', a_{M(h):h-1} \sim \mu^{\pi,h} \right] \\
&= \frac{1}{|\mathcal{A}|^M} \max_{f \in \mathcal{F}_h} \mathbb{E}\left[ |\mathcal{A}| \pi_h(a_h|\bar{z}_h) (g_h(\bar{z}_h) - r_h - g_{h+1}(\bar{z}_{h+1})) f(\tau_h) - 0.5 f(\tau_h)^2; a_{1:M(h)-1} \sim \pi', a_{M(h):h-1} \sim \mu^{\pi,h}, a_h \sim \mathcal{U}(\mathcal{A}) \right] \\
&\le \max_{f \in \mathcal{F}_h} \mathbb{E}\left[ |\mathcal{A}| \pi_h(a_h|\bar{z}_h) (g_h(\bar{z}_h) - r_h - g_{h+1}(\bar{z}_{h+1})) f(\tau_h) - 0.5 f(\tau_h)^2; a_{1:M(h)-1} \sim \pi', a_{M(h):h} \sim \mathcal{U}(\mathcal{A}) \right] \\
&= \max_{f \in \mathcal{F}_h} \mathbb{E}\left[ l_h(\tau_h, a_h, r_h, o_{h+1}; f, \pi, g); a_{1:M(h)-1} \sim \pi', a_{M(h):h} \sim \mathcal{U}(\mathcal{A}) \right].
\end{aligned}
$$

From the first line to the second line, we use [19, Lemma B.2]. From the third to the fourth line, we use the Bellman completeness assumption: $-(\mathcal{B}_h^\pi \mathcal{G}) + \mathcal{G}_h \subset \mathcal{F}_h$. From the fourth line to the fifth line, we use importance sampling.

The third condition

$$
\left| \max_{f \in \mathcal{F}_h} \mathbb{E}[l_h(\tau_h, a_h, r_h, o_{h+1}; f, \pi, g^\pi); a_{1:M(h)-1} \sim \pi', a_{M(h):h} \sim U(\mathcal{A})] \right| = 0.
$$

is easily proved.

Finally, the following norm constraints hold:

$$\|W_h(\pi, g)\| \le 3C_{\mathcal{G}}\sqrt{d}, \quad \|X_h(\pi')\| \le 1.$$

**Sample Complexity Result.** Following the same procedure as Section P, here, we want to find $m$ such that

$$\sqrt{\frac{C_{\mathcal{G}}^2 |\mathcal{A}|^{2+M} \ln(|\Pi_{\max}||\mathcal{F}_{\max}||\mathcal{G}_{\max}|TH/\delta)}{m}} [H^2 d \ln(H d B_X^2 B_W^2 m)]^{1/2} \leq \epsilon.$$

where

$$T = H d \ln(H d B_X^2 B_W^2 m).$$

By organizing terms, we have

$$\sqrt{\frac{C_{\mathcal{G}}^2 H^2 d |\mathcal{A}|^{2+M} \ln(|\Pi_{\max}||\mathcal{F}_{\max}||\mathcal{G}_{\max}|Hd/\delta) \ln(H d B_X^2 B_W^2 m)}{m}} \leq \epsilon.$$

Thus, setting the following $m$ is enough

$$m = \tilde{O}\left(\frac{H^4 d |\mathcal{A}|^{2+M} \ln(|\Pi_{\max}||\mathcal{F}_{\max}||\mathcal{G}_{\max}|/\delta)}{\epsilon^2 \sigma_1^2}\right).$$

The total sample we use $mTH$ is

$$\tilde{O}\left(\frac{d^2 H^6 |\mathcal{A}|^{2+M} \ln(|\Pi_{\max}||\mathcal{F}_{\max}||\mathcal{G}_{\max}|/\delta)}{\epsilon^2 \sigma_1^2}\right).$$

Finally, we plug-in $\ln(|\Pi_{\max}||\mathcal{F}_{\max}||\mathcal{G}_{\max}|/\delta) = \ln(|\mathcal{M}|)$.

## R  Exponential Stability for POMDPs with Low-rank Transition

In this section, we prove that the short memory policy is a globall near optimla policy in low-rank MDPs. We first introduce several notation. Next, we prove the exponential stability of Bayesian fileters, which immediately leads to the main statement.

**Notation.** Given a belief $b \in \Delta(\mathcal{S})$, an action and observation pair $(a, o)$, we define the Bayesian update as follows. We define $B(b, o) \in \Delta(\mathcal{S})$ as the operation that incorporates observation $o$, i.e., $b' = B(b, o)$ with $b'(s) = O(o|s)b(s)/(\sum_{\bar{s}} O(o|\bar{s})b(\bar{s}))$, and $\mathbb{T}_a b$ as the operation that incorporates the transition, i.e., $(\mathbb{T}_a b)(s') = \sum_s b(s)\mathbb{T}(s'|s, a)$. Finally, we denote $U(b, a, o)$ as the full Bayesian filter, i.e.,

$$U(b, a, o) = B(\mathbb{T}_a b, o).$$

Let us denote $b_0 \in \Delta(\mathcal{S})$ as the initial latent state distribution. Given the first observation $o_1 \sim \mathbb{O}(\cdot|s), s \sim b_0$, we denote $b_1 = B(b_0, o_1)$ as the initial belief of the system conditioned on the first observation $o_1$. Given two beliefs $b, b'$, we define the distance $D_2(b, b') := \log \mathbb{E}_{s \sim b}[b(s)/b(s')]$

Consider a POMDP whose latent transition is low rank, i.e., $\mathbb{T}(s'|s, a) = \mu(s')^\top \phi(s, a)$. For notation simplicity, we still consider discrete state, action, and observation space to avoid using measure theory languages.

**Design of initial distribution.** We want to design a good distribution for the initial distribution in an artificial Bayesian filter ignoring the history other than the short history.

The following lemma is from [23, Lemma 4.9] that quantifies the contraction of a Bayesian map.

**Lemma 41** (Contraction propery of beliefs). *Suppose $b, b' \in \Delta(\mathcal{S})$ and $\|b/b'\|_\infty < \infty$. Then we have:*

$$\mathbb{E}_{s \sim b, o \sim \mathbb{O}(s)}\left[\sqrt{\exp\left(\frac{D_2(B(b, o), B(b', o))}{4} - 1\right)}\right] \leq (1 - \sigma_1^4/2^{40})\sqrt{\exp\left(\frac{D_2(b, b')}{4}\right) - 1}$$

Next, we compute the G-optimal design using feature $\phi(s, a) : \mathcal{S} \times \mathcal{A} \to \mathbb{R}$. Denote the G-optimal design as $\rho \in \Delta(\mathcal{S} \times \mathcal{A})$. Here, we use assumption $\|\phi(s, a)\| \leq 1$ for any $(s, a)$ in Assumption 6, which ensures that $\phi(s, a)$ lives in a compact space for any $(s, a)$. The property is given as in

**Theorem 7.** In summary, the support of $\rho$ (denoted by $S_\rho$) is at most $d(d+1)/2$ points and for any $\phi(s,a)$, there exists $\alpha(s,a)$ such that

$$\phi(s,a) = \sum_{i=1}^{|S_\rho|} \alpha_i(s,a)\phi(s^i,a^i)\rho^{1/2}(s^i,a^i), \quad \alpha_i(s,a)/\rho^{1/2}(s^i,a^i) \le d \tag{34}$$

where we denote the points on the support $S_\rho$ as $\{s^i,a^i\}_{i=1}^{|S_\rho|}$.

We set our "empty" belief as follows:

$$\tilde{b}_0(\cdot) := \sum_{\tilde{s},\tilde{a}} \rho(\tilde{s},\tilde{a})\mathbb{T}(\cdot|\tilde{s},\tilde{a}) = \sum_{i=1}^{|S_\rho|} \rho(s^i,a^i)\mathbb{T}(\cdot|s^i,a^i).$$

Note that this belief $\tilde{b}_0$ does not depend on any history. We aim to bound $D_2(b,\tilde{b}_0)$ using the following lemma where $b$ is some belief resulting from applying $\mathbb{T}_a$ for any $a$ to a belief $\tilde{b} \in \Delta(\mathcal{S})$. This is a newly introduce lemma.

**Lemma 42** (Distance between the actual belief and the designed initial distribution)**.** *For any distribution $b \in \Delta(\mathcal{S})$ that results from a previous belief $\tilde{b}$ and a one-step latent transition under action $a$, i.e., $b(s) = \mathbb{T}_a\tilde{b}(\tilde{s})$, we have:*

$$D_2(b,\tilde{b}_0) \le \ln(d^3).$$

*Proof.* For any $b \in \Delta(\mathcal{S})$, using its definition, we have:

$$b(s) = \sum_{\tilde{s}} \tilde{b}(\tilde{s})\phi(\tilde{s},a)^\top \mu(s) \qquad\qquad \text{(Definition)}$$

$$= \sum_{\tilde{s}} \tilde{b}(\tilde{s}) \sum_{i=1}^{S_\rho} \alpha_i(\tilde{s},a)\rho^{1/2}(s^i,a^i)\phi(s^i,a^i)^\top \mu(s) \qquad \text{(Property of G-optimal design)}$$

$$= \sum_{i=1}^{S_\rho} \underbrace{\left(\sum_{\tilde{s}} \tilde{b}(\tilde{s})\alpha_i(\tilde{s},a)\rho^{1/2}(s^i,a^i)\right)}_{:=\beta_i} \phi(s^i,a^i)^\top \mu(s)$$

Similarly, the construction of $\tilde{b}_0$ implies that $\tilde{b}_0(s) = \sum_{i=1}^{S_\rho} \rho(s^i,a^i)\phi(s^i,a^i)^\top\mu(s)$, thus, we have:

$$b(s)/\tilde{b}_0(s) = \sum_{i=1}^{S_\rho} \frac{\beta_i\phi(s^i,a^i)^\top\mu(s)}{\sum_{j=1}^{S_\rho}\rho(s^j,a^j)\phi(s^j,a^j)^\top\mu(s)} \le \sum_{i=1}^{S_\rho} \frac{\beta_i\phi(s^i,a^i)^\top\mu(s)}{\rho(s^i,a^i)\phi(s^i,a^i)^\top\mu(s)}$$

$$= \sum_{i=1}^{S_\rho} \frac{\beta_i}{\rho(s^i,a^i)} = \sum_{i=1}^{S_\rho}\sum_{\tilde{s}} \tilde{b}(\tilde{s})\frac{\alpha_i(\tilde{s},a)}{\rho^{1/2}(s^i,a^i)} = \sum_{\tilde{s}} \tilde{b}(\tilde{s})\sum_{i=1}^{S_\rho} \frac{\alpha_i(\tilde{s},a)}{\rho^{1/2}(s^i,a^i)}$$

$$\le \sum_{\tilde{s}} \tilde{b}(\tilde{s})d^3 = d^3. \qquad\qquad \text{(Use propety of G-optimal design(34))}$$

Thus, $D_2(b,\tilde{b}_0) = \ln\left(\mathbb{E}_{s\sim b}\frac{b(s)}{\tilde{b}_0(s)}\right) \le \ln d^3$. $\qquad\square$

Now we prove the exponential stability by leveraging Lemma 42 and Lemma 41.

**Theorem 14** (Exponential stability for POMDPs with Low-rank Latent Transition)**.** *Consider a $t \ge C\gamma^{-4}\ln(d/\epsilon)$. Consider any policy (full history dependent) $\pi$ and a trajectory $a_{1:h+t-1}, o_{1:h+t} \sim \pi$ for $h \ge 1$. Denote $b_{h+t}$ as the (true) belief conditioned on $a_{1:h+t-1}, o_{1:h+t}$. For approximated belief, first for $h = 1$, we define $\bar{b}_{h+t}$ as:*

$$\bar{b}_1 = b_1, \quad \bar{b}_{1+\tau}(o_{1:1+\tau},a_{1:1+\tau-1}) = U(\bar{b}_n(o_{1:\tau},a_{1:\tau-1}),o_{1+\tau},a_{1+\tau-1}), 1 \le \tau \le t;$$

*for $h \ge 2$, we define $\bar{b}_{h+t}$ as:*

$$\bar{b}_h = B(\tilde{b}_0,o_h), \quad \bar{b}_{h+\tau}(o_{h:h+\tau},a_{h:h+\tau-1}) = U(\bar{b}_{h+\tau-1}(o_{h:h+\tau-1},a_{h:h+\tau-2}),o_{h+\tau},a_{h+\tau-1}), 1 \le \tau \le t;$$

*Then we have:*

$$\forall h \ge 1: \quad \mathbb{E}[\|b_{h+t}(o_{1:h+t},a_{1:h+t-1}) - \bar{b}_{h+t}(o_{h:h+t},a_{h:h+t-1})\|_1 ; a_{1:h+t-1} \sim \pi] \le \epsilon.$$

*Proof.* We define

$$Y_{h+n}(o_{1:h+n}, a_{1:h+n-1}) = \sqrt{\exp(D_2(b_{h+n}(o_{1:h+n}, a_{1:h+n-1}), \bar{b}_{h+n}(o_{h:h+n}, a_{h:h+n-1}))/4) - 1}.$$

Hereafter, we omit $(o_{1:h+n}, a_{1:h+n-1})$ to simplify the notation.

We start from the base case $Y_h$ (i.e., $n = 0$).

First case, consider $h > 1$, $b_h = U(b_{h-1}, o_h, a_{h-1})$. Denote $b'_h = \mathbb{T}_{a_{h-1}} b_{h-1}$. From Lemma 42, we know that:

$$\mathbb{E}[D_2(b'_h, \tilde{b}_0) \mid o_{1:h-1}, a_{h-1}; a_{1:h-1} \sim \pi] \le \ln(d^3).$$

Thus, noting $b_h = B(b'_h, o_h)$ and $\bar{b}_h = B(\tilde{b}_h, o_h)$, we have:

$$\mathbb{E}_{o_h \sim \mathbb{O}b'_h} \left[ \sqrt{\exp(D_2(b_h, B(\tilde{b}_0, o_h))/4) - 1} \mid o_{1:h-1}, a_{1:h-1}; a_{1:h-1} \sim \pi \right]$$

$$\le \mathbb{E}_{o_h \sim \mathbb{O}b'_h} \left[ \sqrt{\exp(D_2(b'_h, \tilde{b}_0)/4) - 1} \mid o_{1:h-1}, a_{1:h-1}; a_{1:h-1} \sim \pi \right] \quad \text{(From Lemma 41)}$$

$$\le (1 - \sigma_1^4/2^{40})d^{3/2}$$

which implies the base case:

$$\mathbb{E}[Y_h \mid o_{1:h-1}, a_{h-1}; a_{1:h-1} \sim \pi] \le (1 - \sigma_1^4/2^{40})d^{3/2}.$$

Now for any $n \ge 1$, we have:

$$\mathbb{E}[Y_{h+n} \mid o_{1:h-1}, a_{1:h-1}; a_{1:h+n-1} \sim \pi]$$

$$= \mathbb{E}\left[ \sqrt{\exp\left(D_2(b_{h+n}, \bar{b}_{h+n})/4\right) - 1} \mid o_{1:h-1}, a_{1:h-1}; a_{1:h+n-1} \sim \pi \right]$$

$$\le (1 - \sigma_1^4/2^{40})\mathbb{E}\left[ \sqrt{\exp\left(D_2\left((\mathbb{T}_{a_{h+n-1}} b_{h+n-1}), (\mathbb{T}_{a_{h+n-1}} \bar{b}_{h+n-1})\right)/4\right) - 1} \mid o_{1:h-1}, a_{1:h-1}; a_{1:h+n-1} \sim \pi \right]$$

$$\le (1 - \sigma_1^4/2^{40})\mathbb{E}\left[ \sqrt{\exp\left(D_2\left(b_{h+n-1}, \bar{b}_{h+n-1}\right)/4\right) - 1} \mid o_{1:h-1}, a_{1:h-1}; a_{1:h+n-1} \sim \pi \right]$$

$$\text{(Data processing inequality from [23, Lemma 2.7])}$$

$$= (1 - \sigma_1^4/2^{40})\mathbb{E}[Y_{h+n-1} \mid o_{1:h-1}, a_{1:h-1}; a_{1:h+n-1} \sim \pi].$$

This completes the induction step. Adding expectation with respect to the history $a_{1:h-1}, o_{1:h-1}$ back, we conclude the proof.

When $h = 1$, we simply start with the original belief $b_1$. For any $0 \le n \le t$, we simply set $\bar{b}_{1+n} = b_{1+n}$, thus the conclusion still holds.

$\square$

The above Theorem 14 indicates that in order to approximate the ground truth belief $b_{h+t}$ that is conditioned on the entire history, we only need to apply the Bayesian filter on the M memory $\bar{z}_{h+t}$ starting from a fixed distribution $\tilde{b}_0$. The existence of such $\tilde{b}_0$ is proven by construction where we rely on the low-rankness of the latent transition and a D-optimal design over $\mathcal{S} \times \mathcal{A}$ using the feature $\phi$.

The above Theorem 14 together with the proof of Theorem 1.2 in [23] immediately implies for $M = \Theta(C(\sigma_1)^{-4} \ln(dH/\epsilon))$ (with $C$ being some absolute constant), there must exists an M-memory policy $\pi^\star$, such that $J(\pi^\star_{gl}) - J(\pi^\star) \le \epsilon$ – thus a globally optimal policy can be approximated by a policy that only relies on short memories.

# S  Auxiliary Lemmas

We use the following in Section 4.2.

**Lemma 43** (Useful inequalities)**.**

- 

$$\|AB\| \le \|A\|\|B\|, \|AB\|_F \le \|A\|\|B\|_F$$
$$\mathrm{vec}(aa^\top) = a \otimes a, \|\mathrm{vec}(A)\|_2 = \|A\|_F, \mathrm{Tr}(AB) = \mathrm{vec}(A^\top)^\top \mathrm{vec}(B).$$

- *When A and B are semi positive definite matrices, we have*

$$\mathrm{Tr}(AB) \le \|A\|\mathrm{Tr}(B).$$

The following lemma is useful when we calculate the sample complexity.

**Lemma 44.** *The following is satisfied*

$$\sqrt{\frac{B_1}{m}\ln^2(B_2 m + B_3)} \le c\epsilon$$

*when*

$$m = c\frac{B_1}{\epsilon^2}\{\ln(m(B_2 + B_3 + 1))\}^2.$$

*for some constant c.*