# OpenReview forum: "Provably Efficient Reinforcement Learning in Partially Observable Dynamical Systems"
_NeurIPS.cc/2022/Conference — NeurIPS 2022 Accept_

### Official Review · Reviewer_KJ4g · 2022-07-10

**Rating:** 4
**Confidence:** 1
**Soundness:** 2 fair
**Presentation:** 2 fair
**Contribution:** 2 fair

**Summary:**

The paper derives a bound on the performance error of a POMDP algorithm, where the policy of the algorithm uses a memory-limited observation-action history.

**Questions:**

- Something I would like to ask the authors, would be: How is the presented result useful, when solving a POMDP in practice?

**Limitations:**

Yes, the authors adequately addressed the limitations and potential negative societal impact of their work.

**Strengths And Weaknesses:**

Unfortunately, I expected something completely different when reading the abstract. I was expecting a new algorithm to solve POMDPs using an actor-critic framework with function approximation. The paper, however, comes from the direction of learning theory, where I have hardly any experience. For me coming from a bit more applied research on POMDPs I was very surprised that there was no numerical evaluation at all for this algorithm. The paper was very hard to read for me and I did not understand very much. As such, I can not judge the significance in the area of learning theory, however, from an applied standpoint, I, unfortunately, could not make out any useful thing from this work. Something I was surprised about was, that there was no discussion on optimal filtering, as usual analysis of the POMDP setup transforms the POMDP to an MDP in belief space.

Overall, I would like to leave it to the other reviewers to judge this work.

---

> ### Author Response · Authors · 2022-08-02
> **Response to KJg**
>
> Thank you very much for your comments.
>
> **Something I was surprised about was, that there was no discussion on optimal filtering, as usual analysis of the POMDP setup transforms the POMDP to an MDP in belief space.**
>
> This is a good point. First of all, we cannot perform Bayesian filtering here since the models (transition and omission) are unknown. We think the reviewer might be referring to the existing POMDP planning literature. In planning literature, the reviewer is certainly correct in the sense that the majority of the work uses filtering and operates in the belief space for planning. This work instead focuses on a harder problem: we learn to solve POMDP from scratch.
>
>
> Secondly, while our algorithms do not explicitly use  filtering, some of our analysis implicitly leverages filtering.
> For example, our PAC learning result on tabular POMDP uses the result in [20] which shows Bayesian filter is a  contraction map on tabular observable POMDPs.
>
>
> **Something I would like to ask the authors, would be: How is the presented result useful, when solving a POMDP in practice?**
>
> Designing a general framework for solving a large family of problems is very useful in practice: whenever one faces a POMDP based application, it is unclear if one can precisely determine the underlying structure of the problem a priori, and thus being able to deploy a very general algorithm (like ours) allows us to solve the problem with a much higher chance of being successful. This is exactly the motivation of the goal of this work --- designing a single algorithm that can simultaneously solve many different types of POMDPs.

---

### Official Review · Reviewer_q8PL · 2022-07-11

**Rating:** 6
**Confidence:** 2
**Soundness:** 3 good
**Presentation:** 3 good
**Contribution:** 3 good

**Summary:**

Provably Efficient Reinforcement Learning in Partially Observable Dynamical Systems
This paper proposes a new POMDP formulation that can unify different existing formalisms, e.g. tabular POMDP, observable LQG, and PSRs, and HSE-POMDPs.  This unified formulation can be used with an actor-critic algorithm where a policy is a mapping from history to actions, and critics receives as input both history and future observations. The proposed algorithm PO-bilinear actor-critic framework (PROVABLE) is shown to perform agnostic learning by searching for the best memory-based policy. The paper also provides sample complexity analysis.

**Questions:**

See below

**Limitations:**


The paper pursues an interesting research direction, which tries to unify existing POMDP formalisms. The approach looks very promising. The proposed design of the critic is very interesting. It would become very interesting if the paper can provides some basic empirical results on toy tasks to show all important claim in practice.
	- As the unified framework can now obtain provably efficient learning for most POMDP formalisms. Is there any limitations of its, e.g. can it do the same for any general POMDP formulations (continuous or infinite spaces)?
	- How can one understand agnostic learning?
In Algorithm, is z just defined as historical observations? Or is it in the form of belief?

**Strengths And Weaknesses:**


* Strength:
    - A framework that unify existing POMDP formalisms.
    - The proposed actor critic in which the critic is a function of both history and future observations. This critic function is an interesting idea.
    - Derivations for different special cases, e.g. tabular POMDPs, PSRs, LQG.

* Weaknesses:
    - Lacking of discussions or motivations for the importance of the proposed idea
    - Empirical results: Can be on toy tasks

---

> ### Author Response · Authors · 2022-08-02
> **Response to q8PL**
>
> Thank you very much for your encouraging comments.
>
> **The paper pursues an interesting research direction, which tries to unify existing POMDP formalisms. The approach looks very promising. The proposed design of the critic is very interesting. It would become very interesting if the paper can provides some basic empirical results on toy tasks to show all important claim in practice.**
>
> Thank you very much for your positive comment. Evaluating the proposed theory empirically is certainly on our plan and is certainly one of the ultimate goals of this line of provably efficient POMDP research. At this moment, we do believe our work has a solid contribution as a theoretical work,  since our work is the **first** paper that shows PAC RL (statistically efficient) results in a unified way beyond the tabular setting.
>
> **\newedit As the unified framework can now obtain provably efficient learning for most POMDP formalisms. Is there any limitations of its, e.g. can it do the same for any general POMDP formulations (continuous or infinite spaces)?**
>
> Our approach can handle continuous or infinite spaces as in HSE-POMDPs. What we need is the realizability of value bridge functions and the low-rank property of the Bellman loss ($d$ in Assumption 3 is a moderate number). As we showed in the paper, these assumptions are satisfied in many reasonable POMDP models. In general, for large-scale POMDPs without any assumptions, unfortunately, there exist lower bounds indicating that no algorithm can ever solve them efficiently.
>
> There are some limitations of this framework. First, what happens if the realizability condition does not hold. While we did not investigate in the current manuscript, we believe our approach can handle some model misspecification. We plan to investigate it formally as a future direction. However, we can handle continuous spaces as we did in HSE-POMDPs.
>
> **How can one understand agnostic learning?**
>
> Thanks for pointing this out. This is terminology used in learning theory literature, and we will certainly clarify that in the revised version. In short, this is referring to the fact that we can learn the best policy from a given policy class $\Pi$. In other words, $\Pi$ might not be rich enough to capture the optimal policy, however, as long as it contains some high quality policy, our approach can learn it.
>
> **In Algorithm, is z just defined as historical observations? Or is it in the form of belief?**
>
> Yes. $z$ are historical observations.  It is hard to form the belief since we do not know the models. Thus, our algorithm only relies on purely observable quantities.

---

### Official Review · Reviewer_6BgD · 2022-07-12

**Rating:** 7
**Confidence:** 3
**Soundness:** 3 good
**Presentation:** 3 good
**Contribution:** 3 good

**Summary:**

The paper describes a class of models for decision making under partial observability. It introduces the concept of "value bridge functions", which take both history and future observations as inputs as a means to resolve state uncertainty. Then it defines a model class such that the bridge-function analog of Bellman equations has error bounded by a low-rank inner product. The paper shows that this model class captures several common types of POMDPs and PSRs, and provides an algorithm for near-optimal decision making that has polynomial sample complexity.

**Questions:**

(See numbered items above.)

**Strengths And Weaknesses:**

**Strengths:**
- The concept of value bridge functions is interesting and novel, and is a nice way to resolve state ambiguity using only observable quantities. It feels akin to value-preserving vs. model-preserving abstractions in the state aggregation literature.
- The theoretical results appear to be correct, although I don't have the background to be able to check them as thoroughly as I would like.
- The paper does a great job of clearly communicating the simplest results in the main text, with references to extended versions in the supplementary materials.

**Weaknesses:**
1. The argmax in step 8 of the algorithm seems extremely expensive. Is it really searching over all M-memory policies and all value bridge functions? What's the runtime?
2. It's unclear how much of the theoretical results hinge on being able to collect IID samples. This criticism is of course not unique to this paper, as most theoretical results rely on similar assumptions. Nevertheless, it may limit how effective the approach can be in practice.
3. The sample complexity results rely on an assumption of uniform convergence. What specifically is ensuring that Assumption 2 holds?

**Comments:**
- The paper presents the theoretical results quite well, but all the examples are purely abstract. It would be helpful to have a concrete example, such as the Tiger problem (which does not have a full-column-rank observation matrix), to help ground the discussion.


**Minor Points:**
- Line 29: "middle ground" typo
- Line 145: "optimality" typo
- Line 151: "Given a" typo
- Line 209: "undercompleteness" typo
- Line 228: seems like $K_h$ is missing a `\bar`?
- Line 7 of Algorithm: unbalanced `{`
- Line 305: should be $J(\pi)$, not $J(\pi^\*)$
- Lines 370 and 374: $M$ is overloaded, since it is also used to refer to M-memory policies. I suggest using another symbol here.
- I'm not a huge fan of the name of the algorithm, as "PROVABLE" isn't especially descriptive and it seems like it would be hard to search for.

-----

**Review Summary:**

This paper seems valuable for two reasons: the concept of value bridge functions is a useful way to address state uncertainty without relying on unobservable quantities like states; and the PO-bilinear model class is a concise way to describe a wide range of related partially observable decision problems. While I can only assess the correctness of the approach at a high level, the results seem likely to be of interest to anyone studying sample efficiency under partial observability. Accept.

---

> ### Author Response · Authors · 2022-08-02
> **Response to 6BGD**
>
> Thank you very much for your valuable comments.
>
> **The argmax in step 8 of the algorithm seems extremely expensive. Is it really searching over all M-memory policies and all value bridge functions? What's the runtime?**
>
> This is a great point. The worse case running time is $|\Pi||\mathcal{G}|$, and we acknowledge that it is unclear if our algorithm is computationally efficient in a formal computation complexity perspective (e.g., unclear if it has polynomial complexity or it is NP hard to solve the $\mathrm{argmax}$). Nevertheless, in practice, when $\Pi$ and $\mathcal{G}|l$ consist of differentiable functions, it is not unreasonable that one can approximately solve this constrained optimization problem via Lagrangian and gradient descent. We plan to work on this approximate result in the revised version, i.e., we will show that an $\delta$-approximate optimizer of the $\mathrm{argmax}$ only is going to affect the sub-optimality by $O(\delta)$.
>
> We also want to point out that ensuring a formal polynomial computation complexity in POMDP is extremely difficult even in the planning perspective (i.e., models are known): there are many lower bounds that indicate such hardness results. There are several strategies to overcome such difficulties in POMDP, and our strategy here is to formalize the core optimization problem as a standard constrained optimization problem where in practice we can easily plug in differentiable function approximators (e.g., neural networks), and use whatever the latest optimization techniques to optimize it (e.g., Lagrangian with gradient descent). While this is certainly not the only approach to achieve efficiency in practice, we believe it allows us to integrate the advancement from RL side (e.g., how to do exploration) together with the advancement from differentiable function approximation and stochastic optimization.
>
> **It's unclear how much of the theoretical results hinge on being able to collect IID samples. This criticism is of course not unique to this paper, as most theoretical results rely on similar assumptions. Nevertheless, it may limit how effective the approach can be in practice.**
>
> We suppose the reviewer talk about the sampling procedure of Line 5 in Algorithm 1. This sampling procedure operates under the standard stochastic episodic online RL setting [27,54,29,14]. We understand that we can gain potential efficiency in practice by reusing samples even though it induces some correlation among samples. There is certainly a statistical tradeoff here that potentially can be analyzed. In practice, we think this tradeoff is one of the tuning parameters, i.e., how many samples we should re-use to give the exact i.i.d property.
>
> **The sample complexity results rely on an assumption of uniform convergence. What specifically is ensuring that Assumption 2 holds?**
>
> It essentially says that the policy class and value bridge function class have bounded statistical complexities. This is a very standard assumption used in almost all theory for studying generalization bound of supervised learning. To give some examples here, as remarked in Remark 4, it incurs $\sqrt{\ln(|\Pi||\mathcal{G}|)/n}$ when these two function classes are discrete. When function classes are infinite, these $\ln(|\Pi||\mathcal{G}|)$ are replaced with log covering numbers. When $\Pi$ and $\mathcal{G}$ are some linear functions, this becomes the dimension of the linear function. In tabular models, LQG and HSE-POMDPs, we calculate these log covering numbers and confirm the uniform convergence results in the proof.
>
> **The paper presents the theoretical results quite well, but all the examples are purely abstract. It would be helpful to have a concrete example, such as the Tiger problem (which does not have a full-column-rank observation matrix), to help ground the discussion.**
>
> This is a great suggestion. Even if the observation matrix is not full-column-rank, we conjecture that if we use multi-step future observations as discussed at the end of Section 4.1, we might be able to ensure the $|\mathcal{O}|^K||\mathcal{A}|^{K-1} \times |\mathcal{S}|$ matrix is full-column rank. We plan to investigate it more.  We are happy to add concrete examples in the revised version.
>
> **Minor points**
>
> We appreciate that you catch these typos. We will fix it.

---

> > ### Comment · Reviewer_6BgD · 2022-08-03
> > **Response to authors**
> >
> > Thanks for the response.
> >
> > It would be really interesting to see an approximate version that avoids the worst-case dependence on $|\Pi|$ and $|\mathcal{G}|$. It would also be helpful if there was an acknowledgement of the worse-case running time in the text to make it more obvious to the community where we still need to improve.
> >
> > Regarding IID samples: yes, I was referring to Line 5 of Algorithm 1. Okay, so it sounds like the approach uses a new episode for each sample? I agree that will produce IID samples. The problem is that it may require an inconveniently large number of episodes for rare events to be adequately represented. I've seen some papers use policy covers to make this IID sampling more efficient. Perhaps that concept could help here as well.

---

> > > ### Author Response · Authors · 2022-08-03
> > > **Response to 6BGD**
> > >
> > >
> > > Thank you very much for your quick response. We will incorporate your comments and try to improve the draft/result!

---

### Official Review · Reviewer_JmWL · 2022-07-18

**Rating:** 6
**Confidence:** 1
**Soundness:** 3 good
**Presentation:** 2 fair
**Contribution:** 2 fair

**Summary:**

This paper proposed a model-free actor-critic framework for POMDPs, where the critic is a value bridge function class that uses fixed memory and future observations as inputs, and obtained PAC-guarantees for models such as HSE-POMDPs, and PSRs.

**Questions:**

Is it necessary to include an actor-critic with future observation for solving HSE-POMDP? Could you give a comparison with the method that is used in the paper "Computationally Efficient PAC RL in POMDPs with Latent
Determinism and Conditional Embeddings"?

**Limitations:**

The result still has an exponential dependence on the memory size M.

**Strengths And Weaknesses:**

Strengths: theoretical results for interesting models such as HSE-POMDPs and PSRs. Also, sample complexity does not have exponential dependence on the horizon.

Weakness: the motivation for using future observations feels a bit unclear to me.

---

> ### Author Response · Authors · 2022-08-01
> **Response to JmWL**
>
> Thank you for your incisive comments.
>
> **Weakness: the motivation for using future observations feels a bit unclear to me.**
>
> In POMDPs, we cannot observe latent states and we do not have access to value functions on latent states. Hence, we define value bridge functions as embeddings of value functions onto future observations. Then, we can learn value bridge functions since they take future observations as inputs but not latent states. We will try to clarify this point more.
>
> **Is it necessary to include an actor-critic with future observation for solving HSE-POMDP? Could you give a comparison with the method that is used in the paper "Computationally Efficient PAC RL in POMDPs with Latent Determinism and Conditional Embeddings"?**
>
> To the best of the authors' knowledge, our result is the only algorithm that operates under the assumptions in Example 3. There possibly exists other style approaches (e.g., model-based learning) to solve this model.
>
> Thank you for giving us a heads up on this paper. We will include the comparison in the next version. In general, these two models are not comparable directly. Our model has an assumption on HSE embedding of the latent transition while the work you pointed to does not have such an assumption but instead assumes deterministic latent transition. However, we would like to point out that our framework captures more models while the paper you mentioned only focused on solving a specific model.
>
> **The result still has an exponential dependence on the memory size M.**
>
> This will depend on the structure of the specific problems. This is true if we naively use the policy class consisting of whole $M$-memory policies (i.e., the policy class size in the worst case is $A^M$). However, using the structure of models, we can obtain sample complexities to compete with globally optimal policies without incurring $\exp(H)$ in many POMDP models (where $H$ is the horizon). For instance, in observable tabular POMDPs, with results in [20], we can ensure quasi-polynomial sample complexity to compete with the globally optimal policy since we can set $M=\ln(SH/\epsilon)$ (Line 317). As another example, in observable LQG, we do not incur the exponential dependence at all since we can restrict the policy class to be a linear policy class using the observation that LQG's globally optimal policy is a linear policy (Theorem 4).
> Finally, in several models such as M-step block-step decodable POMDPs, it is known that $A^M$ is unavoidable from the lower bound.

---

### Meta-Review · Area_Chair_TLLH · 2022-08-24

**Recommendation:** Accept
**Confidence:** Certain

**Metareview:**

This paper contributes to advancing our understanding on when and how sample efficient learning is possibly in partially observable dynamical systems. The authors introduces a framework that encompasses a wide range of relevant settings (e.g., LQG, POMDPs, PSR) and propose a general algorithm to solve this general setting and derive theoretical guarantees on its sample complexity. Overall the contribution is novel, non-trivial, and interesting. I encourage the authors to include part of the rebuttal, which effectively clarifies some aspects of the paper and makes it accessible to a broader audience.

**Award:**

No

---

### Decision · Program_Chairs · 2022-09-14

Accept